# Spatiotemporal gene expression and cellular dynamics of the developing human heart

Enikő Lázár ®[1,2,17] ✉, Raphaël Mauron ®[1,17], Žaneta Andrusivová[1,17], Julia Foyer[1], Mengxiao He ®[1], Ludvig Larsson[1], Nick Shakari ®[3], Sergio Marco Salas ®[3], Christophe Avenel ®[4], Sanem Sariyar[5], Jan Niklas Hansen ®[5,6], Marco Vicari[1,7], Paulo Czarnewski[1], Emelie Braun[8], Xiaofei Li ®[9], Olaf Bergmann ®[2,10,11], Christer Sylvén[12], Emma Lundberg[5,6,13,14], Sten Linnarsson ®[8], Mats Nilsson ®[3], Erik Sundström ®[9], Igor Adameyko ®[15,16] & Joakim Lundeberg ®[1] ✉

Heart development relies on topologically orchestrated cellular transitions and interactions, many of which remain poorly characterized in humans. Here, we combined unbiased spatial and single-cell transcriptomics with imaging-based validation across postconceptional weeks 5.5 to 14 to uncover the molecular landscape of human early cardiogenesis. We present a high-resolution transcriptomic map of the developing human heart, revealing the spatial arrangements of 31 coarse-grained and 72 fine-grained cell states organized into distinct functional niches. Our findings illuminate key insights into the formation of the cardiac pacemaker-conduction system, heart valves and atrial septum, and uncover unexpected diversity among cardiac mesenchymal cells. We also trace the emergence of autonomic innervation and provide the first spatial account of chromaffin cells in the fetal heart. Our study, supported by an open-access spatially centric interactive viewer, offers a unique resource to explore the cellular and molecular blueprint of human heart development, offering links to genetic causes of heart disease.

The heart is the central organ of the cardiovascular system, enabling blood circulation throughout the body. Its function relies on coordinated contractions of four chambers and tandem operation of two sets of cardiac valves, tightly regulated by an intrinsic pacemaker-conduction system, and the endocrine and autonomic nervous systems. This structural and functional complexity of the heart is established within the first trimester of intrauterine development[1,2]. Cardiogenic mesoderm emerges during the third postconceptional week (PCW) and evolves into a linear heart tube, which undergoes extensive spatial rearrangements and cellular diversification to form the multilayered, four-chambered heart by PCW 8, with maturation and growth continuing into fetal and postnatal life. In addition to cardiac precursors, extracardiac proepicardial and neural crest cells also contribute to heart formation. Overall, early cardiogenesis follows a precise spatiotemporal progression, disruptions to which can lead to

congenital heart anomalies, highlighting the importance of understanding its governing molecular mechanisms[3,4].

Anatomical changes throughout heart development have been studied in several species, including humans[5–10]. On the molecular tier, single-cell RNA sequencing (scRNA-seq) studies have yielded insights into the cardiogenic sequence in both animal models[11–21] and humans[22–29]; however, these techniques are not able to retain the essential spatial context of the analyzed cells. Importantly, spatial molecular profiling technologies, including spatial transcriptomics, now allow the capture of molecular arrangements across two dimensions within tissue sections. Combining single-cell and spatial approaches enables a more nuanced understanding of cell identities and interactions by considering both their transcriptomic signatures and positional cues within the tissue[30,31]. This strategy was successfully used in a recent study investigating cardiogenesis in chicken[20], the first published

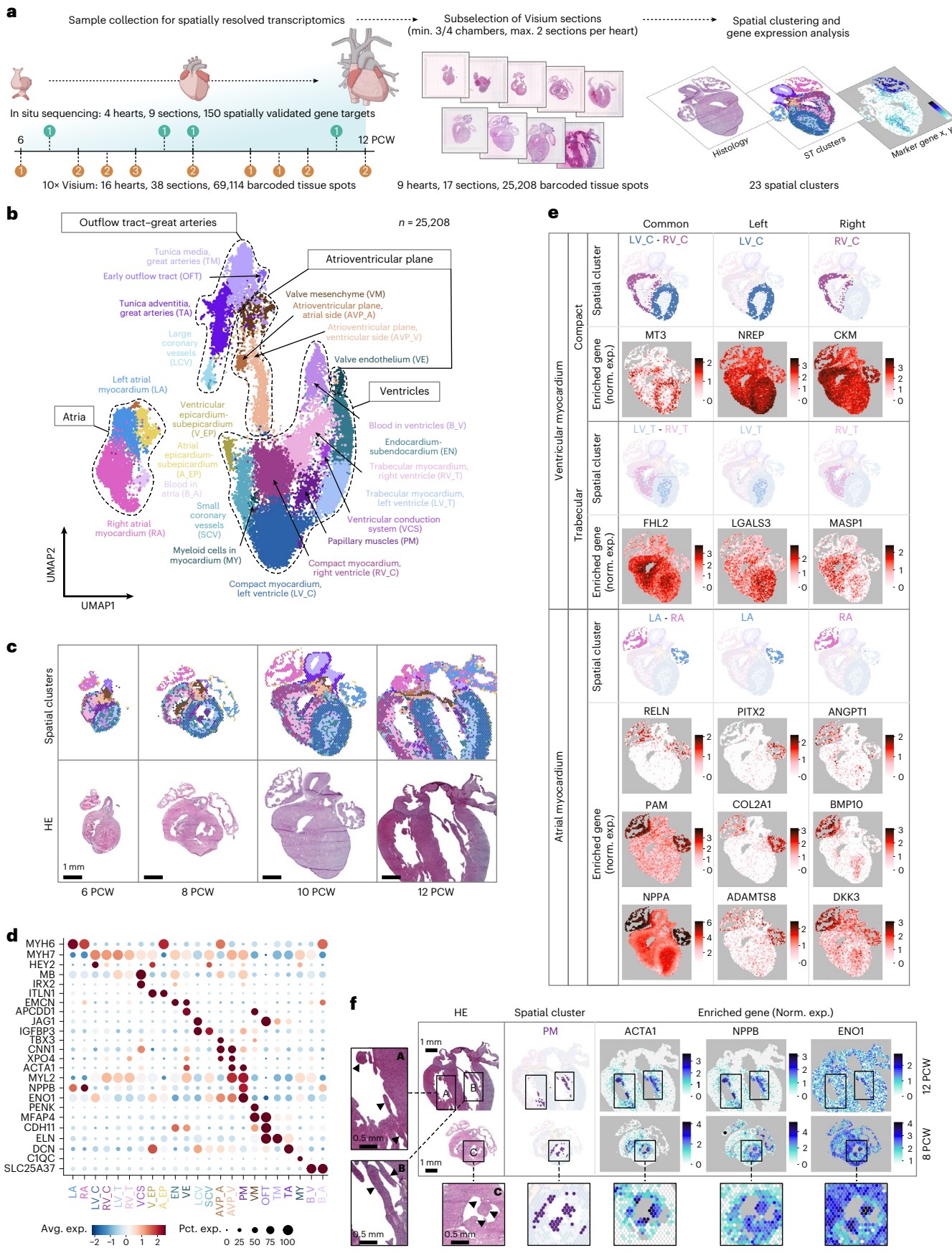

**Fig. 1 | Spatial profiling of the developing human heart. a**, Overview of spatially resolved transcriptomic dataset generation, with donor numbers by PCW indicated in orange (Visium) and green (ISS) circles. **b**, The 23 spatial clusters of the developing heart, corresponding to major cardiac structural components (dashed lines). UMAP, uniform manifold approximation and projection. **c**, Temporal evolution of spatial cluster distribution, presented in 6, 8, 10 and 12 PCW heart sections. HE, hematoxylin & eosin. **d**, Relative enrichment of consensus cell type markers and selected differentially expressed genes (DEGs)

($\log_2FC > 0$, $P < 0.05$) across the 23 spatial clusters. Avg. exp., average expression; Pct. exp., percent of expressing cells; FC, fold change. **e**, Spatial feature plots displaying side-specific enrichment of selected DEGs of the left (LV_C) and right (RV_C) compact, left (LV_T) and right (RV_T) trabecular and left (LA) and right (RA) atrial myocardial spatial clusters in a 10 PCW heart section. **f**, Spatial feature plots illustrating stress-related gene signature in papillary muscles (arrowheads in ROIs A–C), featuring selected DEGs of the corresponding spatial cluster (PM).

spatiotemporal atlas of human heart development[32] and a recent report assessing cellular communities in the fetal human heart[33].

Here, we present a deep spatiotemporal cellular and molecular map of the developing human heart during the late first and early second trimesters. We analyzed 36 hearts between PCWs 5.5 and 14 and assembled an extensive dataset of 69,114 spatially barcoded tissue spots and 76,991 isolated cells, complemented by spatial detection of 150 selected transcripts by in situ sequencing (ISS). We discerned 23 molecular compartments within the cardiac tissue and identified 11 primary cell types and 72 fine-grained cell states, which we mapped to corresponding cardiac tissue regions to enable their refined, spatially aware annotation. We characterized distinct components of the cardiac pacemaker-conduction system (CPCS), investigated their interactions with the emerging autonomic innervation and described a novel cardiac chromaffin cell population. We also investigated position-related endothelial cell (EC) and mesenchymal cell (MC) heterogeneity in the developing cardiac valves and atrial septum and described an array of spatially defined cardiac MC and fibroblast (FB) states. Based on spatial cell coordinates, we delineated the architecture of prominent developmental cardiac niches, enabling their targeted analysis. We present our datasets and results in an interactive viewer (https://hdcaheart.serve.scilifelab.se) to facilitate their independent exploration.

In summary, our work substantially broadens our understanding of early human cardiogenesis, with a primary focus on its spatial context.

## Results

### Spatial and cellular landscape of early human cardiogenesis

To create a comprehensive account of spatial molecular patterns during early cardiogenesis, we performed 10× Genomics Visium spatial transcriptomics analysis on 16 hearts between PCWs 6 and 12, complemented by ISS of 150 selected transcripts in 4 additional hearts. We compiled a dataset of 69,114 tissue spots from 38 heart sections, covering all major structural components of the developing organ (Fig. 1a). To investigate the molecular determinants of regionality, we selected 17 sections encompassing at least three cardiac chambers and performed unsupervised clustering of the corresponding 25,208 tissue spots, resulting in 23 temporally largely consistent spatial clusters with different transcriptomic profiles (Fig. 1a–d, Extended Data Fig. 1a–e, Supplementary Discussion 1 and Supplementary Table 1). This analysis highlighted side-specifically enriched markers of major myocardial compartments and a stress-related molecular signature in the papillary muscles (Fig. 1e,f and Supplementary Discussion 1), further supported by an independent, neighborhood-based region identification approach[34] (Supplementary Fig. 1a–e and Supplementary Discussion 1). Moreover, non-negative matrix factorization on the

entire Visium dataset delineated 20 spatial gene modules, available in our online viewer.

Next, we explored the cellular composition of the developing heart tissue by generating a scRNA-seq dataset from 15 hearts between PCWs 5.5 and 14, using the 10× Genomics Chromium platform. Unsupervised clustering of the 76,991 high-quality cardiac cells defined 31 coarse-grained and 72 fine-grained cell states with distinct transcriptomic profiles (Supplementary Fig. 2a and Supplementary Tables 2–6), creating a common framework for downstream analysis throughout the investigated period. We deconvolved the spatial transcriptomics dataset with these clusters to map their positions within heart sections, informing a spatially aware cell state annotation strategy (Fig. 2a). The predicted spatial localization, expression of canonical cell type markers and exploration of differentially expressed gene profiles enabled the identification of major populations of cardiomyocytes (CMs), endothelial cells (ECs) and non-mural MCs and FBs, along with less abundant populations of neuroblasts and neurons (NB-N), and Schwann cell progenitors and glial cells (SCP-GC), further explored by fine-grained clustering, in addition to other core cardiac cell types (Fig. 2b–d, Extended Data Fig. 2a–c, Supplementary Fig. 2a and Supplementary Table 2). Temporal changes in coarse-grained cluster distribution corresponded to key early cardiogenesis events (Fig. 2e,f, Supplementary Fig. 2b and Supplementary Discussion 1). We also identified and characterized mural cells, such as pericytes (PC) and two spatially distinct smooth muscle cell populations (OFT_SMC and CA_SMC) (Supplementary Fig. 3a–f and Supplementary Discussion 1), in addition to two clusters (TMSB10$^{high}$_C_1–2) enriched in thymosin transcripts, previously implicated in coronary vessel development[20,35]. Furthermore, we detected myeloid (MyC) and lymphoid cell (LyC) populations, along with two red blood cell-dominant clusters that were excluded from downstream analysis (Supplementary Fig. 4a–e and Supplementary Discussion 1), and identified an epicardial cell (EpC) population (Supplementary Fig. 5a–e and Supplementary Discussion 1).

We observed previously unappreciated molecular transitions within the EC and non-mural MC–FB populations (Supplementary Fig. 6a–c and Supplementary Discussion 1) and spatiotemporal patterns in the region of the great arteries and cardiac valves (Supplementary Fig. 7a–c and Supplementary Discussion 1). Enrichment analysis of heart-disease-associated gene panels further confirmed our cluster annotations and highlighted early and widespread expression of genetic determinants in various cardiac pathologies (Supplementary Fig. 8a,b).

### Molecular map of the developing pacemaker-conduction system

CMs are the fundamental contractile units of the myocardium, yet their coordination relies on specialized cells of the CPCS, enabling

**Fig. 2 | Cellular landscape of early human cardiogenesis. a**, Overview of Chromium single-cell dataset generation and spatially informed annotation of cardiac cell states, with donor numbers by PCW shown in blue circles. DGE, differential gene expression. **b**, UMAP representing 31 coarse-grained cardiac single-cell clusters, corresponding to 11 main cell types, including CMs, ECs and non-mural MC–FBs (dashed lines). **c**, Enrichment of canonical cell type markers across coarse-grained single-cell clusters. Dashed rectangles highlight

consensus markers of CMs, ECs and FBs. **d**, Spatiotemporal mapping of selected coarse-grained clusters in 6, 8, 10 and 12 PCW heart sections. **e**, Spatially aware annotation of coarse-grained CM, EC and MC–FB clusters. **f**, Temporal changes in coarse-grained cluster proportions in the CM, EC and MC–FB subsets across four developmental age groups (5.5–6, 7–8, 9–11 and 12–14 PCW). ▲2×, ▲▲3×, ▲▲▲4× increase; ▼2×, ▼▼3×, ▼▼▼4× decrease between the 5.5–6 and 12–14 PCW age groups.

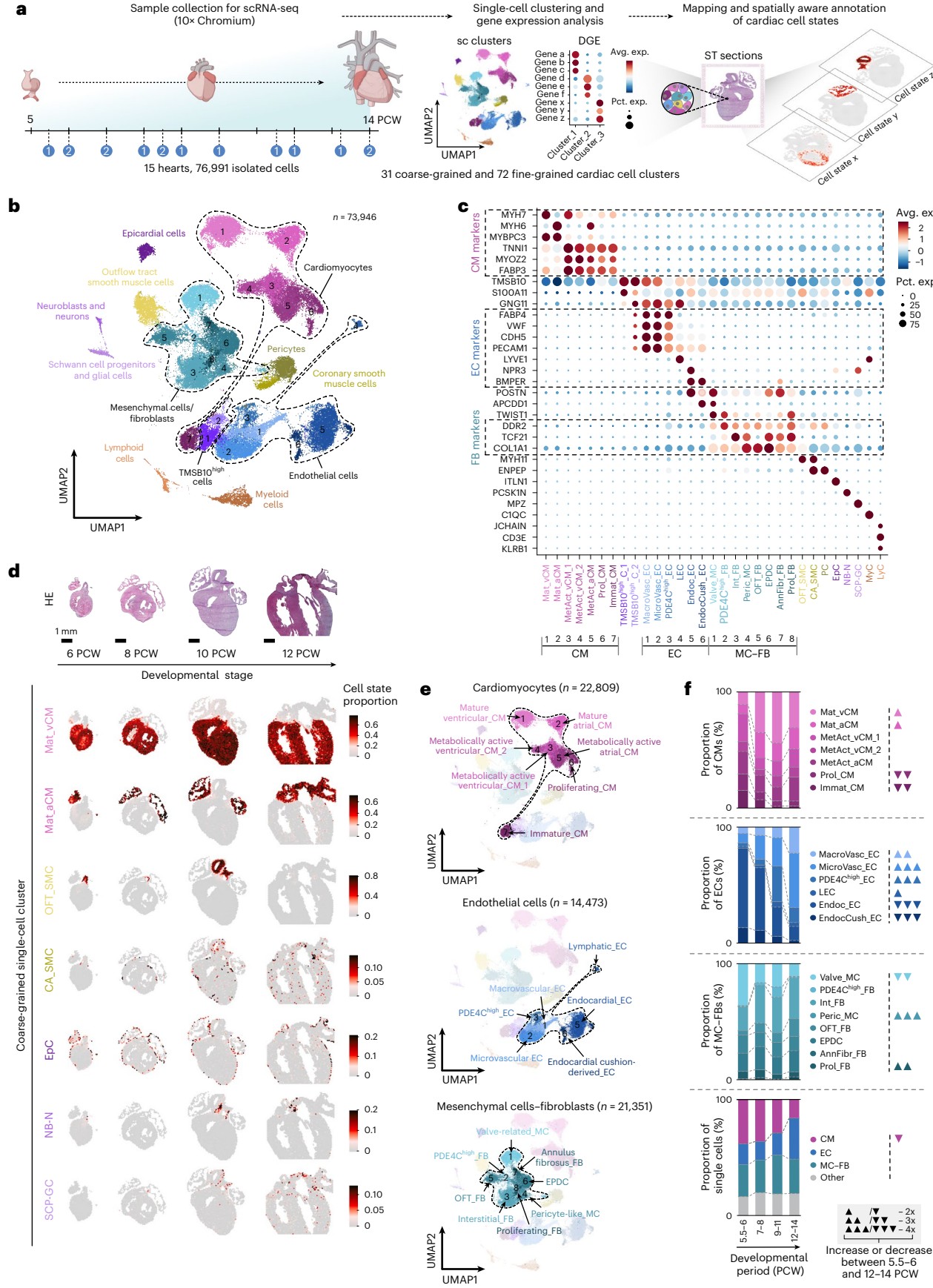

intrinsic automaticity and timely electrical impulse propagation. Molecular characteristics of CPCS cells, previously explored in animal models[36–41] and adult human hearts[42], remain uncharted during human development.

Here, we identified 19 distinct cell states through fine-grained clustering of the CM population, confirming its high transcriptional diversity[43] (Fig. 3a,b, Extended Data Fig. 3a,b and Supplementary Table 3). Among these, we observed clusters with robust expression of previously described CPCS markers[36–42]. Spatial mapping traced these clusters to tissue regions consistent with the position of the sinoatrial node (SAN_CM), atrioventricular node (AVN_CM), atrioventricular bundle and bundle branches (AVB-BB_CM) and Purkinje fibers (PF_CM) (Fig. 3c). Notably, we observed an additional ventricular cluster with shared transcriptional characteristics and close spatial association with PF_CMs, presumably representing transitional PFs (TsPF_CM) (Extended Data Fig. 3c).

CPCS CMs displayed substantial transcriptomic overlaps. SAN_CMs and AVN_CMs expressed transcription factors (*SHOX2, TBX18, PRRX1, ZNF385B*) specifically or strongly enriched in pacemaker cells, as well as axon guidance molecules (*TENM2, TENM3, TENM4, GDF10, SLIT2*), outlining the nodes as essential targets of the developing cardiac innervation (Fig. 3d and Supplementary Fig. 9a,c). We also found the axon guidance molecule partner *LRRC4C*, not yet discussed in the context of the CPCS, specifically expressed in SAN_CMs (Extended Data Fig. 3a). Genes highly enriched in AVN_CMs reflected the node's position in the atrioventricular plane (*BMP2, RSPO3, ADAMTS19*) and neuronal features of these cells (*NRXN3, ZNF536*) (Supplementary Table 3). PF_CMs expressed several characteristic markers (*CSMD1, BRINP3, SGCD, NTN1, SEMA3A*), partly shared with TsPF_CMs (Fig. 3e, Supplementary Fig. 9b,c). Although the recently described atrioventricular bundle markers *CNTN5* and *CRNDE*[42] were more widely distributed in our dataset, several genes enriched in AVB-BB_CMs (*RCAN1, HS3ST3A1*) were specifically concentrated at the upper edge of the ventricular septum, probably marking this structure (Fig. 3e, Extended Data Fig. 3d and Supplementary Fig. 9b). Interestingly, the vCM_1 cluster showed the highest *CNTN5* expression and shared enriched genes with both CPCS (*TBX3, HS3ST3A1, BRINP3, TENM3*) and contractile vCM states (*PRDM16, NLGN1, SORBS2, MYL2*) (Supplementary Table 3). Furthermore, its transcriptomic profile (*XPO4, MYL2, TNFRSF19, SORBS2, ZFP36L1*) and predicted localization in valve roots and the atrioventricular region matched the elusive AVP_V spatial cluster (Extended Data Fig. 3b,e and Supplementary Tables 1 and 3), suggesting a transitional identity probably contributing to atrioventricular conductive tissue formation, as previously described in mice[38].

Electrophysiological properties of CPCS components are shaped by their ion channel repertoire[44] (Fig. 3f and Extended Data Fig. 3f), as recently explored in the adult human heart[42]. In our dataset, developmental SAN_CMs and AVN_CMs featured ion channel profiles typical of nodal cells, with marked enrichment of hyperpolarization-activated cation channels (*HCN1, HCN4*) and several L-type and T-type calcium channel genes (*CACNA1C, CACNA1D, CACNA1G, CACNA2D2, CACNB2*),

all contributing to the depolarization phase of the nodal action potential. Unlike in adults, *CACNA1G* was most enriched in AVN_CMs, while SAN_CMs and ventricular CPCS cell states displayed lower expression. CM repolarization, primarily driven by potassium channels, also showed cell-type-specific patterns. SAN_CMs highly expressed the G-protein-coupled inwardly rectifying potassium channel subunit *KCNJ3* (but not *KCNJ5*), a known marker of adult PF cells, absent in the developmental equivalents in our dataset. By contrast, PF_CMs were enriched in *KCNJ2* and *KCNH7*. AVN_CMs showed high levels of *KCNQ1*, previously associated with atrioventricular block[45,46], as well as ion channel subunits shared with contractile CMs (for example, *CACNA1C, KCNQ3, KCNQ5*), suggesting a hybrid electrophysiological profile. Although *KCND2* and the functionally related *KCND3*, enriched in the adult atrioventricular bundle[42], showed low expression in our dataset, their known interactor *KCNIP4* was highly enriched in vCM_1 cells, further supporting the plausible role of this cell state in the development of atrioventricular conductive tissue (Extended Data Fig. 3a). Gap junction genes followed adult-like patterns, with the low-conductance Cx45 (GJC1) enriched in SAN_CMs and AVN_CMs, and high-conductance Cx40 (GJA5) marking PF_CMs[42].

In addition to several chloride, volume-regulated anion and transient receptor potential channels with as-yet unclear relevance in CPCS function, we also observed enrichment of several ionotropic glutamate receptor genes (*GRID2, GRIK1*) in SAN_CMs, supporting a glutamatergic signaling machinery proposed in the adult node[42], and the GABA receptor ion channel *GABRB2* gene in PFs, specific to the developmental phase (Fig. 3f and Extended Data Fig. 3f).

To investigate the molecular drivers of pacemaker phenotype specification, we inferred a gene regulatory network of SAN_CMs (Fig. 3g). Key transcription factors (*SHOX2, PRDM6, THRB, FOXP2*) and ion channel targets (*HCN1, CACNA1D, KCNT2, CACNB2*) showed substantial overlap with a recent analysis of adult SAN_CMs[42]. Other known pacemaker regulators enriched in SAN_CMs included *TBX18, BNC2* and *TBX3*, with the latter exhibiting markedly higher expression in AVN_CMs, AVB-BB_CMs and vCM_1 cells (Supplementary Table 3). TBX3 suppresses the atrial contractile gene program; thus, its broader expression in CPCS components aligns with their progressive fate restriction, previously described in mouse[47,48]. Notably, the SAN_CM regulatory network also included the transcription factors *BHLHE41* and *RORA*, regulators of the mammalian circadian clock, which were proposed to underlie time-of-day variation in arrhythmia susceptibility[49,50].

## Formation of the cardiac innervation and chromaffin cells

Cardiac function is tightly regulated by sympathetic and parasympathetic signals from the autonomic nervous system, transmitted by the intracardiac ganglionated plexi. Cells of this structure originate from the cardiac neural crest, but their developmental pathway in humans remains unclear[51].

To explore the formation of early cardiac innervation, we reclustered the coarse-grained NB-N and SCP-GC populations into ten

---

**Fig. 3 | Molecular map of CPCS development. a**, UMAPs displaying seven coarse-grained and 19 fine-grained CM clusters, representing a range of contractile CMs of ventricular (vCM_1–6) and atrial origins (Left_aCM, Right_aCM and Cond_aCM aligning with smooth-walled conduit atrial regions), cell states of lower maturation levels (Immat_CM_1–3) and with cell cycle signature (Prol_CM_1–2) as well as components of the CPCS. **b**, Feature plots of *MYH7* and *MYH6* outlining ventricular and atrial identities, respectively. **c**, CM components of the CPCS with characteristic markers and predicted localization in an 11 PCW heart section. **d**, Heatmap visualizing per-gene scaled expression of transcription factors and axon guidance molecules in SAN_CMs and AVN_CMs, and mature contractile atrial CM populations (left). Spatial enrichment of these genes in sinoatrial and atrioventricular node regions is presented in a 6.5 PCW heart section by ISS (right). **e**, Heatmap illustrating per-gene scaled expression of CPCS marker genes in AVN-BB_CMs, PF_CMs, TsPF_CMs and vCM_4 cells,

characterized by the highest maturation state among contractile ventricular CM clusters (left). Spatial expression patterns of these genes detected by ISS in a 9 PCW heart section, corroborating their association with the proximal and distal ventricular conduction system (VCS) (right). **f**, Dot plot featuring relative expression of selected ion channels in SAN_CMs and AVN_CMs compared to contractile atrial CM clusters (upper left). Illustration of ion currents of the nodal cell action potential (upper right). Spatial pattern of HCN and T-type Ca$^{2+}$ channel transcripts in 6.5 PCW and 9 PCW heart sections, detected by ISS (lower). **g**, Gene regulatory network of SAN_CMs, including the top ten differentially expressed transcription factors compared to all other CM states and their associated target genes. la, left atrium; ra, right atrium; rv, right ventricle; ao, aorta; I$_f$, funny current; I$_{CaT}$, T-type calcium current; I$_{CaL}$, L-type calcium current; I$_K$, delayed rectifier potassium current; IK$_{ACh}$, acetylcholine-activated potassium current.

fine-grained clusters, including five SCP (*SOX10*, *FOXD3*), a myelinating Schwann cell (My_SC; *MPZ*, *PMP22*, *MBP*), an intermediate bridge state (*ASCL1*), two autonomic neuroblast-neuron clusters (Aut_Neu_1-2; *PRPH*, *STMN2*) and, notably, a cluster enriched in *CHGA*, *CHGB* and *PENK*, marking a local neuroendocrine chromaffin cell population (Chrom_C) (Fig. 4a,b, Extended Data Fig. 4a, Supplementary Fig. 10 and

Supplementary Table 4). In line with the essential role of chromaffin cells in hypoxia response, cardiac Chrom_Cs showed robust expression of molecular hypoxia sensors, such as *EPAS1* (encoding HIF2α), *COX4I2* and *HIGD1C*, recently described in the oxygen-sensing machinery of the carotid body[52] (Fig. 4b). To our knowledge, this is the first in situ observation of intracardiac chromaffin cells in the developing

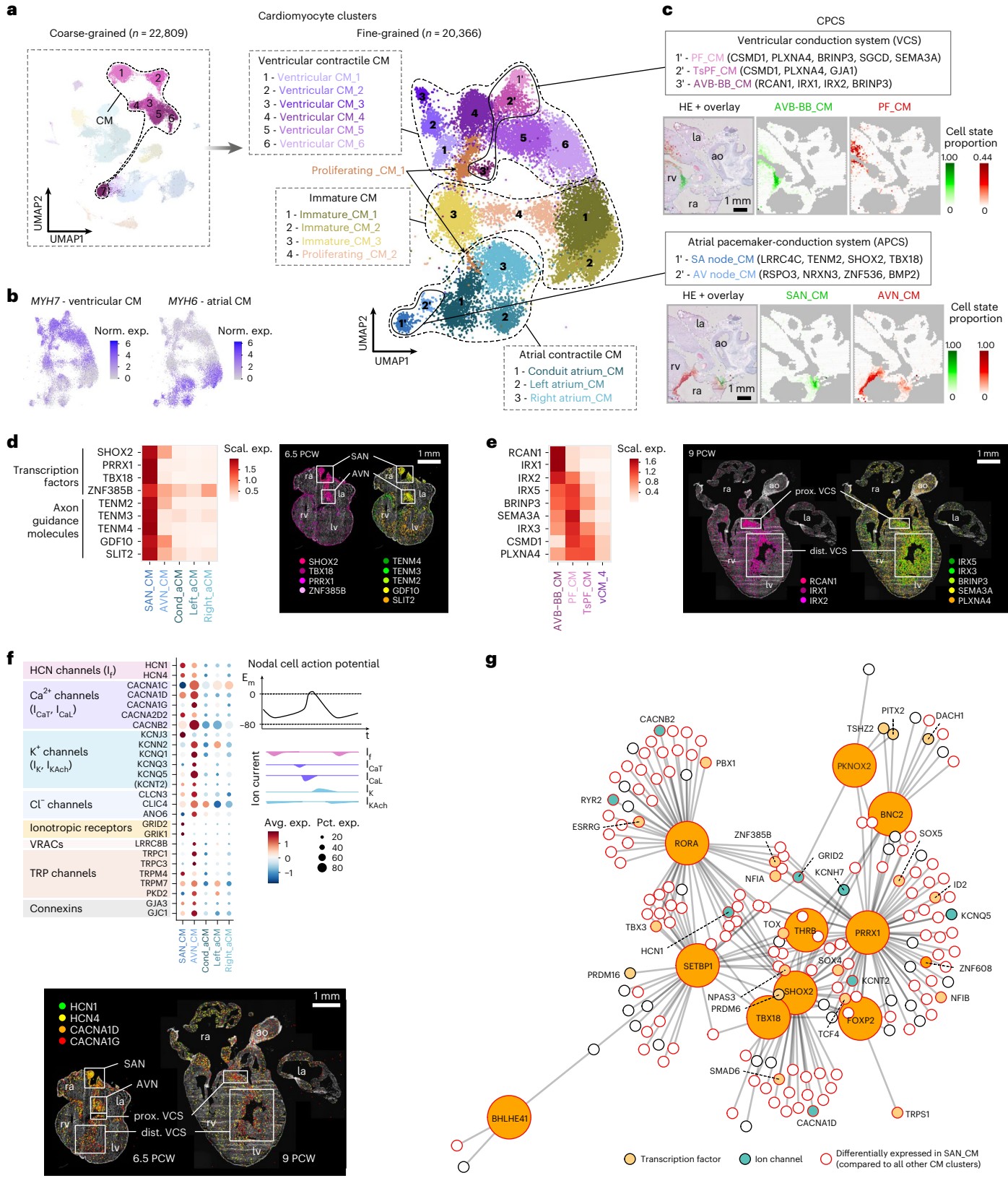

heart, as they are absent in the mouse model and thus appear to be a human-specific cardiac feature. RNA velocity and pseudotime analyses confirmed a fork-like transition from early SCPs towards two parallel trajectories of neuronal-chromaffin and glial cell lineages (Fig. 4c,d and Supplementary Fig. 10).

Chrom_Cs and Aut_Neu_2 cells occupied nearby but distinct regions in the atria and adventitia of great arteries, reflecting their common origin but divergent roles (Fig. 4e and Extended Data Fig. 4b). Meanwhile, Aut_Neu_2-enriched areas near SAN_CMs probably represent neurons that directly innervate the nodal tissue (Fig. 4e). Most SCP states appeared dispersed, except for SCP_5 cells and My_SCs, which concentrated around nerve structures (Extended Data Fig. 4c).

We identified somatostatin (*SST*) as the main neurotransmitter gene expressed in the Aut_Neu_1 and Aut_Neu_2 populations, with cholinergic marker *CHAT* present in a smaller proportion of these cells (Fig. 4b,f). Meanwhile, Chrom_Cs strongly expressed *TH*, and to a lower extent in *DBH* and *PNMT*, encoding key enzymes of catecholamine synthesis (Fig. 4b,f). This indicates that developing intracardiac ganglionated plexi neurons primarily use peptidergic and, to a lesser degree, cholinergic transmission, while local chromaffin cells are the main source of norepinephrine (and to a lesser extent, epinephrine) in the early fetal heart tissue.

Interestingly, we found negligible expression of somatostatin receptors in our datasets; meanwhile, β1 adrenergic receptor (*ADRB1*) was strongly enriched in SAN_CMs and AVN_CMs, and M2 acetylcholine receptor (*CHRM2*) was more broadly expressed across CPCS CMs (Fig. 4f,g). A modified ligand–receptor score, calculated between the Chrom_C, Aut_Neu_2 and SAN_CM states based on coordinated expression levels of key enzymes (*TH* and *CHAT*) and receptors (*ADRB1* and *CHRM2*), reaffirmed distinct sources of adrenergic and cholinergic mediation of sinoatrial node function (Extended Data Fig. 4d).

Additionally, several genes regulating neuronal migration, neurite growth and synapse formation were enriched in SAN_CMs, AVN_CMs, Aut_Neu_1 and Aut_Neu_2 populations (Extended Data Fig. 4e,f). Cell–cell communication analysis between ligands and receptors specifically enriched in SAN_CMs and Aut_Neu_2 cells highlighted the *PTPRS–NTRK3* and *TENM2–ADGRL1* pairs, known for excitatory synapse organization[53] and axon guidance in the central nervous system[54], respectively (Extended Data Fig. 4g). These putative interactions might also be relevant for establishing precise connections between the developing cardiac innervation and pacemaker cells.

Based on these insights, we propose that in the early fetal heart, parasympathetic signals are transmitted by local neurons, while sympathetic modulation is mediated by resident chromaffin cells through paracrine signaling in response to tissue hypoxia. These interactions probably modulate ion channels in CPCS CMs to adapt heart rate and conduction velocity to environmental conditions (Fig. 4h). Furthermore, the presence of intracardiac chromaffin cells may also explain the origin of rare cardiac pheochromocytomas[55].

### Endothelial states in endocardial cushion-derived structures

Cardiac ECs can be divided into vascular and endocardial subsets. The former rapidly expands and diversifies with the swift growth of coronary vessels during the first trimester, while the latter gives rise to endocardial cushions, contributing to the subsequent formation of cardiac valves, atrial septum and the upper membranous septum of the ventricles. Although the major cardiac EC populations were recently characterized in the fetal human heart[28], we still lack a transcriptome-wide description of the scarce endocardial cushion-related ECs and their various derivatives[23,56].

In the merged coarse-grained EC and TMSB10$^{high}$_C_2 clusters, we defined 18 fine-grained cell states (Fig. 5a, Extended Data Fig. 5a–c and Supplementary Table 5), encompassing diverse EC populations from the great arteries and coronary vasculature (Supplementary Fig. 11a,b) and the endocardial lining (Supplementary Fig. 11c). We also observed three clusters with common enrichment of endothelial–mesenchymal transition regulators central to endocardial cushion formation (*NFATC1*, *APCDD1*, *TWIST1*) but also featuring largely distinct transcriptional profiles (Extended Data Fig. 5a,d,e and Supplementary Table 5). Spatial mapping traced two of these cell states to opposite sides of the developing heart valves, outlining inflow and outflow valve ECs (IF_VEC and OF_VECs, respectively) consistently in the atrioventricular and semilunar valves already from PCW 6, suggesting their early specification (Fig. 5b, Extended Data Fig. 5f and Supplementary Fig. 11d). Highly enriched genes of IF_VECs included components of the WNT signaling axis (*WNT2*, *WNT4*, *WNT9B*, *DKK2*, *PTHLH*), with several of them displaying spatial enrichment in the coaptation zone of the valves (Fig. 5c). OF_VECs featured high expression of BMP ligands (*BMP4*, *BMP6*), WNT signaling modulators (*LGR5*) and the endocannabinoid receptor *CNR1*, with strong spatial enrichment on the fibrosa side of the semilunar valves and moderate enrichment on the ventricular side of the atrioventricular valves (Fig. 5c). Meanwhile, the third cluster exhibited highly distinct spatial localization within the atrial septum (AtrSept_EC), marking a previously uncharacterized cardiac EC state (Fig. 5d and Extended Data Fig. 5g). In addition to more specific markers (*NEGR1*, *ALCAM*, *SEMA3A*, *CCDC80*, *MSX1*), AtrSept_ECs shared several strongly enriched genes (*LRRC4C*, *LSAMP*, *OPCML*) with an endocardial population (Endoc_EC_4) which delineates the smooth-walled atrium (Extended Data Fig. 5a,e,g and Supplementary Table 5).

Gene regulatory network and regulon enrichment analysis illuminated further molecular differences between these three cell states (Fig. 5e,f). We identified *MSX1* and *LEF1*—known endothelial–mesenchymal transition mediators during endocardial cushion formation that were recently detected in mouse atrial septum[57,58] and implicated in human atrial septal defects[58–61]—as the most highly enriched transcription factors in AtrSept_ECs. In OF_VECs, we found enrichment of *PROX1* and *FOXC2* transcription factors, known to act in concert in response to oscillatory shear stress, maintaining extracellular matrix structure and preventing myxomatous degeneration of cardiac valves[62]. Several transcription factors enriched in IF_VECs have an established role in modulating WNT signaling, including *KLF4* and *SMAD6*, which are known transducers of laminar shear stress in ECs[63,64]. Importantly, we observed pronounced differences in the regulon enrichment of these transcription factors between IF_VECs and OF_VECs from as early on as PCW 5.5–6, suggesting that hemodynamic forces are core factors in VEC diversification already in the embryonic period.

**Fig. 4 | Early formation of the autonomic cardiac nervous system and resident chromaffin cells. a**, UMAPs showing the NB_N and SCP-GC coarse-grained and ten fine-grained clusters related to the cardiac autonomic innervation.
**b**, Dot plot illustrating the relative expression of glial, neuronal and chromaffin cell differentiation markers, neurotransmitter-related genes and components of the acute oxygen-sensing machinery across all fine-grained innervation-related cell clusters. **c**, RNA velocity and pseudotime analysis revealing two parallel developmental trajectories among innervation-related cell clusters.
**d**, Expression of glial and neuronal differentiation markers. **e**, Spatial mapping of Chrom_Cs (blue), Aut_Neu_2 cells (red) and SAN_CMs (green) in an 11 PCW heart section, outlining their neighboring compartments in the adventitia of the great vessels (ROIs A and B), and an Aut_Neu_2 subset in proximity to the sinoatrial node (ROI C). la, left atrium; ra, right atrium; rv, right ventricle; ao, aorta. **f**, Spatial feature plots of neurotransmitter (NT) metabolism-related genes and receptors in the same section with ROIs A–C in consistent positions to panel **e**. **g**, Dot plot presenting enrichment of β1 adrenergic and M2 cholinergic receptor transcripts in pacemaker-conduction system CMs. **h**, Proposed model of autonomic control of fetal heart function. Blue, green and pink dots beside the displayed genes represent relative enrichment in SAN_CMs, AVN_CMs and PF_CMs, respectively, in comparison to relevant contractile CM states displayed in Fig. 3f and Extended Data Fig. 3f. ICGP, intracardiac ganglionated plexi; CNS, central nervous system; ROS, reactive oxygen species.

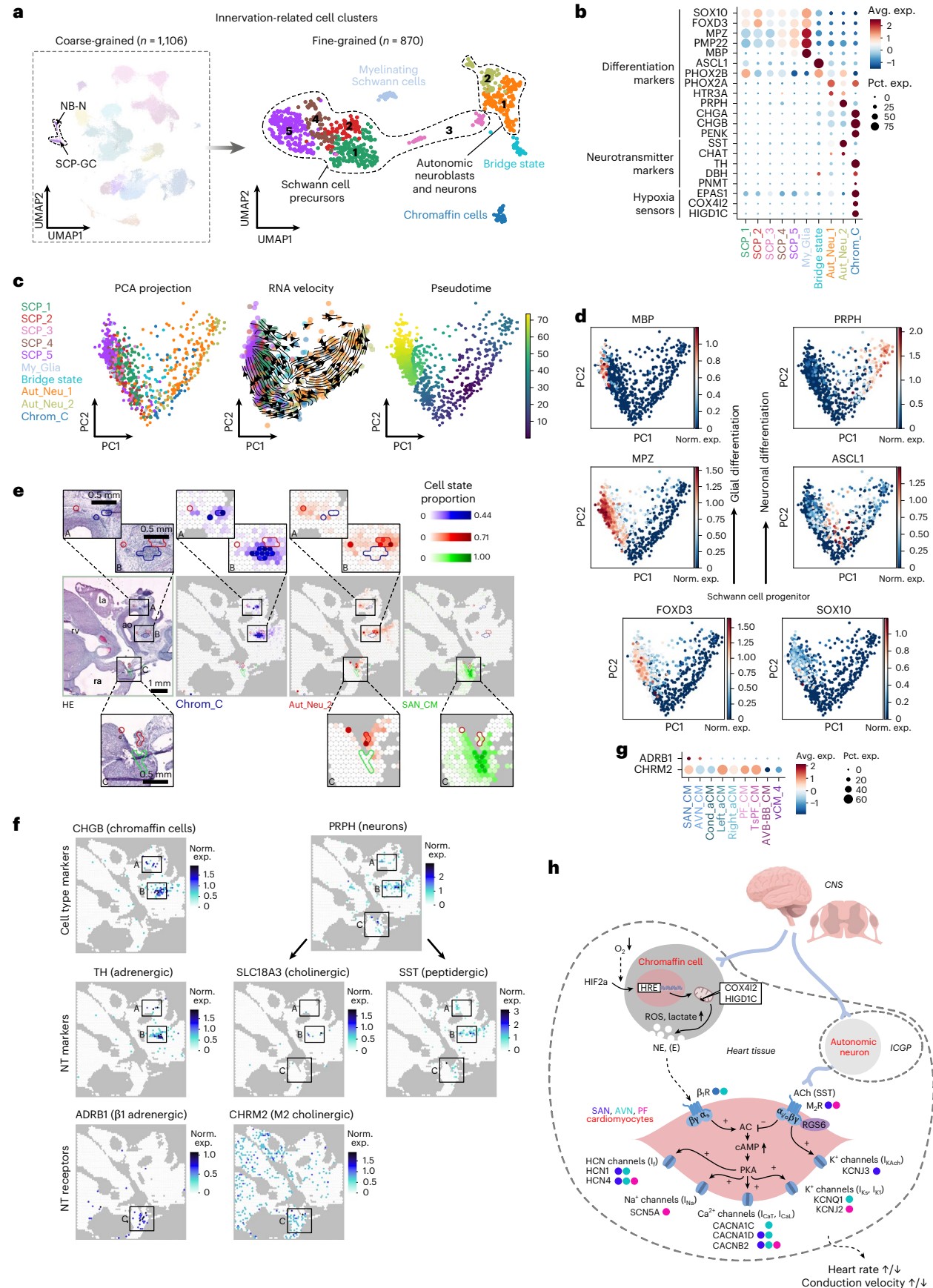

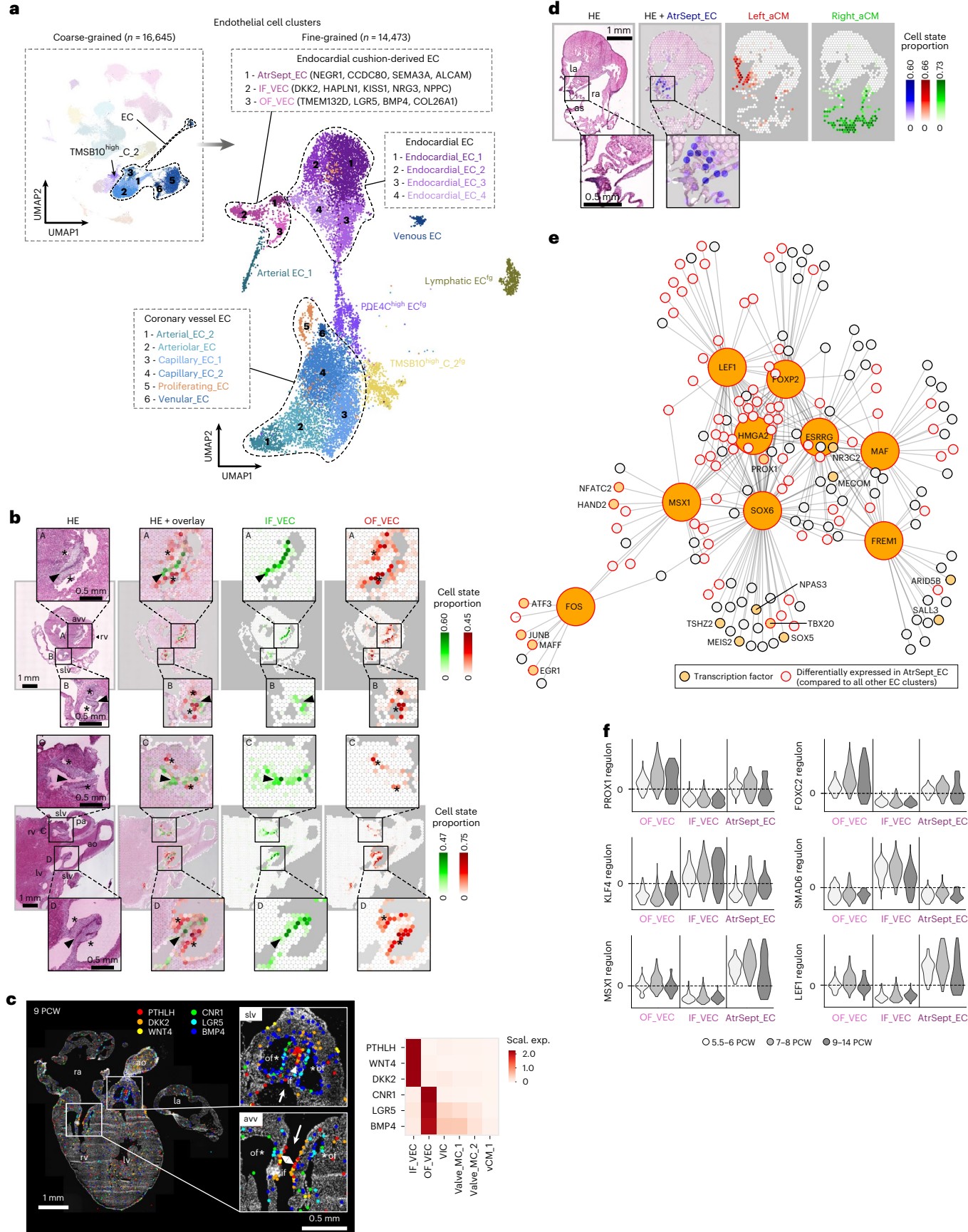

**Fig. 5 | Position-dependent EC diversity in endocardial cushion-derived structures. a**, UMAPs displaying six coarse-grained EC clusters and the TMSB10[high]_C_2 clusters, divided into 18 fine-grained cell states representing endocardial (Endoc_EC_1–4) and endocardial cushion-related cells (OF_VEC, IF_VEC, AtrSept_EC), endothelium of great arteries and large veins (Atr_EC_1, Ven_EC) and consecutive segments of the coronary vasculature, including arterial (Art_EC_2), arteriolar (Arteriol_EC), capillary (Cap_EC_1–2) and venular (Venul_EC) endothelial cells. Additionally, proliferating (Prol_EC) and lymphatic endothelial cells (LEC[fg]), and refined populations of thymosin-enriched (TMSB10[high]_C_2[fg]) and PDE4C-enriched endothelial cells (PDE4C[high]_EC[fg]) are also present. **b**, Spatial mapping of IF_VECs (green) and OF_VECs (red) to opposite sides of the atrioventricular (ROI A) and semilunar valves (ROIs B–D), presented in 8 PCW (upper) and 12 PCW (lower) heart sections. **c**, Side-specific spatial enrichment of selected IF_VEC and OF_VEC markers in atrioventricular and semilunar valves in a 9 PCW heart section detected by ISS (left). White arrows mark the direction of blood flow. Heatmap displaying per-gene scaled expression of the same genes across EC, MC and CM states enriched in the valves (outlined in Fig. 7d) (right). **d**, Spatial mapping of AtrSept_ECs (blue) to the atrial septum between Right_aCM-dominated (green) and Left_aCM-dominated (red) areas in an 8 PCW heart section. **e**, Gene regulatory network of AtrSept_ECs, including transcription factors enriched compared to IF_VECs and OF_VECs, and their associated target genes. **f**, Violin plots illustrating regulon enrichment of selected transcription factors in AtrSept_ECs, IF_VECs and OF_VECs across three age groups (5.5–6, 7–8 and 9–14 PCW). slv, semilunar valve; avv, atrioventricular valve; lv, left ventricle; pa, pulmonary artery; as, atrial septum; arrowhead or if, inflow side of valve; asterisk or of, outflow side of valve.

## Spatial heterogeneity of developmental cardiac FBs

FBs and MCs constitute an ambiguous subset of developmental heart cells owing to their pronounced transcriptional heterogeneity and lack of well-established molecular markers. Spatial factors, reflecting the positions of their progenitors and niche-related environmental signals, have a central role in defining cellular identities in this population[65]. To explore this topological diversity, we spatially mapped 18 fine-grained cardiac FB and MC clusters (Fig. 6a, Extended Data Fig. 6a,b, Supplementary Table 6 and Supplementary Discussion 1) to characteristic tissue locations, including the adventitia of great arteries and coronary arteries (Fig. 6b), the subepicardium (Fig. 6c) and the myocardium (Fig. 6d). Notably, we identified a cell state with a marker profile resembling fibro-adipogenic progenitors in the atrioventricular sulcus and the myocardium, with potential implications for epicardial fat formation and the pathogenesis of arrhythmogenic right ventricular dysplasia (Fig. 6e, Extended Data Fig. 6a,c–e and Supplementary Discussion 1).

Our clustering also identified three distinct MC states in the developing cardiac valves (Fig. 6e,f and Extended Data Fig. 6f,g), further delineating their cellular diversity. Valve interstitial cells (VICs) expressing APCDD1, LEF1, TMEM132C and ADAMTS19 localized to the free segments of both valve sets, while Valve_MC_1 cells (FGF14, HDAC9, PLCXD3) appeared enriched in the roots of the semilunar valve cusps and surrounding segments of the annulus fibrosus but were largely absent from the atrioventricular valves. Meanwhile, Valve_MC_2 cells (PENK, FMOD, TIMP3) were present in and around the roots of both semilunar and atrioventricular valves and within the internal regions of the annulus fibrosus, in contrast to AnnFibr_FBs, which were more prominent in the subepicardial region and probably form the outer rim of this structure. Importantly, Valve_MC_2 cells featured a specific and robust expression of PENK, a previously described marker of mesenchymal neural crest derivatives in mouse hearts[66] (Fig. 6f and Extended Data Fig. 6a,b). We observed strong spatial PENK signals in and around the developing semilunar and atrioventricular valves, supporting the long-debated contribution of neural crest cells to both valve structures in humans. Indeed, we observed PENK expression in the medially located leaflets of atrioventricular valves already at PCW 6.5, which appeared more spread out towards the lateral leaflets by PCW 9, along with a semilunar valve-related PENK signal present in both sampled aortic valve cusps (Fig. 6g). Taken together, our results imply a more substantial contribution of neural crest-derived mesenchyme to human cardiac valves than previously assumed. Based on these results, we assembled a spatial model of MC and FB cell states contributing to the fibrous skeleton and embedded valve structures in the fetal heart (Fig. 6h).

## Spatially informed analysis of developmental cardiac niches

Cellular microcompartments have key roles in defining environmental cues that guide cell behaviors in early cardiogenesis. We used our in-depth spatial analysis to outline the cellular composition of major developmental tissue compartments and refined niches, enabling their targeted analysis. We investigated the spatial overlap of the 72 fine-grained single-cell states by calculating pairwise correlation-based co-detection scores and visualizing their topological relations (Fig. 7a and Supplementary Table 7), which yielded broadly consistent results across three developmental age groups (Extended Data Fig. 7a).

Within the CPCS, we found substantial overlap between predicted positions of CALN1[high]_FBs, SAN_CMs and AVN_CMs, supporting the assumption that CALN1[high]_FBs represent a mesenchymal component of the developing nodal tissue (Supplementary Discussion 1). Of note, AVN_CMs showed the highest spatial co-detection scores with the AtrSept_EC state, implying the presence of conductive CMs in the atrial septum of the fetal heart (Fig. 7b). We also identified several potential ligand–receptor pairs between SAN_CMs and their cellular surroundings, including ones directed from the pacemaker cells towards closely localized venous ECs (Ven_EC), and others probably involved in contact formation with autonomic neurons (Aut_Neu_2) (Fig. 7c). Notably, CALN1[high]_FBs displayed a significant overlap in their interaction profile with SAN_CMs to a glial cell state recently discovered in the adult heart[42], which suggests a potential hybrid function for this population under development. The ventricular CPCS components were most closely associated with one another and with the ventricle-enriched endocardial Endoc_EC_1 and Endoc_EC_2 cell states, displaying a temporal shift in PF_CM co-detection from Endoc_EC_1 to Endoc_EC_2, as

**Fig. 6 | Spatial decomposition of FB and MC heterogeneity in the developing heart. a**, UMAPs displaying eight coarse-grained and 18 fine-grained FB and MC clusters, representing WT1-enriched and CD34-enriched, adventitial, valve-related and proliferating subsets, along with additional CALN1-enriched (CALN1[high]_FB) and refined pericyte-like (Peric_MC[fg]) and PDE4C-enriched populations (PDE4C[high]_FB[fg]). **b**, Spatial mapping of Adv_FB_1 (red) and Adv_FB_2 (green) cells to the great arteries (ROI A) and coronary arteries (ROI B), respectively (upper), and Peric_MC[fg] cells (red) and pericytes (PC) (green) to complementary positions in the atrial and ventricular myocardium (lower) in a 10 PCW heart section. **c**, Spatial mapping of epicardial cells (EpC) (red), EPDC_1 (green) and EPDC_2 (blue) cells to the heart surface, subendocardium and atrioventricular groove (dashed line) in a 10 PCW heart section. **d**, Spatial mapping of Int_FB_1 (red), Int_FB_2 (green) and Int_FB_3 (blue) cells, illustrating their gradual spatial expansion and layer-specific positions in 6, 8, 10 and 12 PCW heart sections. **e**, Spatial mapping of MC and FB states enriched in the atrioventricular plane (dashed line) and cardiac valves. VICs (red), Valve_MC_2 (green) and Valve_MC_1 (blue) cells (upper), and EPDC_1 cells (red), AnnFibr_FBs (green) and FAPs (blue) (lower) are presented in a 12 PCW heart section. **f**, Dot plot displaying the top ten DEGs (log$_2$FC > 0, P < 0.05) between the VIC, Valve_MC_1 and Valve_MC_2 clusters. **g**, ISS detection of PENK and LEF1 in cardiac valves in 6.5 PCW and 9 PCW heart sections. **h**, Spatial model of FB and MC state arrangements in the fetal heart's fibrous skeleton and cardiac valves. EPDC_1, FAP and AnnFibr_FBs cells are enriched in the subepicardium of the atrioventricular groove, while Valve_MC_2 cells dominate the inner annulus fibrosus. Valve roots, linking to this rim, harbor Valve_MC_2 in atrioventricular and Valve_MC_1–2 cells in semilunar valves. ca, coronary artery; avg, atrioventricular groove; af, annulus fibrosus; mv, mitral valve; tv, tricuspid valve.

well as differences in ligand–receptor interactions linked to trabecular morphogenesis and PF specification (Extended Data Fig. 7b).

Our analysis also demonstrated compositional differences between the great arteries (Art_EC_1, OFT_SMC, Adv_FB_1-2) and

coronary arteries (Art_EC_2, CA_SMC, Adv_FB_2), consistent with differences in their developmental origins and functions, and a close spatial association between lymphatic endothelial (LEC^fg) and myeloid cells (MyC), essential for de novo lymphangiogenesis (Extended Data

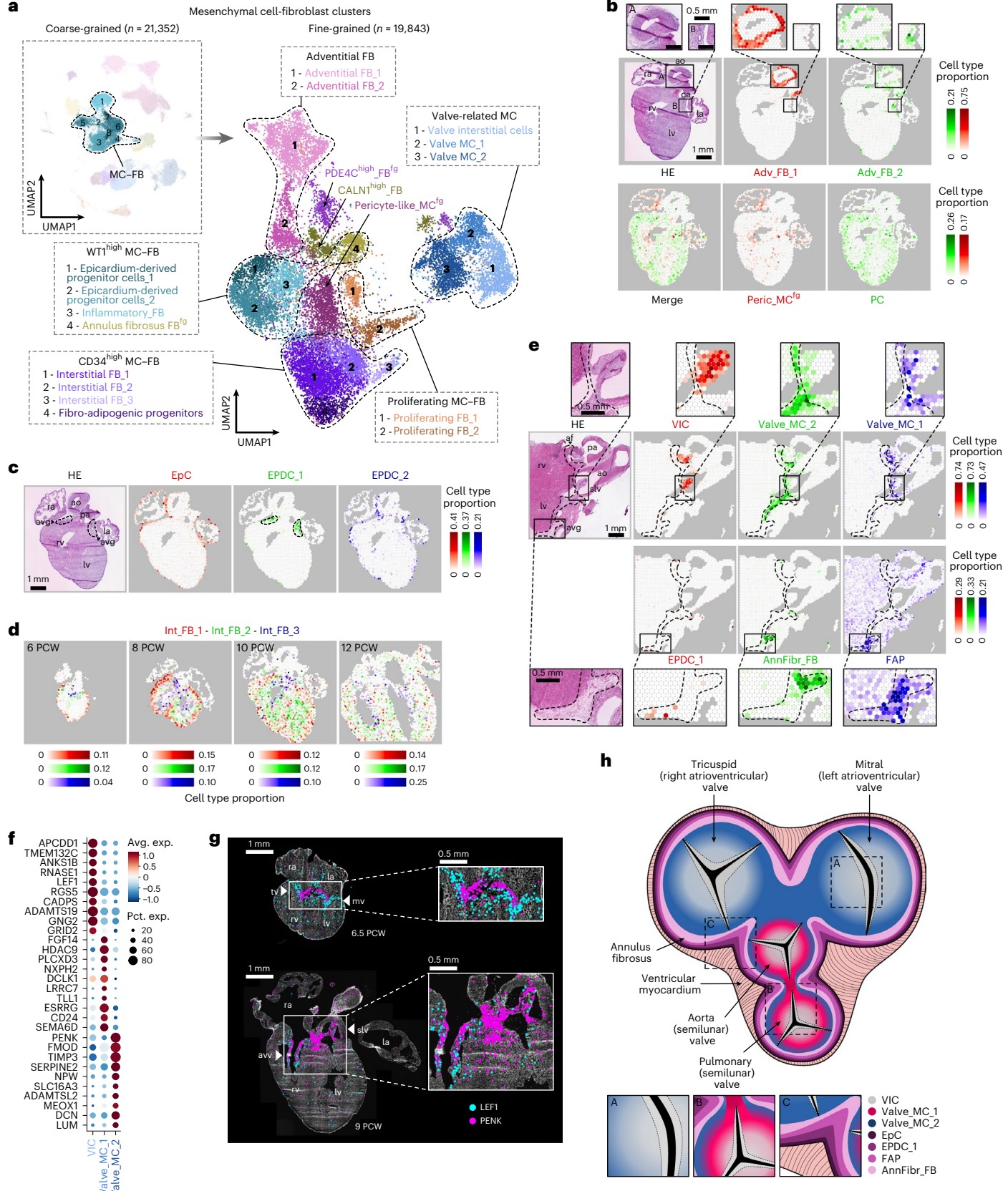

Fig. 7c)[67]. In parallel, we observed a temporal decrease of co-detection scores between EC states in consecutive segments of the coronary vasculature (Art_EC_1-2, Arteriol_EC, Cap_EC_1-2), reflecting their gradual spatial separation (Extended Data Fig. 7d).

By identifying five spatially distinct valvular EC and MC states, along with the vCM_1 population enriched at the valve roots and atrioventricular region of the ventricular myocardium, our study maps the developing human cardiac valve architecture with unprecedented spatial and cellular resolution (Fig. 7d and Extended Data Fig. 7e)[56,68]. Hence, we further charted the distribution of WNT, BMP and TGFβ signaling components, which are key regulators of valvulogenesis and contributors to congenital and acquired valve disease[69], in these populations (Fig. 7e). IF_VECs displayed pronounced enrichment of WNT ligands (*WNT2, WNT2B, WNT4, WNT9B, WNT11*), antagonists and signaling inhibitors (*ZNRF3, DKK2, DKK3*), while OF_VECs expressed higher levels of WNT receptors (*FZD10, ROR1, ROR2*) and signal enhancers (*LGR5*), in addition to several BMP ligands (*BMP4, BMP6*) and antagonists (*SOST, FST, FSTL4*). These patterns align with the role of fluid forces, previously described to be mediated by KLF2 (and KLF4), in modulating WNT activation and BMP suppression[63,70]. Although side-specific distribution of WNT and BMP signaling has been reported in animal models, our study is the first to consistently map their compartmentalization in developing human valves.

The TGFβ axis exhibited subtler cell state differences, with IF_VECs expressing higher levels of ligands (*TGFB1, TGFB2*), and OF_VECs expressing higher levels of signaling modulators (*LTBP1, BAMBI, THBS1*). The valvular MC populations shared a more homogenous expression profile of these genes, including *WNT5B* and *RORA*, mediators of non-canonical WNT signaling linked to calcification and fibrosis in diseased adult aortic valves[71]. Notably, Valve_MC_1 cells showed marked enrichment of *RSPO2*, implicated in bicuspid aortic valve disease[72], and Valve_MC_2 cells of *FMOD*, a regulator of the formation of a fibrous anchor between the valves and adjacent tissue[73], consistent with their predicted positions.

Cardiac valves are frequently affected in both congenital and acquired heart diseases, yet the specific cellular drivers of these pathologies often remain unidentified. Here, we examined the expression patterns of genes linked to syndromic and non-syndromic forms of two major valve anomalies—bicuspid aortic valve and mitral valve prolapse[74–76]—across distinct valvular cell states (Fig. 7f). Although mitral valve prolapse-associated genes were broadly expressed, several genes implicated in non-syndromic bicuspid aortic valve disease showed marked enrichment in IF_VECs and VICs, highlighting these cell states as particularly vulnerable. These findings illustrate how our dataset can support future genetic studies by revealing spatial enrichment of candidate disease genes in relevant developmental contexts.

## Discussion

Most congenital and several acquired heart diseases originate during early development, highlighting the importance of this period in defining a healthy cardiac architecture. Exploring concurrent spatiotemporal cellular and molecular patterns can provide cues for the pathomechanisms, and thus potential therapeutic targets, in these conditions. In recent years, several human cardiac single-cell atlases have been generated, characterizing the cellular composition of developing, healthy and diseased adult hearts[23,77,78]. Although representing invaluable resources for understanding the cellular complexity of cardiogenesis, these developmental datasets remain challenging to interpret, often because of limited sample sizes or the preselection of specific cell populations. Crucially, most of these studies lack spatial dimension, omitting essential information for contextualizing the collected data in terms of cardiogenic events. The combination of single-cell analysis with unbiased, transcriptome-wide spatial profiling has recently opened new horizons for decoding cardiac architecture[42].

In this study, we integrated single-cell and spatial transcriptomic datasets to generate the most comprehensive spatiotemporal atlas of first-trimester human heart development to date. In addition to region-specific gene expression profiling, this approach also allows for the precise delineation of cellular heterogeneity, even within smaller structural and functional compartments of the forming heart.

Accordingly, we performed an in-depth molecular characterization of CMs within the developing CPCS, revealing their unique electrophysiological properties and close spatial and functional association with specialized FBs in the nodes, endocardial cells in the ventricular conduction system and neurons of the autonomic innervation. Importantly, we identified a population of resident neuroendocrine chromaffin cells in the fetal human heart, suggesting a role in coordinating the organ-level response to hypoxia—such as during birth—and indicating a potential cellular origin for rare cardiac pheochromocytomas.

We resolved key cellular components of the developing cardiac valves, including putative endocardium-derived and neural crest-derived MCs, and distinct EC states lining the opposite sides of the valves, detectable from as early as PCW 6. We also explored the expression patterns of key signaling networks involved in valvulogenesis, reflecting flow-related transcriptomic signatures and highlighting ventricular-side ECs and VICs as susceptible targets in non-syndromic bicuspid aortic valve disease. In the atrial septum, we identified a distinct endocardial cushion-related population marked by strong *LEF1* and *MSX1* expression, supporting a potential causal role for these transcription factors in atrial septal defects. This cell state also displayed increasing expression of genes regulating the endothelial–mesenchymal transition over time, which is intriguing given the limited extension of the mesenchymal cap on the developing atrial septum and the absence of defined, persisting mesenchymal components in this structure, in contrast to the cardiac valves.

Given the lack of characteristic marker genes, we leveraged predictions on spatial cell state distributions to untangle the diversity of non-mural cardiac FB and MCs in the developing heart, revealing diverse cell states associated with the annulus fibrosus, vessel adventitia, subepicardium and myocardial interstitium. Among them, we identified a fetal cardiac FB-like population with fibro-adipogenic

---

**Fig. 7 | Spatially informed analysis of developmental cardiac niches.**
**a**, Schematic representation of the applied niche discovery strategy, using spatial mapping results of 72 fine-grained cell states for co-detection analysis (upper). Network graph of cardiac niches generated based on co-detection scores of >0.07, with gray lines representing spatial association and circle size reflecting the number of co-detected cell states (lower). Major cardiac structural components are delineated and labeled manually. **b**, Atrial septum and CPCS niche network graph and corresponding co-detection scores (left). Spatial mapping of SAN_CMs (red), AVN_CMs (green), CALN1^high_FBs (blue) (upper right), Ven_ECs (red) and AtrSept_ECs (green) (lower right), outlining the atrial septum (arrowheads) and nodal tissue (dashed lines) in an 11 PCW heart section. **c**, Spatially informed cell–cell communication analysis between SAN_CMs, its cellular neighborhood and Chrom_Cs, highlighting ligand–receptor interactions included in a recently published neural-GPCR module of CellPhoneDB[42], differentially expressed in the analyzed cell states within their respective subsets. **d**, Cardiac valve niche network graph and corresponding co-detection scores (upper). Spatial mapping of OF_VECs (red), IF_VECs (green), VICs (blue) (lower, ROI A) and Valve_MC_2 (red), Valve_MC_1 (green) and vCM_1 (blue) cells (lower, ROI B), highlighting dominant cellular components of semilunar valves (ROI A) and neighboring atrioventricular plane regions (ROI B), respectively, in a 12 PCW section. **e**, Heatmap representing per-gene scaled expression of WNT, BMP and TGFβ signaling components in valvular EC and MC states. **f**, Heatmap displaying per-gene scaled expression of bicuspid aortic valve disease-related and mitral valve prolapse-related genes in valvular EC and MC states. avp, atrioventricular plane.

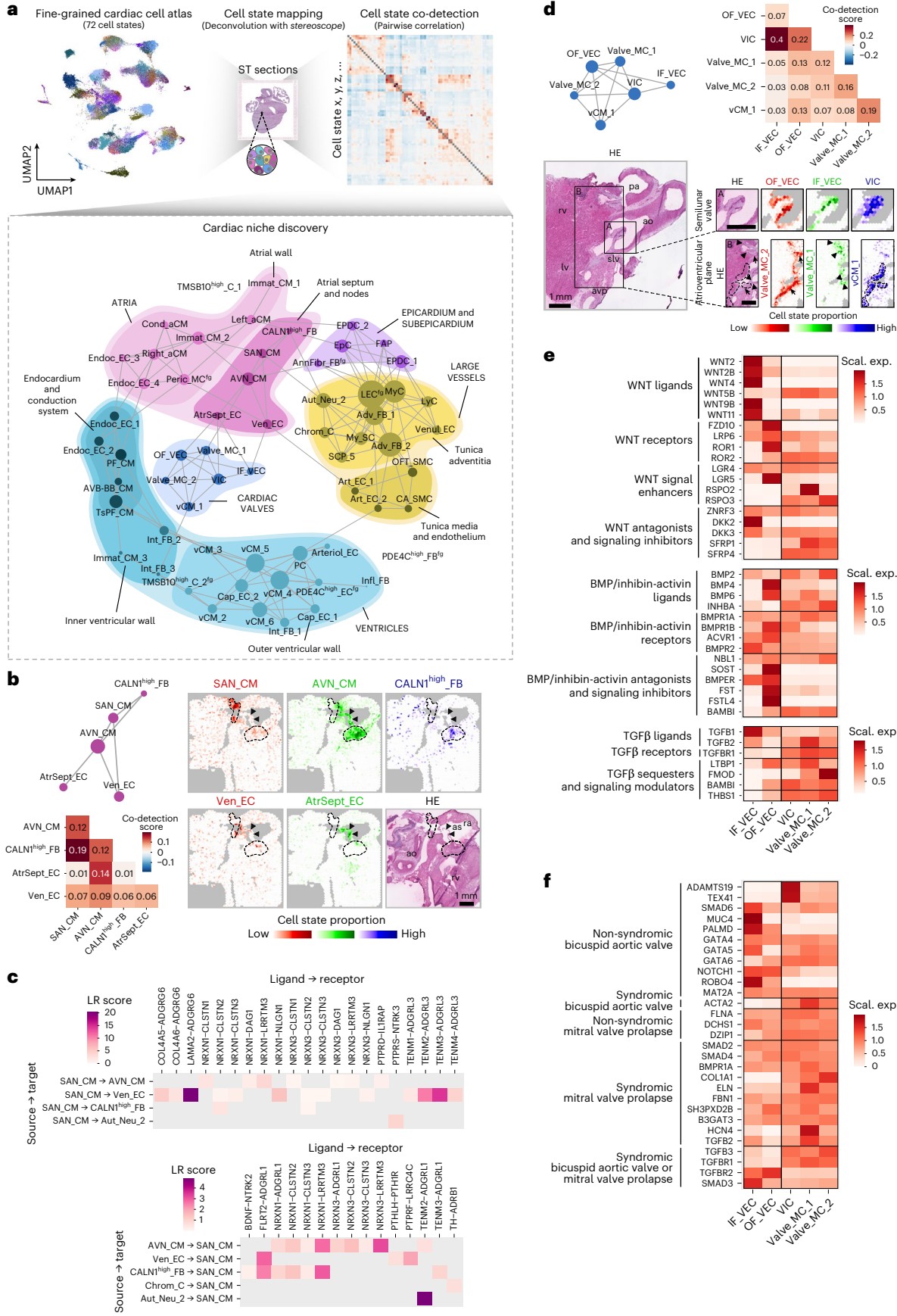

progenitor features and proposed it as a potential developmental precursor to cells involved in fibro-fatty remodeling in arrhythmogenic right ventricular cardiomyopathy. Finally, we delineated the cellular architecture of fine-grained developmental cardiac niches, providing a foundation for more nuanced and spatially informed interrogation of cell–cell communication and lineage trajectories before bioinformatic tools directly grounded in spatial transcriptomics data analysis become more established.

Our findings align with and significantly extend those of a recent study that used targeted MERFISH technology to investigate cellular communities in the fetal human heart[33] (Supplementary Discussion 2). Ongoing convergence of transcriptome coverage and spatial resolution between sequencing-based and imaging-based spatial transcriptomics, refinement of cell segmentation approaches and increasing capture rates across RNA profiling methods promise significant enhancements in data quality and analysis outcomes for future studies applying these technologies.

Our study is intrinsically constrained by the developmental time frame investigated, which does not cover the first 2 weeks of cardiogenesis, during which many congenital heart disease-related genes are already active. An increased sample size would improve the resolution and robustness of our analysis and enable further validation or refinement of the identified cell populations. Achieving balanced spatial sampling of cardiac regions for spatial transcriptomics, particularly for morphologically indistinct structures such as the nodal tissue, remains challenging. Finally, integrating multi-omics data would further enrich the proposed transcriptome-based cellular and architectural framework of cardiogenesis.

Our datasets may furnish novel insight into other aspects of early cardiogenesis not covered by our current study and serve as a spatiotemporal reference for early expression patterns of candidate genes in congenital heart diseases and for benchmarking human pluripotent stem cell-derived cardiac cell and tissue models, often resembling embryonic phenotypes. In conclusion, our work underscores the vast yet underexploited potential of spatially resolved molecular analysis in advancing cardiac research.

## Online content

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

[1]Department of Gene Technology, KTH Royal Institute of Technology, Science for Life Laboratory, Solna, Sweden. [2]Department of Cell and Molecular Biology, Karolinska Institute, Solna, Sweden. [3]Department of Biochemistry and Biophysics, Stockholm University, Science for Life Laboratory, Solna, Sweden. [4]Department of Information Technology, Uppsala University, Science for Life Laboratory, Uppsala, Sweden. [5]Division of Cellular and Clinical Proteomics, Department of Protein Science, KTH Royal Institute of Technology, Science for Life Laboratory, Solna, Sweden. [6]Department of Bioengineering, Stanford University, Stanford, CA, USA. [7]Unit of Integrative Metabolomics, Institute of Environmental Medicine, Karolinska Institute, Stockholm, Sweden. [8]Division of Molecular Neurobiology, Department of Medical Biochemistry and Biophysics, Karolinska Institute, Solna, Sweden. [9]Division of Neurogeriatrics, Department of Neurobiology, Care Sciences and Society, Karolinska Institute, Solna, Sweden. [10]Department of Pharmacology and Toxicology, University Medical Center Göttingen, Göttingen, Germany. [11]DZHK (German Centre for Cardiovascular Research), Lower Saxony Partner Site, Göttingen, Germany. [12]Department of Medicine, Karolinska Institute, Huddinge, Sweden. [13]Department of Pathology, Stanford University, Stanford, CA, USA. [14]Chan Zuckerberg Biohub, San Francisco, CA, USA. [15]Department of Physiology and Pharmacology, Karolinska Institute, Solna, Sweden. [16]Department of Neuroimmunology, Medical University of Vienna, Vienna, Austria. [17]These authors contributed equally: Enikő Lázár, Raphaël Mauron, Žaneta Andrusivová. ✉e-mail: eniko.lazar@scilifelab.se; joakim.lundeberg@scilifelab.se

## Methods

Details on sample collection, the experimental workflow up to data generation and ethical considerations are provided below, while the bioinformatic analysis of the scRNA-seq and spatial transcriptomics datasets is described in the Supplementary Methods, supported by Supplementary Tables 9–19 and Supplementary Figs. 12–19.

### Ethics

The study was performed with approval of the Swedish Ethical Review Authority and the National Board of Health and Welfare under ethical permit number 2018/769-31, in accordance with Swedish regulations governing the use of prenatal tissue for medical research and treatment. All heart specimens included in this study were collected from elective medical abortions at the Department of Obstetrics and Gynecology at Danderyd Hospital and Karolinska Huddinge Hospital in Stockholm, Sweden. Only individuals over 18 years old with full decision-making capacity and without diagnosed psychiatric conditions affecting consent were eligible to donate embryonic tissues. The patients donated tissue with written informed consent after receiving both oral and written information about the purpose of the research project and the possibility of retracting their consent at any time, including later destruction of the donated tissue. The study participants did not receive any kind of compensation for their donation. All procedures adhered to the ethical principles outlined in the World Medical Association Medical Ethics Manual and the Declaration of Helsinki, and all experiments were carried out in compliance with applicable guidelines and regulations.

### Collection of human developmental heart tissue samples

Following collection, the specimens were transferred from the clinic to the dissection laboratory. Heart tissue samples were then swiftly dissected under sterile conditions within 2 h post abortion. In total, 36 human developmental hearts between the ages of PCW 5.5 and 14 were included in this study. The embryonal or fetal age was determined using clinical data (ultrasound and last menstrual period), anatomical landmarks and actual crown–rump length. Both sexes were present in the full dataset; specifically, 17 hearts were from female and 14 from male donors. Information on sex for the five hearts used for ISS and immunofluorescence experiments was not obtained.

### Sample preparation for spatial methods

In total, 21 collected heart samples were embedded in Tissue-Tek O.C.T., snap-frozen and stored at −80 °C. Samples were cryosectioned at 10 μm thickness. Sections from 16 hearts were placed on Visium Gene Expression glass slides (10x Genomics), while sections from five other hearts were placed on Superfrost Plus glass slides (Thermo-Fisher, 22-037-246) for ISS and immunohistochemistry.

### Visium spatial gene expression assay

Spatial gene expression libraries were generated using the Visium Gene Expression kit (10× Genomics) according to the manufacturer's protocol[79]. A minimum of two close-to-consecutive tissue sections were included in the analysis as technical replicates for all but one heart sample, for which replicates had to be excluded from downstream analysis because of experimental deficiencies. Tissue images were taken at ×20 magnification using the Metafer Slide Scanning platform (microscope, AxioImager.Z2 with ScopeLED Illumination, Zeiss; camera, CoolCube 4 m, MetaSystems; objective, Plan-Apochromat ×20/0.80 M27, Zeiss; software, Metafer5). The permeabilization time was adjusted to 20 min. Raw images were stitched with VSlide software (MetaSystems). In total, 38 Visium libraries were prepared from 16 hearts. Libraries were sequenced by using the Illumina NextSeq 2000 platform, in which the length of read 1 was 28 bp and the length of read 2 was 120 bp.

### ISS

To generate the ISS datasets, a panel of 150 gene targets was created, including known regulators of cardiogenesis, consensus markers of major cardiac cell types and highly enriched genes of clusters of interest (Supplementary Table 8). For these genes, a total of five padlock probes per gene were designed, following a pipeline described previously[80]. Next, a direct RNA ISS protocol[80] was applied to nine tissue sections, corresponding to PCW 6.5 (section_1–3), PCW 8.5 (section_4), PCW 9 (section_5–6) and PCW 11.5 (section_7–9).

For ISS sequencing library preparation, the sections were thawed at room temperature (20 °C to 25 °C) for 5 min, washed with PBS and fixed in 3% paraformaldehyde for 5 min. The sections were then washed three times in PBS, permeabilized with a 0.1 M HCl solution for 5 min and washed three times in PBS. Next, the sections were sequentially dehydrated in a 70% and 100% ethanol bath for 2 min each. Secure-seal chambers (SA50, 1–13 mm inner diameter × 0.8 mm depth, 22 mm × 25 mm outer diameter, 1.5 mm diameter ports) were laid over each sample, which were then rehydrated with 0.5% Tween-PBS, followed by a PBS wash. A padlock probe solution (2× saline sodium citrate (SSC), 10% formamide, 10 nM of each padlock probe) was added to the chambers for overnight incubation in a humidified chamber at 37 °C. The sections were then washed twice with a washing solution (2× SSC, 10% formamide) and then twice with 2× SSC. The sections were then incubated in a ligation mix (1× T4 Rnl2 reaction buffer (NEB, B0239SVIAL), 0.05 μM RCA primer, 1 U μl$^{-1}$ RiboProtect (BLIRT, RT35), 1.0 U μl$^{-1}$ T4 Rnl2 (NEB, M0239)) in a humidified chamber at 37 °C for 2 h. The sections were subsequently washed twice with PBS and then incubated in a rolling circle amplification mix (1× phi29 buffer (50 mM Tris-HCl pH 8.3, 10 mM MgCl$_2$, 10 mM (NH$_4$)$_2$SO$_4$, 5% glycerol, 0.25 mM dNTPs (BLIRT, RP65), 0.2 μg ml$^{-1}$ BSA and 1 U μl$^{-1}$ Φ29 polymerase (Monserate Biotech, 4002)) at 30 °C in a humidified chamber overnight.

After washing the sections three times with PBS and two times with 2× SSC, fluorescent probes were hybridized to the samples with incubation in a bridge probe mix (20% formamide, 2× SSC, 0.1 μM of each bridge probe) for 30 min in a humidified chamber at room temperature. The sections were washed twice with PBS and twice with 2× SSC, followed by hybridizing a detection oligonucleotide mix (20% formamide, 2× SSC, 0.5 μM of each detection oligonucleotide (conjugated with Atto425, Alexa Fluor 488, Cy3, Cy5, Alexa Fluor 750), 0.25 μM DAPI (Biotium, S36936)) for 30 min in a humidified chamber at room temperature. Sections were washed twice with PBS, then mounted with SlowFade Gold Antifade Mountant (Thermo Scientific, S36936).

Cyclical imaging of the sections[80] was performed by stripping bridge probes and detection oligonucleotides after each cycle with three washes in 100% formamide for 5 min each, followed by five washes in 2× SSC. A new set of bridge probes was then hybridized in the same manner, and the process was repeated until all five cycles were imaged. Image acquisition was performed using a Leica epifluorescence microscope (microscope, DMi6000, Lumencor SPECTRA X Light Engine, LMT200-HS automatic multi-slide stage; camera, sCMOS, 2048 × 2048 resolution, 16 bit; objective, HC Plan-Apochromat ×20/0.80 air, and Plan-Apochromat ×40/1.10 W CORR oil objectives; filters, filter cubes 38HE, Chroma 89402 ET 391-32/479-33/554-24/638-31 Multi LED set, Chroma 89403 ET 436-28/506-21/578-24/730-40 Multi LED set and an external filter wheel DFT51011). Each region of interest (ROI) was marked and saved in the Leica LASX software for repeated imaging. Each ROI was automatically subdivided into tiles, and a z-stack with an interval of 0.5 μm was acquired for each tile in all the channels, with a 10% overlap between each tile. The ROIs were saved as TIFF files with associated metadata.

To create the ISS maps[80], the images obtained for each dataset across cycles were orthogonally projected using maximum intensity projection with the aim of collapsing the different z-planes into a single image per tile. Next, tiles from individual cycles were stitched, and images from different cycles were aligned using ASHLAR, obtaining

five large, aligned TIFF files for each dataset, corresponding to the different imaging rounds. Owing to computational considerations, images were then re-tiled into smaller aligned images. A pre-trained CARE model was then applied to each of the re-tiled images, resulting in a reduction of the point spread function and an increase in the signal-to-noise ratio. The output was then converted to the SpaceTx format, and spots were identified using the PerRoundMaxChannel decoder included in starfish. Every spot was given a quality score for each of the cycles by normalizing for the most intense channel. Using the quality scores from all rounds, two quality metrics were composed for each spot: the mean quality score, defined as the mean of the qualities across cycles, and the minimum quality score, which corresponds to the score of the round in which the quality was the lowest. Using these metrics, spots were filtered, retaining only those presenting a minimum quality score over 0.4 for further analysis. Finally, using the identity and position of all the high-quality score spots decoded, expression maps were assembled for each of the datasets.

For ISS panels included in the figures, visualization of selected target genes over the DAPI image of the tissue section was performed using TissUUmaps[81], with high-resolution capture of the image viewport (zoom factor 8). Orientation of the images was adjusted according to the anatomical position (left–right, superior–inferior) of the sampled cardiac structural components.

## Indirect immunofluorescence and imaging

Immunostaining was conducted on three developing heart samples (PCWs 6.5, 9 and 11.5), using at least two consecutive sections as technical replicates for each sample. After fixation with 4% paraformaldehyde (Thermo Scientific, 043368.9M) for 15 min, the heart sections were washed three times in PBS for 15 min. Then, the samples were incubated overnight at 4 °C with the following antibodies: anti-ARL13B (RRID:AB_3073658, ab136648, Abcam; 1:400), anti-PDE4C (RRID:AB_3094595, HPA054218, Atlas Antibodies; 1:100) and anti-ATF3 (RRID:AB_1078233, HPA001562, Atlas Antibodies; 1:100) in 0.3 % Triton-PBS (Sigma-Aldrich, T8787). The sections were then washed three times in TBS-Tween for 15 min and then blocked for 30 min in 1× TBS (Medicago, 097500100) with 0.5% TSA blocking reagent (Perkin Elmer, FP1020) and Hoechst (Invitrogen, H3570, 2 µg ml$^{-1}$). Next, the samples were incubated for 90 min at room temperature with anti-mouse-IgG2a-Alexa647 (RRID:AB_2535810, A21241, Invitrogen; 1:800) and anti-rabbit-A488 (RRID:AB_2576217, A11034, Invitrogen; 1:800) secondary antibodies and rhodamine-conjugated Ulex Europeaus Agglutinin I (Vector Laboratories, RL-1062-2; 1:200), diluted in the blocking buffer. After incubation, the slides were washed three times with TBS-Tween for 15 min. Coverslips (VWR, 631-0147) were mounted on the slides using Prolong Gold Antifade Mountant (Thermo Fisher Scientific, P36930). Imaging was performed with a Carl Zeiss LSM 700 laser scanning microscope with Zen 2012 Black Edition software, using a Carl Zeiss Plan-Apochromat ×20/0.8 objective, with a 2,048 × 2,048-pixel resolution and 1.8 µm and 2 µm optical thickness. Z-stacks for ROIs A and B were collected with the same imaging parameters, from nine planes distributed across the entire depth of the tissue sections, and the captured images were further processed with the maximum intensity projection function. Linear adjustments of intensity and contrast were performed separately for each detection channel on the collected and processed images before they were exported in TIFF format for embedding in the final figure panels.

## Sample preparation for scRNA-seq

In total, 15 heart samples were analyzed by scRNA-seq. The specimens were cut into smaller pieces and minced using a blade. The tissue was transferred into a 15 ml Falcon tube containing an enzymatic solution of Collagenase II (200 U ml$^{-1}$; Worthington Biochemical, LS004174), DNase I (1 mg ml$^{-1}$; Worthington Biochemical, LK003170) in Earle's balanced salt solution (EBSS; Worthington Biochemical, LK003188) that

had been pre-oxygenated with 95% $O_2$:5% $CO_2$ on ice for 5–10 min. Tubes were put into a water bath, and the samples were digested at 37 °C, with an incubation time ranging from approximately 45 min to over 2.5 h, depending on the age of the tissue (tissues from later developmental stages required longer dissociation times). During the incubation, the cell suspension was triturated and dissociated approximately every 25 min, using glass fire-polished Pasteur pipettes, with gradually decreasing tip diameters for each round to enhance dissociation. The cell suspension was manually inspected under the microscope to assess the level of digestion, followed by a filtering step using a pre-wet 30 µm cell strainer (CellTrics, Sysmex) to remove debris, fibers and smaller undissociated remains. A small volume of EBSS was added to the filtered suspension to dilute and inactivate the enzyme, then cells were pelleted through centrifugation at 200g for 5 min. If needed, gradient centrifugation was performed to remove blood cells. For this step, the cell pellet was resuspended in a solution containing 900 µl EBSS, 100 µl albumin inhibitor solution (Ovomucoid protease inhibitor and BSA, OI-BSA, LK003182) reconstituted in EBSS and 50 µl of DNase I (larger volumes with the same reagent concentrations were used in case of higher cell numbers). The resuspended solution was carefully layered on top of 3 ml albumin inhibitor solution in a 15 ml Falcon tube, and gradient centrifugation was performed at 70g for 4–6 min. Then the supernatant was removed, and the cells were resuspended in a minimum amount of EBSS, starting from 100 µl (or more in case of higher cell numbers). The cell suspension was transferred to a 1.5 ml Eppendorf tube, pre-coated with 30% BSA (A9576, Sigma-Aldrich). If a few cellular clumps had formed, a small volume of BSA was added to the suspension (starting with 0.3 % BSA of the total volume). Cells were counted on a hemocytometer (Bürker/Neubauer chamber), and cell viability was assessed with Trypan blue (Gibco, 15250061). Cell concentrations were adjusted to optimal values for the 10× Genomics Chromium analysis.

## ScRNA-seq

Single cells were captured using the droplet-based platform Chromium (10× Genomics), and the Chromium Single Cell 3′ Reagent Kit v2 and v3. Owing to a change in the chemistry provision of the kits, five out of 21 libraries were sampled with v2. Cell concentrations were adjusted to between 800 and 1,200 cells per µl, where possible, for a target recovery of 5,000 cells per library. Cells were loaded on the Chromium controller, and downstream procedures such as cDNA synthesis, library preparation and sequencing were done according to the manufacturer's instructions (10× Genomics). Libraries were sequenced between approximately 100,000 and 250,000 reads per cell on the Illumina NovaSeq platform, reaching a saturation of 60–90%. All libraries were demultiplexed using Cell Ranger (cellranger mkfastq v.4.0.0; 10× Genomics) and filtered through the index-hopping-filter tool (v.1.1.0) by 10× Genomics. STARSolo (v.2.7.10a) was used to determine unique molecular identifier (UMI) counts, with the following parameters used:

- –soloFeatures Gene Velocyto
- –soloBarcodeReadLength 0
- –soloType CB_UMI_Simple
- –soloCellFilter EmptyDrops_CR %s 0.99 10 45000 90000 500 0.01 20000 0.01 10000
- –soloCBmatchWLtype 1MM_multi_Nbase_pseudocounts
- –soloUMIfiltering MultiGeneUMI_CR
- –soloUMIdedup 1MM_CR
- –clipAdapterType CellRanger4
- –outFilterScoreMin 30

Reads were aligned using the GRCh38.p13 gencode V35 primary sequence assembly. To minimize the loss of reads that occur when they map to more than one gene, we applied a read-filtering step with certain criteria, which is described in detail along with the retrieval of count matrices and read alignments, in a previous publication[82].

**Reporting summary**

Further information on research design is available in the Nature Portfolio Reporting Summary linked to this article.

## Data availability

All the data required to replicate the analysis, including Cell Ranger output, Space Ranger output, metadata, processed ISS data, supplementary figures and tables as well as main RDS objects are shared on the Mendeley DATA repositories part one (https://doi.org/10.17632/fhtb99mdzd.1) and two (https://doi.org/10.17632/w65jtfsvpr.1) under the following links: https://data.mendeley.com/datasets/fhtb99mdzd/1 and https://data.mendeley.com/datasets/w65jtfsvpr/1. Processed Visium, scRNA-seq and ISS data are available for browsing gene expression, clusterings and other analysis results at https://hdcaheart.serve.scilifelab.se/web/index.html, supported by simple instructions available in the viewer. The sensitive raw sequencing files for single-cell (https://ega-archive.org/studies/EGAS50000001029) and spatial transcriptomics (https://ega-archive.org/studies/EGAS50000001122) are deposited in the European Genome-Phenome Archive (EGA) and are available upon formal request. Owing to file size limitations, the corresponding authors can provide access to the raw ISS dataset upon reasonable request.

## Code availability

The code used in this study, together with the material and instructions required to reproduce the Docker and Conda environments, is available at https://github.com/rmauron/HDCA_heart_dev. A permanent version of the code is available on Zenodo (https://doi.org/10.5281/zenodo.15912655)[83].

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

## Acknowledgements

We acknowledge the services of the Developmental Tissue Bank of Karolinska Institute for providing prenatal tissue for this investigation, and the BioImage Informatics Facility unit of the National Bioinformatics Infrastructure Sweden (NBIS) for designing the TissUUmaps-based interactive viewer for our study. We thank M. Minaeva for her support with several bioinformatic tools used in this study, U. Axelsson for providing immunostaining reagents, Q. H. Nguyen, P. Pavithra and O. Mulay for scientific discussions, M. Karlén for his help with illustrations and K. Thrane, S. Sarenpää and Á. Kerényi for their useful suggestions on improving the text and figures. Finally, we thank the donors of tissue specimens, whose invaluable contributions made this study possible. This work was supported by grants from the Erling-Persson Foundation (HDCA to S.L., E.S., M.N., E.L. and J.L.), the Knut and Alice Wallenberg Foundation (2018.0220 to S.L., E.S., M.N., E.L. and J.L.) and SciLifeLab. I.A. was supported by ERC Synergy grant KILL-OR-DIFFERENTIATE 856529, Knut and Alice Wallenberg Foundation, Swedish Research Council, Bertil Hallsten Foundation, Paradifference Foundation, Cancerfonden, Hjärnfonden and Austrian Science Fund (project grants and SFB consortia). O.B. was supported by the Swedish Research Council, the Grönbergska Stiftelsen and the Leducq Foundation. J.N.H. was supported by an EMBO Postdoctoral Fellowship (ALTF 556-2022). The BioImage Informatics Facility was supported by SciLifeLab and the National Microscopy Infrastructure (VR-RFI 2019-00217).

## Author contributions

E.L., R.M., Ž.A., I.A. and J.L. conceptualized the study. E.L., Ž.A., E.B., S.M.S., N.S., S.S. and J.N.H. developed the methodology. R.M., J.F., L.L., C.A. and P.C. implemented the software. E.L., S.M.S., N.S., S.S. and J.N.H. validated the research. E.L., R.M., Ž.A., J.F., M.H., L.L., M.V. and P.C. conducted the formal analysis. E.L., Ž.A., E.B., S.M.S., N.S., S.S. and J.N.H. performed the investigation. X.L., E. Lu., S.L., M.N., E.S. and J.L. provided resources. E.L., R.M., Ž.A., J.F., L.L., S.M.S., N.S., M.V. and E.B. curated the data. E.L., R.M., Ž.A., J.F., S.M.S., N.S., S.S., J.N.H. and E.B. wrote the original draft of the manuscript; E.L., R.M., Ž.A. and J.L. contributed to manuscript review and editing. E.L., R.M. and Ž.A. visualized the project. E.L., O.B., C.S., E. Lu., S.L., M.N., E.S., I.A. and J.L. supervised the research. E.L., R.M., Ž.A. and J.L. provided project administration. E. Lu., S.L., M.N., E.S. and J.L. acquired funding.

## Funding

## Competing interests

S.L. is a paid scientific advisor to Moleculent, Combigene and the Oslo University Center of Excellence in Immunotherapy. S.M.S. is a co-founder of spatialist. M.N. is co-founder of VoxlBio. P.C. is a founder of Precisium AI. All other authors declare no competing interests.

## Additional information

**Extended data** is available for this paper at https://doi.org/10.1038/s41588-025-02352-6.

**Correspondence and requests for materials** should be addressed to Enikő Lázár or Joakim Lundeberg.

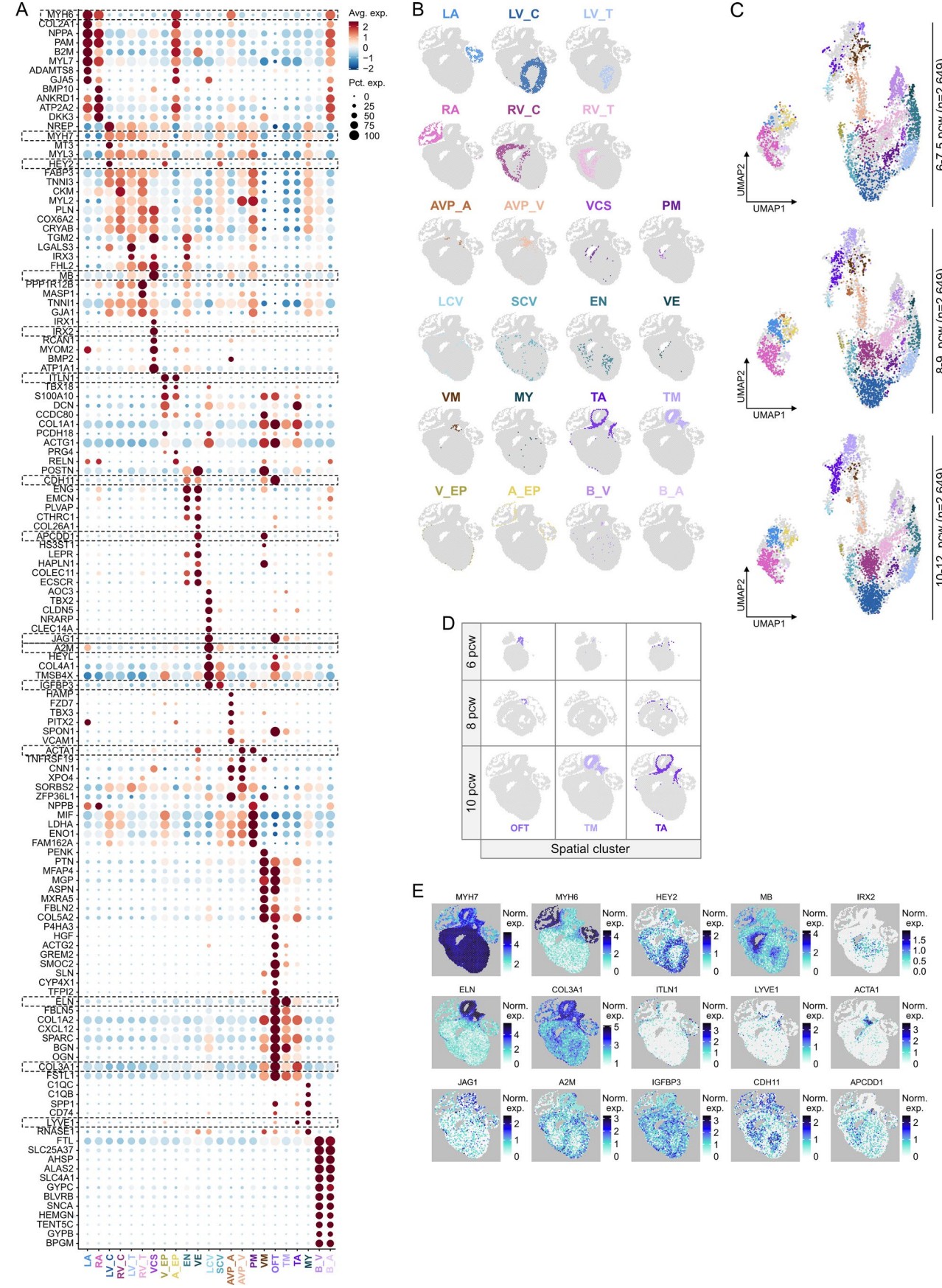

**Extended Data Fig. 1 | See next page for caption.**

**Extended Data Fig. 1 | Transcriptomic Profiles and Temporal Transitions of Cardiac Spatial Clusters. A**. Dot plot depicting the top 5 DEGs ($\log_2$FC > 0, $P$ < 0.05) in the 23 spatial clusters in the developing heart. Genes highlighted by dashed rectangles were selected for spatial expression plots included in Extended Data Fig. 1E. Avg. exp., average expression; Pct. exp., percent of expressing cells. **B**. Localization of spatial clusters in the developing heart, presented in a 10 PCW heart section. The cluster corresponding to the early outflow tract (OFT) is absent in this sample. **C**. UMAPs illustrating changes in distribution of tissue spots between spatial clusters across three developmental age groups (6-7.5, 8-9, 10-12 PCW, each age group downsampled to 2,649 spots). **D**. Temporal evolution of outflow tract- and great artery-related spatial clusters, presented in 6, 8 and 10 PCW heart sections. The cluster corresponding to the early outflow tract (OFT) gradually disappears, while the one representing the tunica media of great arteries (TM) expands during this period. **E**. Spatial feature plots displaying selected DEGs ($\log_2$FC > 0, $P$ < 0.05) of spatial clusters in a 10 PCW heart section.

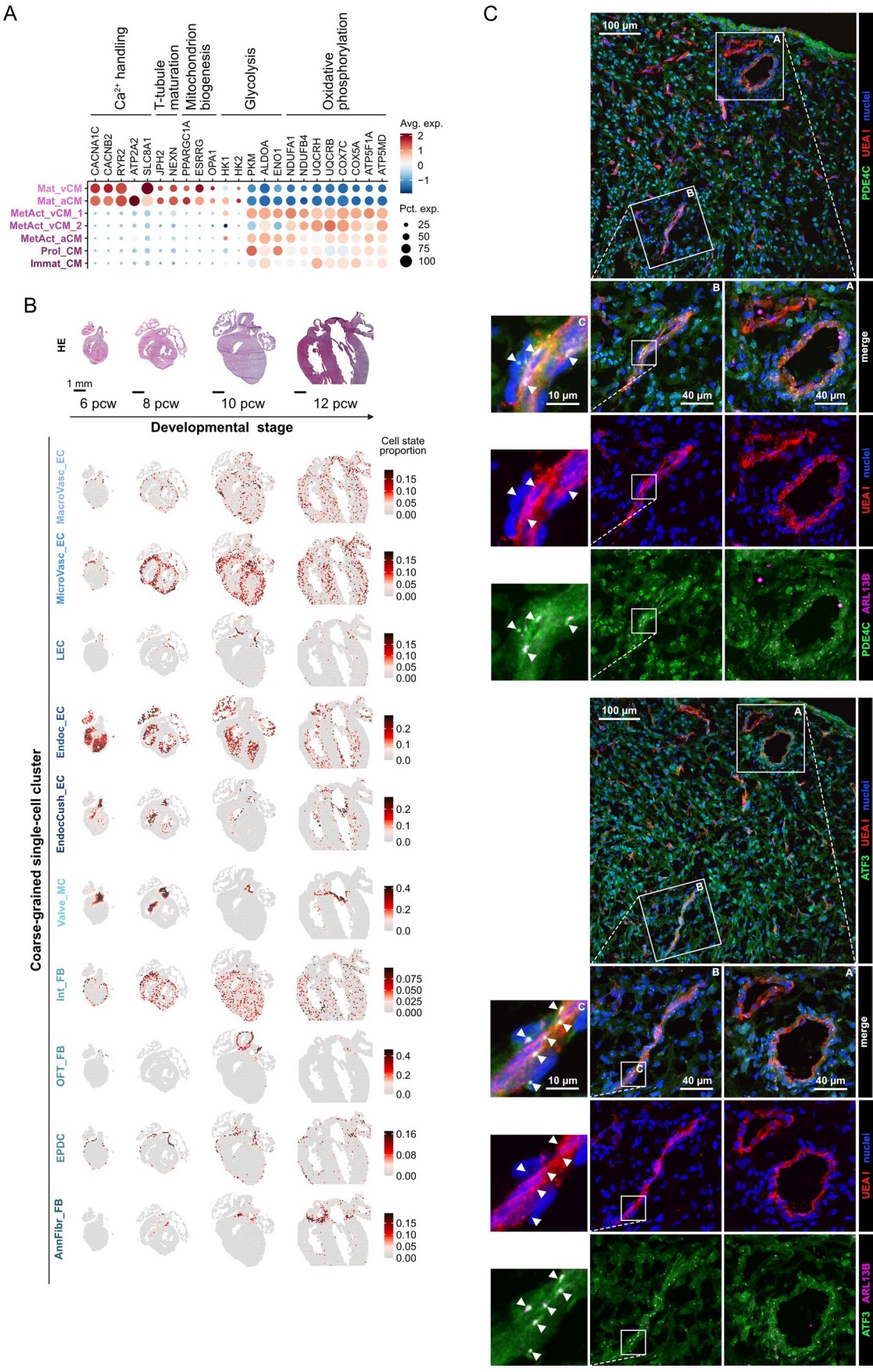

**Extended Data Fig. 2 | See next page for caption.**

**Extended Data Fig. 2 | Molecular Characteristics of Major Cardiac Cell Populations. A**. Dot plot showing relative expression of genes involved in electromechanical activation and bioenergetics of CMs, aligning with different maturation states of coarse-grained CM clusters. **B**. Spatiotemporal mapping of major EC and MC-FB coarse-grained clusters in 6, 8, 10, and 12 PCW heart sections. HE, hematoxylin & eosin; the scale bar represents 1 mm. **C**. Immunofluorescence demonstrating widespread expression of PDE4C (upper, green) and ATF3 (lower, green) proteins in the fetal heart, along with UEA I EC (red) and Hoechst nuclear staining (blue), presented in a 9 PCW heart section; the scale bar represents 100 μm. ROI A and B highlight coronary vessels on the heart surface and in a deeper layer of the ventricular myocardium, respectively; the scale bar represents 40 μm. ROI C presents a segment of the latter vessel structure with higher magnification, demonstrating subcellular enrichment of PDE4C (upper, green) and ATF3 (lower, green) proteins in close association to ARL13B-labeled primary cilia (magenta), consistent with the position of the basal body; the scale bar represents 10 μm. Arrowheads indicate PDE4C/ATF3+ basal body structures with connected ARL13B+ primary cilia. The images are representative of n = 3 independent experiments.

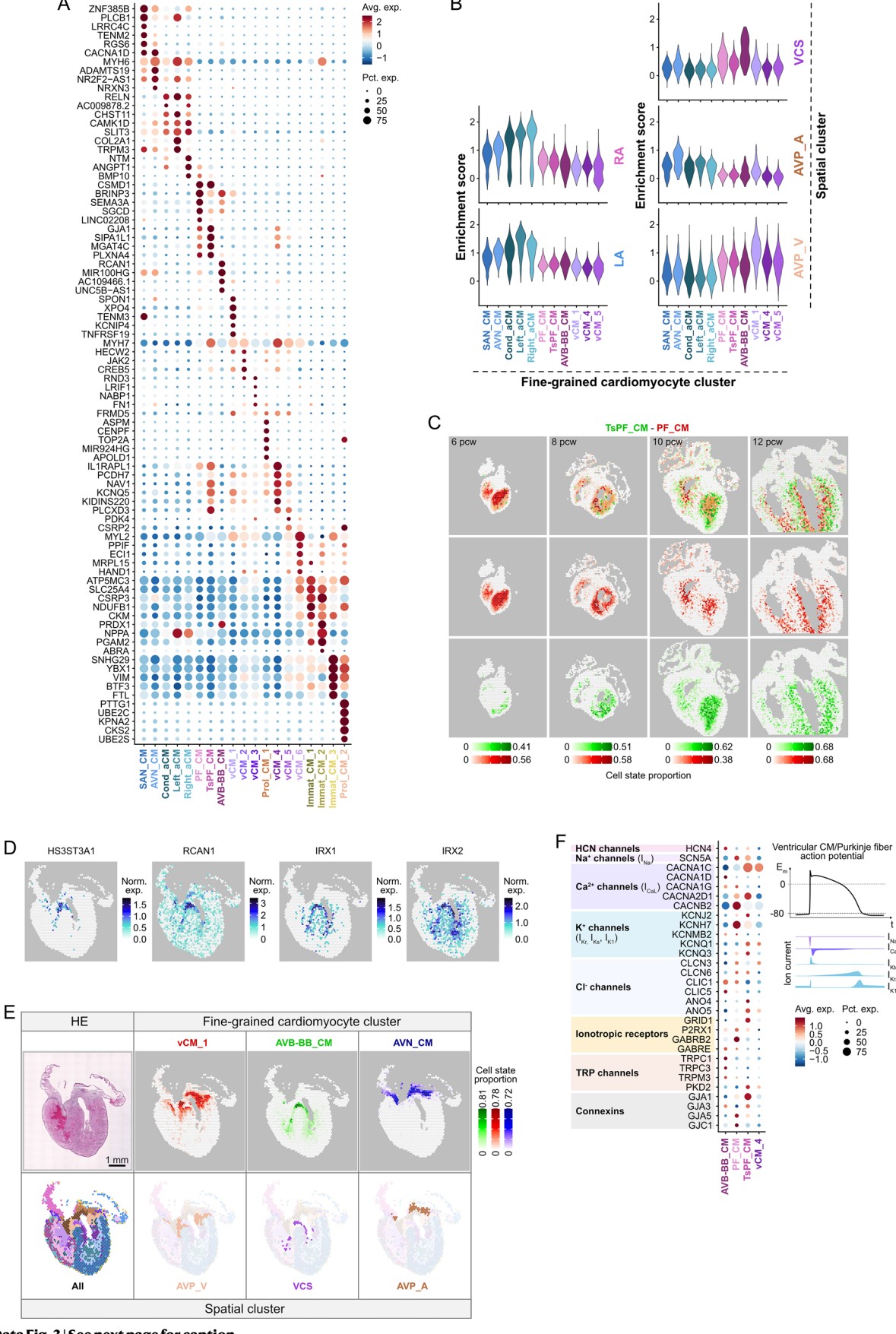

Extended Data Fig. 3 | See next page for caption.

**Extended Data Fig. 3 | Molecular and Spatial Features of Fine-Grained Cardiomyocyte States. A**. Dot plot depicting the top 5 DEGs ($\log_2$FC > 0, $P$ < 0.05) between the 19 fine-grained CM clusters. **B.** Violin plots showing gene signature enrichment of selected spatial clusters (LA, left atrial myocardium; RA, right atrial myocardium; VCS, ventricular conduction system; AVP_A, atrioventricular plane, atrial side; AVP_V, atrioventricular plane, ventricular side) in selected fine-grained CM states. **C.** Spatial mapping of PF_CMs (red) and TsPF_CMs (green) in 6, 8, 10, and 12 PCW heart sections, illustrating their temporally consistent arrangement along different depths of the ventricular wall. **D.** Spatial feature plots displaying selected DEGs ($\log_2$FC > 0, $P$ < 0.05) of AVB-BB_CMs in a 10 PCW heart section. **E.** Spatial arrangement of fine-grained CM and spatial clusters around the atrioventricular plane, presented in a 10 PCW heart

section. vCM_1 cells (red), AVB-BB_CMs (green) and AVN_CMs (blue) (upper) display highly similar localization to the AVP_V, VCS and AVP_A spatial clusters (lower), respectively. The scale bar represents 1 mm. **F.** Dot plot displaying the relative enrichment of selected ion channel genes in AVB-BB_CMs, PF_CMs and Ts-PF_CMs, compared to vCM_4 cells, characterized by the highest maturation state among the contractile ventricular CM clusters. Major ion currents of the ventricular CM/PF action potential are illustrated in colors consistent with the ion channel plot. $I_{Na}$–fast sodium current, $I_{CaL}$– L-type calcium current, $I_{Kto}$–transient outward potassium current, $I_{Kr}$–rapid delayed rectifier potassium current, $I_{Ks}$–slow delayed rectifier potassium current, $I_{K1}$–inward rectifier potassium current.

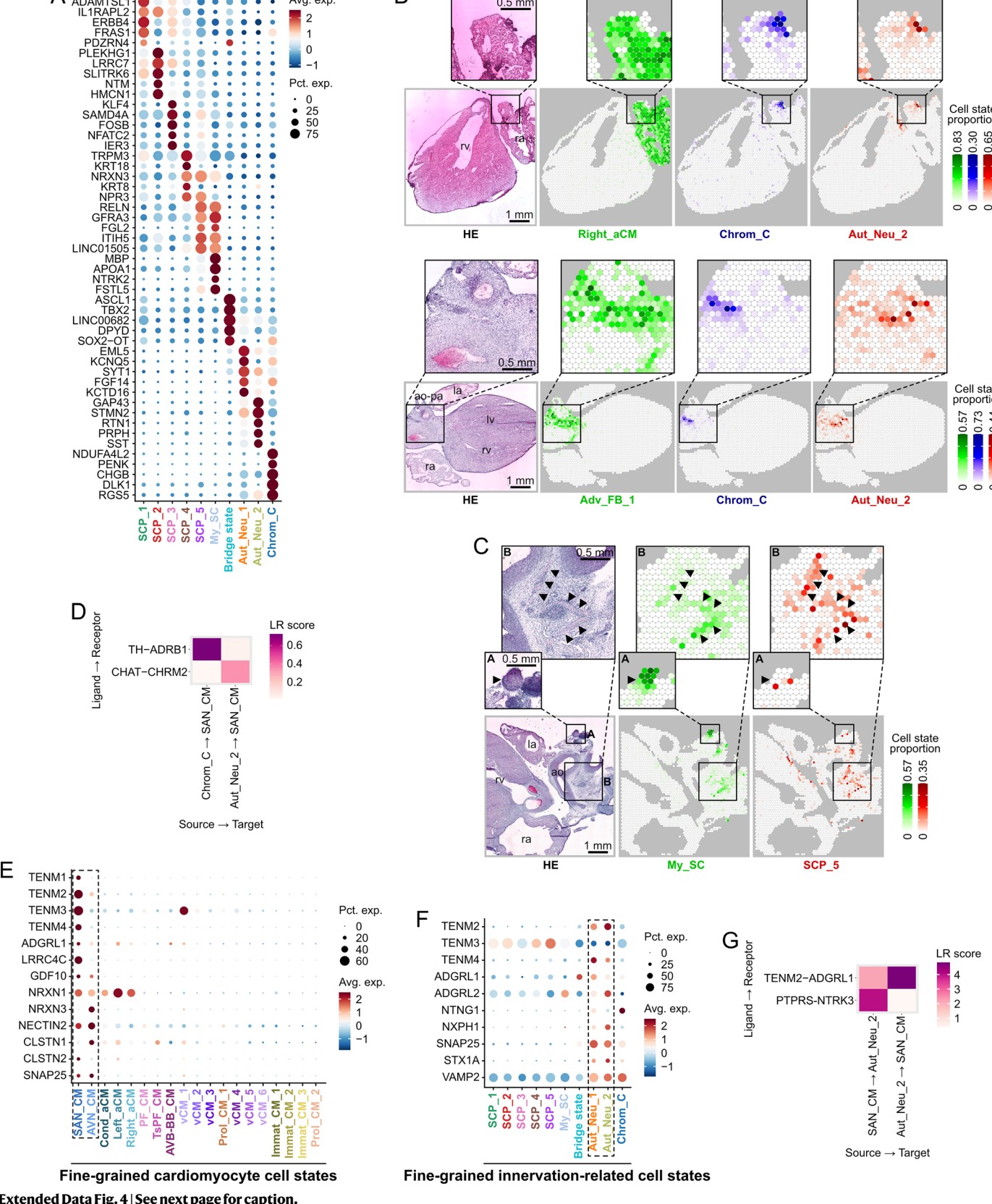

**Extended Data Fig. 4 | See next page for caption.**

**Extended Data Fig. 4 | Transcriptomic and Spatial Characteristics of Fine-Grained Innervation-Related Cell States. A**. Dot plot depicting the top 5 DEGs ($\log_2$FC > 0, $P$ < 0.05) between the 10 fine-grained innervation-related cell clusters. **B**. Spatial mapping of Chrom_Cs (blue) and Aut_Neu_2 cells (red) presented in a 10.5 PCW (upper) and a 11 PCW (lower) heart section, showing their close association with the atrial wall (outlined by Right_aCMs, green, upper) and the adventitia of the great arteries (outlined by Adv_FB_1 cells, green, lower). ROIs highlight areas enriched in innervation-related cell states. **C**. Spatial mapping of My_SCs (green) and SCP_5 cells (red) presented in a 11 PCW heart section, demonstrating their spatial association with nerves (arrowheads) in the adventitia of the great arteries (ROI A and B). **D**. Ligand-receptor scores, adapted from a recently published neural-GPCR module of CellPhoneDB[42], highlighting different sources of adrenergic and cholinergic mediation of SAN_CM function between the Chrom_C and Aut_Neu_2 states. **E**. Dot plot displaying selective enrichment of genes related to axon guidance and synapse formation in SAN_CMs and AVN_CMs (dashed box), compared to other fine-grained CM states. **F**. Dot plot displaying selective enrichment of genes related to axon guidance and synapse formation in Aut_Neu_1 and Aut_Neu_2 cells (dashed box), compared to other fine-grained innervation-related cell states. **G**. Neural/GPCR-mediated ligand-receptor interactions enriched between SAN_CMs and Aut_Neu_2 cells. la, left atrium; lv, left ventricle; ra, right atrium; rv, right ventricle; ao, aorta; pa, pulmonary artery; the scale bars represent 1 mm in the main and 0.5 mm in the zoom-in panels.

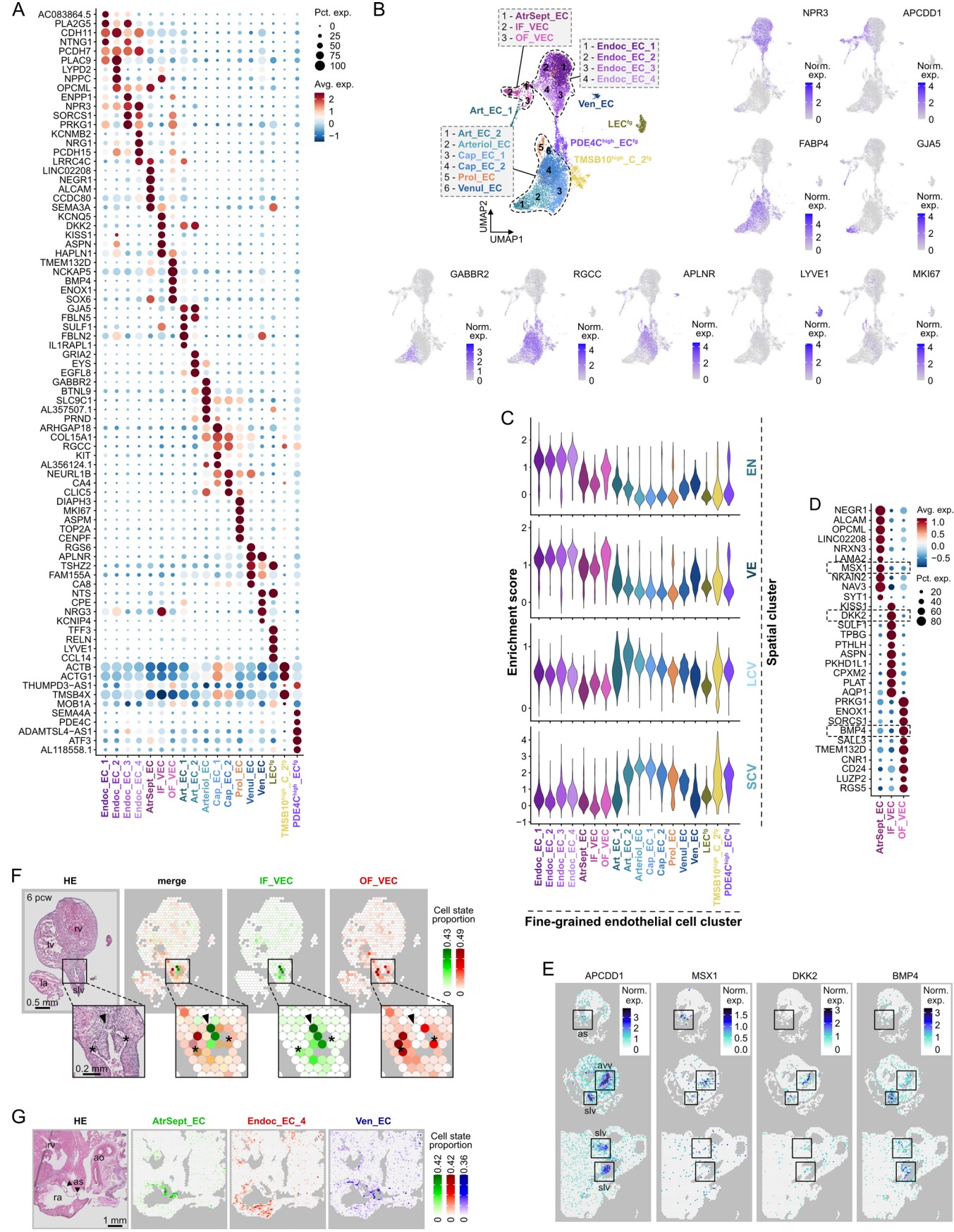

**Extended Data Fig. 5 | See next page for caption.**

**Extended Data Fig. 5 | Molecular and Spatial Profiles of Fine-Grained Endothelial Cell States. A**. Dot plot depicting the top 5 DEGs (log$_2$FC > 0, $P$ < 0.05) between the 18 fine-grained EC clusters. **B**. Feature plots of selected DEGs (log$_2$FC > 0, $P$ < 0.05) outlining EC populations of the endocardium (*NPR3*), endocardium-derived endothelial structures (*APCDD1*), great arteries and coronary vasculature (*FABP4, GJA5, GABBR2, RGCC, APLNR*), and lymphatic vessels (*LYVE1*), as well as proliferating cells (*MKI67*). **C**. Violin plots showing gene signature enrichment of selected spatial clusters (EN, endocardium and subendocardium; VE, –valve endothelium; LCV, large coronary vessels; SCV, small coronary vessels) in fine-grained EC states. **D**. Dot plot displaying the top 10 DEGs (log$_2$FC > 0, $P$ < 0.05) between the AtrSept_EC, IF_VEC and OF_VEC clusters. Genes highlighted by dashed rectangles were selected for spatial expression plots included in Extended Data Fig. 5E. **E**. Spatial feature plots visualizing the expression patterns of selected AtrSept_EC (*MSX1*), IF_VEC (*DKK2*) and OF_VEC

(*BMP4*) marker genes in the atrial septum, and atrioventricular and semilunar valves (marked by rectangles) in 8 (top, middle) and 12 PCW (bottom) heart sections. **F**. Spatial mapping of IF_VECs (green) and OF_VECs (red) in a 6 PCW heart section, demonstrating their selective enrichment on opposite sides of the developing semilunar valves already in the earliest analyzed developmental stages. The scale bars represent 0.5 mm in the main and 0.2 mm in the zoom-in panels; arrowhead, inflow side; asterisk, the outflow side of the valve. **G**. Spatial mapping of Endoc_EC_4 cells (red) and Ven_ECs (blue), visualized in a 11 PCW heart section, showing their respective enrichment in the smooth-walled atrium and neighboring venous structures. The atrial septum (arrowheads), enriched in AtrSept_ECs (green), is shown in close proximity to these regions. The scale bar represents 1 mm. slv, semilunar valve; avv, atrioventricular valve; as, atrial septum.

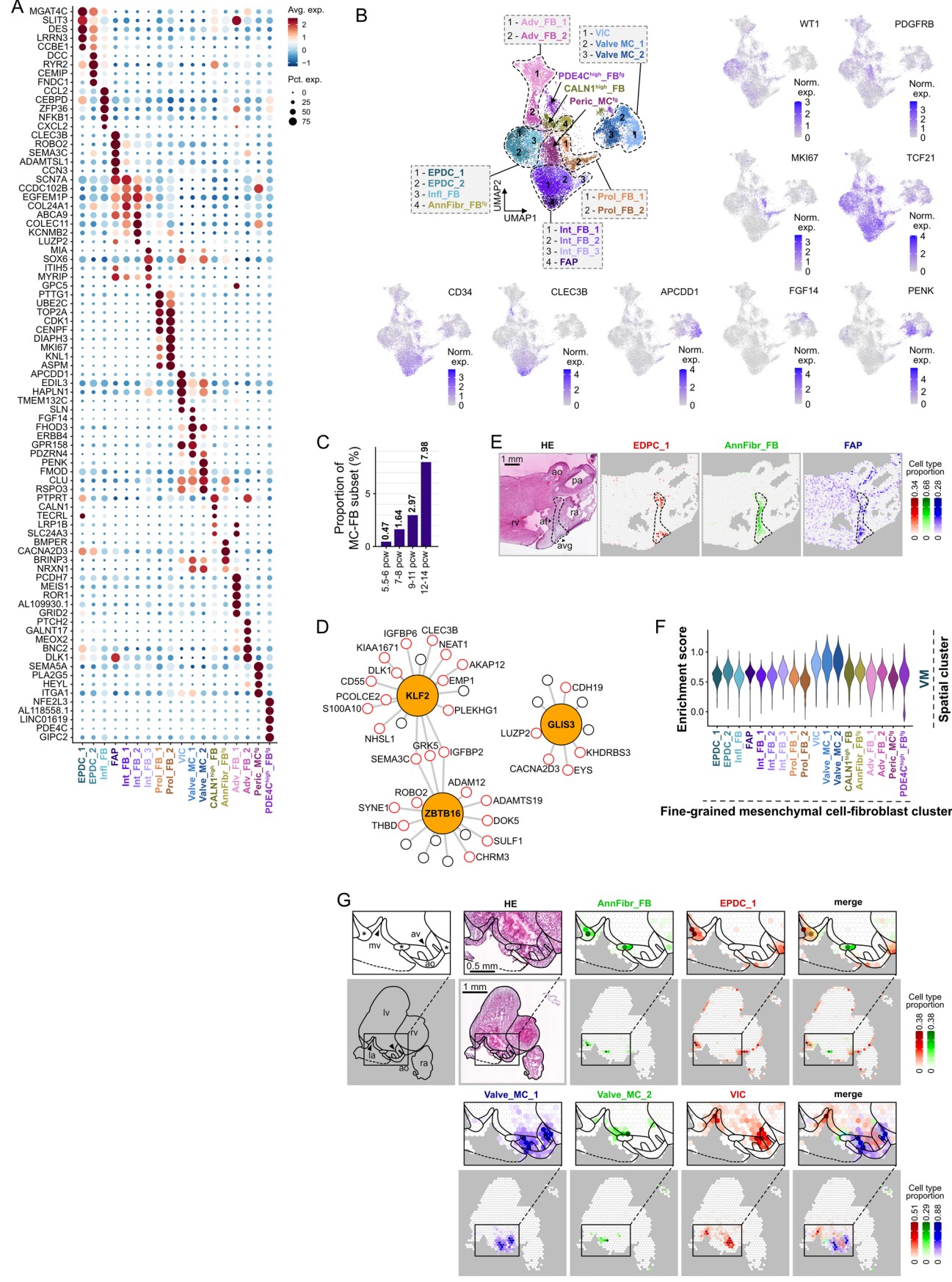

**Extended Data Fig. 6 | See next page for caption.**

**Extended Data Fig. 6 | Transcriptomic and Spatial Features of Fine-Grained Fibroblast-Mesenchymal Cell States. A**. Dot plot depicting the top 5 DEGs (log$_2$FC > 0, P_val < 0.05) between the 18 fine-grained FB-MC clusters. **B**. Feature plots of selected DEGs (log$_2$FC > 0, $P$ < 0.05) outlining cell populations with high expression of consensus epicardial (*WT1*), pericyte (*PDGFRB*), fibro-adipogenic progenitor (*CLEC3B*), putative cardiac fibroblast (*TCF21*, *CD34*), and proliferation (*MKI67*) markers, and genes enriched in distinct cardiac valve-related mesenchymal cell states (*APCDD1*, *FGF14*, *PENK*). **C**. Bar plot illustrating temporal changes in the proportion of FAPs in the MC-FB subset across four developmental age groups (5.5-6, 7-8, 9-11 and 12-14 PCW). **D**. Gene regulatory network of FAPs, including differentially expressed transcription factors (orange-filled circles) and their target genes. Red-rimmed circles represent DEGs (log$_2$FC > 0, $P$ < 0.05)

of FAPs, compared to other fine-grained MC-FB clusters. **E**. Spatial mapping of EPDC_1 cells (red), AnnFibr_FBs (green) and FAPs (blue), presented in a 11 PCW heart section, showing their enrichment in the atrioventricular groove and external annulus fibrosus (dashed line). **F**. Violin plot showing the gene signature enrichment of the valve mesenchyme (VM) spatial cluster in all three fine-grained MC clusters related to the cardiac valves. **G**. Spatial mapping of EPDC_1 cells (red) and AnnFibr_FBs (green, upper panel), and VICs (red), Valve_MC_2 (green) and Valve_MC_1 cells (blue, lower panel), presented in a 7 PCW heart section, showing their enrichment in the atrioventricular plane and cardiac valves. Borders of major cardiac structural components are marked by solid lines. avg, atrioventricular groove; af, annulus fibrosus; mv, mitral valve; av, aortic valve; the scale bars represent 1 mm in the main and 0.5 mm in the zoom-in panels.

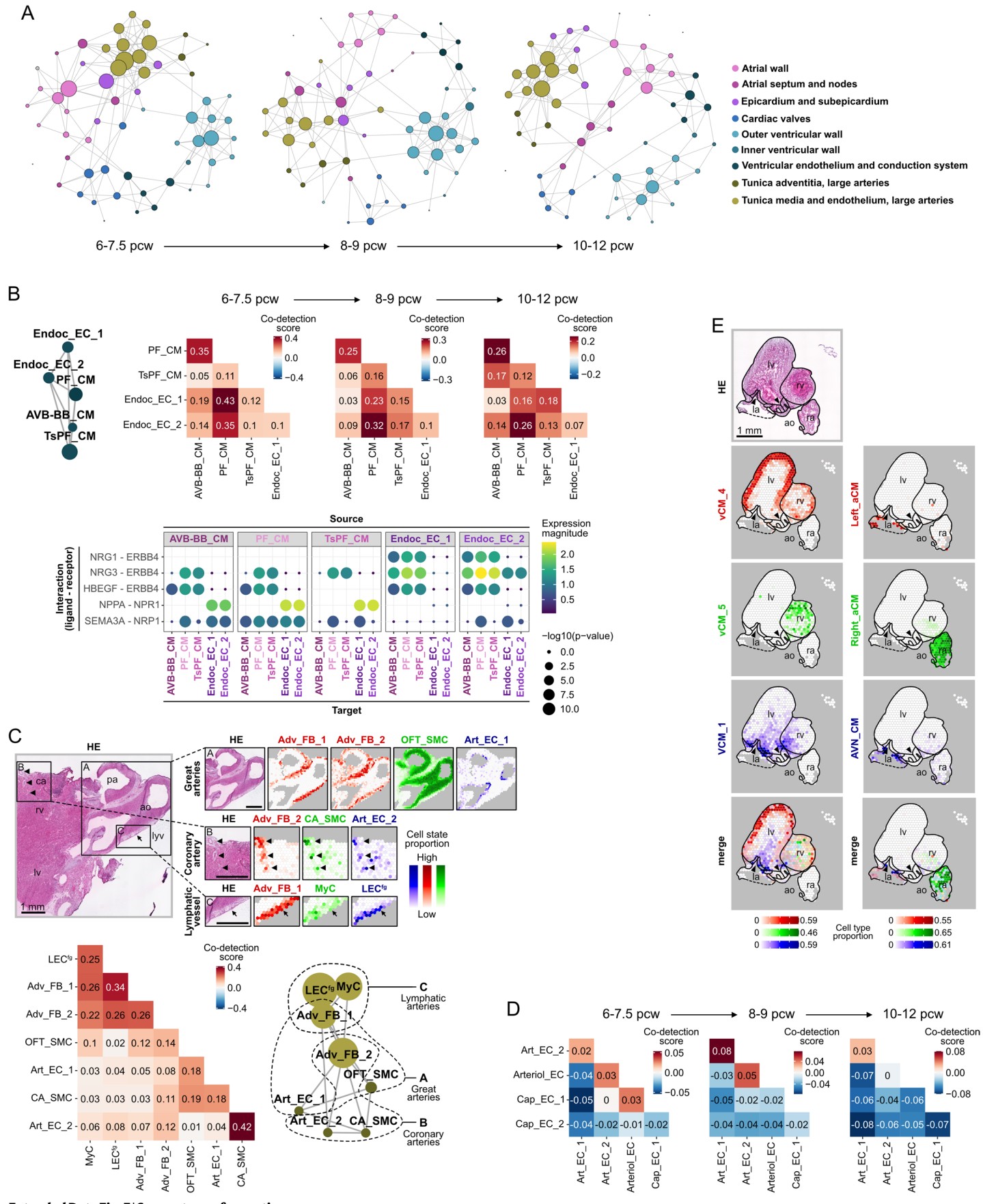

**Extended Data Fig. 7 | See next page for caption.**

**Extended Data Fig. 7 | Temporal Transitions and Analysis of Selected Developmental Cardiac Niches. A**. Temporal niche network graphs displayed across three developmental age groups (6-7.5, 8-9 and 10-12 PCW), calculated from time-resolved cell state co-detection scores. Circles represent fine-grained cell states, and grey lines indicate their spatial association. Circle size reflects the number of closely associated cell states, and different colors indicate distinct cardiac structural compartments, consistently with Fig. 7a. **B**. Niche network graph representing cellular components of the ventricular endocardium and conduction system (upper left), along with corresponding co-detection scores across three developmental age groups (6-7.5, 8-9 and 10-12 PCW, upper right). Selected ligand-receptor interactions, implicated in trabecular myocardium formation and Purkinje fiber specification, indicate functional differences between EndocEC_1 and Endoc_EC_2 cells, and PF_CMs and TsPF_CMs, respectively (lower). **C**. Spatial mapping displaying enrichment of Adv_FB_1 (red) and Adv_FB_2 cells (red), OFT_SMCs (green) and Art_EC_1 cells (blue) in the great arteries (ROI A); Adv_FB_2 cells (red), CA_SMCs (green) and Art_EC_2 cells (blue) in coronary vessels (ROI B, arrowheads); and Adv_FB_1 cells (red), MyCs (green) and LECfg cells (blue) around lymphatic vessels (ROI C, arrows, upper). Niche network graph illustrates spatial relations between cellular components of distinct cardiac vessel structures, outlined by dashed lines (lower right), with corresponding co-detection scores (lower left). **D**. Co-detection scores between vascular EC states, presented across three developmental age groups (6-7.5, 8-9 and 10-12 PCW), demonstrating their gradual spatiotemporal separation. **E**. Spatial mapping of vCM_4 (red), vCM_5 (green) and vCM_1 cells (blue) in the ventricles (left panel), and Left_aCMs (red), Right_aCMs (green) and AVN_CMs (blue) in the atria, highlighting cell states enriched on the left and rights sides of the heart and in the atrioventricular plane region, respectively. Notably, vCM_1 cells appear in high proportion at the roots of the cardiac valves. ca, coronary artery; lyv, lymphatic vessel; the scale bars represent 1 mm in both the main and the zoom-in panels.

# Reporting Summary

Nature Research wishes to improve the reproducibility of the work that we publish. This form provides structure for consistency and transparency in reporting. For further information on Nature Research policies, see our Editorial Policies and the Editorial Policy Checklist.

## Statistics

For all statistical analyses, confirm that the following items are present in the figure legend, table legend, main text, or Methods section.

| n/a | Confirmed | |
|---|---|---|
| ☐ | ☒ | The exact sample size ($n$) for each experimental group/condition, given as a discrete number and unit of measurement |
| ☐ | ☒ | A statement on whether measurements were taken from distinct samples or whether the same sample was measured repeatedly |
| ☐ | ☒ | The statistical test(s) used AND whether they are one- or two-sided *Only common tests should be described solely by name; describe more complex techniques in the Methods section.* |
| ☒ | ☐ | A description of all covariates tested |
| ☒ | ☐ | A description of any assumptions or corrections, such as tests of normality and adjustment for multiple comparisons |
| ☐ | ☒ | A full description of the statistical parameters including central tendency (e.g. means) or other basic estimates (e.g. regression coefficient) AND variation (e.g. standard deviation) or associated estimates of uncertainty (e.g. confidence intervals) |
| ☐ | ☒ | For null hypothesis testing, the test statistic (e.g. $F$, $t$, $r$) with confidence intervals, effect sizes, degrees of freedom and $P$ value noted *Give P values as exact values whenever suitable.* |
| ☒ | ☐ | For Bayesian analysis, information on the choice of priors and Markov chain Monte Carlo settings |
| ☒ | ☐ | For hierarchical and complex designs, identification of the appropriate level for tests and full reporting of outcomes |
| ☒ | ☐ | Estimates of effect sizes (e.g. Cohen's $d$, Pearson's $r$), indicating how they were calculated |

*Our web collection on statistics for biologists contains articles on many of the points above.*

## Software and code

Policy information about availability of computer code

| Data collection | Sequencing data from the embryonic and fetal heart spatial samples was processed with 10x Genomics Space Ranger version 1.2.1. Sequencing data from the embryonic and fetal heart single-cell samples was processed with 10x Genomics Cell Ranger version 4.0.0. |
|---|---|
| Data analysis | Data were analyzed on the statistical software R, with some scripts ran with Python, and some analyses performed on Excel. All analysis scripts and custom code are available at the public GitHub page: https://github.com/rmauron/HDCA_heart_dev. |

For manuscripts utilizing custom algorithms or software that are central to the research but not yet described in published literature, software must be made available to editors and reviewers. We strongly encourage code deposition in a community repository (e.g. GitHub). See the Nature Research guidelines for submitting code & software for further information.

## Data

Policy information about availability of data

All manuscripts must include a data availability statement. This statement should provide the following information, where applicable:
- Accession codes, unique identifiers, or web links for publicly available datasets
- A list of figures that have associated raw data
- A description of any restrictions on data availability

All the data required to replicate the analysis, including cellranger output, spaceranger output, metadata, processed ISS data, extended figures and tables, as well as main RDS objects are shared on the Mendeley DATA repository (links shared in the manuscript). The processed data of Visium, scRNA-seq and ISS are publicly available for browsing gene expression, clusterings, and other analysis results at https://hdcaheart.serve.scilifelab.se/web/index.html. The raw sequencing data can be shared upon reasonable request to the corresponding authors through EGA.

April 2020

# Field-specific reporting

Please select the one below that is the best fit for your research. If you are not sure, read the appropriate sections before making your selection.

☒ Life sciences ☐ Behavioural & social sciences ☐ Ecological, evolutionary & environmental sciences

For a reference copy of the document with all sections, see nature.com/documents/nr-reporting-summary-flat.pdf

# Life sciences study design

All studies must disclose on these points even when the disclosure is negative.

| | |
|---|---|
| Sample size | No sample-size calculation was performed. A close-to-even number of samples were included in the single-cell RNA-sequencing and Visium datasets (15 and 16 hearts, respectively), providing balanced coverage of the overlapping investigated developmental windows (5.5th-14th pcw for the single-cell RNA-sequencing and 6th-12th pcw for the Visium analysis). Age-resolved analysis of both datasets was performed with having at least three independent hearts in each compared age group. In the Visium analysis, two to four close-to-consecutive sections were processed, approximating technical replicates. For temporal comparison of single-cell cluster distributions, age-resolved populations were randomly downsampled to the size of the least abundandt group. The ISS dataset was generated from four hearts reflecting the age groups compared in the single-cell and Visium analyses, while supporting immunostaining was perfomed on three hearts and images were included in the manuscript from a single fetal heart sample. |
| Data exclusions | In the Visium analysis, we filter out low-count measurements and measurements outside the tissue. This is done for two reasons: first, it allows us to crop the data volume to the tissue area, thus reducing the size of the data and speeding up computing steps. Second, while there should be no gene expression outside the tissue, measurements may, in practice, be non-zero due to technical errors and diffusion. Therefore, we instead introduce a virtual measurement of the area outside the tissue which is forced to zero. These preprocessing steps were pre-established. The resulting count matrix was filtered for MALAT1, ribosomal, mitochondrial and hemoglobin genes.<br>In the single-cell analysis, cells with low counts were excluded. Additionally, cells with more than 30% mitochondrial transcript counts, less than 3% ribosomal transcripts, or more than 10% hemoglobin transcripts were removed. |
| Replication | Replication is attempted with close-to-consecutive sections for the Visium, ISS and immunostaining datasets, as described in the Methods section of the manuscript. |
| Randomization | Not applicable (no experimental groups) |
| Blinding | Not applicable (no experimental groups) |

# Reporting for specific materials, systems and methods

We require information from authors about some types of materials, experimental systems and methods used in many studies. Here, indicate whether each material, system or method listed is relevant to your study. If you are not sure if a list item applies to your research, read the appropriate section before selecting a response.

### Materials & experimental systems

| n/a | Involved in the study |
|---|---|
| ☐ | ☒ Antibodies |
| ☒ | ☐ Eukaryotic cell lines |
| ☒ | ☐ Palaeontology and archaeology |
| ☒ | ☐ Animals and other organisms |
| ☐ | ☒ Human research participants |
| ☒ | ☐ Clinical data |
| ☒ | ☐ Dual use research of concern |

### Methods

| n/a | Involved in the study |
|---|---|
| ☒ | ☐ ChIP-seq |
| ☒ | ☐ Flow cytometry |
| ☒ | ☐ MRI-based neuroimaging |

## Antibodies

| | |
|---|---|
| Antibodies used | anti-ARL13B (RRID:AB_3073658, ab136648, Abcam, 1:400), anti-PDE4C (RRID:AB_3094595, HPA054218, Atlas Antibodies, 1:100), anti-ATF3 (RRID:AB_1078233, HPA001562, Atlas Antibodies, 1:100) |
| Validation | All the utilized antibodies have been validated for IF application in human cells or tissues, according to the product documentation. |

## Human research participants

Policy information about studies involving human research participants

| | |
|---|---|
| Population characteristics | Gender and age of each sample included in the single-cell and Visium datasets is described in the manuscript. Although not |

| Population characteristics | chosen, sex distribution in the study was balanced (17 hearts from female and 14 from male donors). Sex of the donors included in the ISS dataset and IHC images was not investigated. Samples collected are all from the developmental period between 5.5th-14th postconceptional weeks (spatial datasets: 6th-12th pcw; single-cell dataset: 5.5th-14th pcw). |
| --- | --- |
| Recruitment | All heart specimens included in this study were collected from elective medical abortions at the Department of Obstetrics and Gynecology at Danderyd Hospital and Karolinska Huddinge Hospital in Stockholm, Sweden. Only individuals over 18 years old with full decision-making capacity and without diagnosed psychiatric conditions affecting consent were eligible to donate embryonic tissues. The patients donated tissue with written informed consent, after receiving both oral and written information about the purpose of the research project, and the possibility of retracting their consent at any time, including later destruction of the donated tissue. The study participants did not receive any kind of compensation for their donation. |
| Ethics oversight | The study was performed with approval of the Swedish Ethical Review Authority and the National Board of Health and Welfare, under the ethical permit number 2018/769-31, in accordance with Swedish regulations governing the use of prenatal tissue for medical research and treatment. |

Note that full information on the approval of the study protocol must also be provided in the manuscript.

