## [Peer Review File · Nature Genetics]

Spatiotemporal Gene Expression and Cellular Dynamics of the Developing Human Heart

Corresponding Author: Professor Joakim Lundeberg

Version 0:

Decision Letter:

17th Dec 2024

Dear Joakim,

Your Article, entitled "Spatial Dynamics of the Developing Human Heart", has now been seen by 3 referees. I apologize for the slow review process. You will see from the reviewers' comments below that while they find your work of interest, some important points are raised. We are interested in the possibility of publishing your study in Nature Genetics, but would like to consider your response to these concerns in the form of a revised manuscript before we make a final decision on publication.

All three reviews are overall positive about the technical aspects of this work. Importantly, reviewers #2 and #3 would like to see a better comparison with recently published datasets (in particular, Farah et al., recently published in Nature).

We invite you to revise your manuscript taking into account all reviewer comments. Please highlight all changes in the manuscript text file. At this stage we will need you to upload a copy of the manuscript in MS Word .docx or similar editable format.

We are committed to providing a fair and constructive peer-review process. Do not hesitate to contact me if there are specific requests from the reviewers that you believe are technically impossible or unlikely to yield a meaningful outcome.

*2) If you have not done so already please begin to revise your manuscript so that it conforms to our Article format instructions, available

[here](http://www.nature.com/ng/authors/article_types/index.html).

*3) Include a revised version of any required Reporting Summary: <https://www.nature.com/documents/nr-reporting-summary.pdf>

Please be aware of our [guidelines](https://www.nature.com/nature-research/editorial-policies/image-integrity) on digital image standards.

Link Redacted

Note: This URL links to your confidential home page and associated information about manuscripts you may have submitted, or that you are reviewing for us. If you wish to forward this email to co-authors, please delete the link to your

homepage.

Nature Genetics is committed to improving transparency in authorship. As part of our efforts in this direction, we are now requesting that all authors identified as 'corresponding author' on published papers create and link their Open Researcher and Contributor Identifier (ORCID) with their account on the Manuscript Tracking System (MTS), prior to acceptance. ORCID helps the scientific community achieve unambiguous attribution of all scholarly contributions. You can create and link your ORCID from the home page of the MTS by clicking on 'Modify my Springer Nature account'. For more information please visit please visit www.springernature.com/orcid.

Sincerely,

Tiago

Tiago Faial, PhD
Chief Editor
Nature Genetics
<https://orcid.org/0000-0003-0864-1200>

Reviewers' Comments:

Reviewer #1 (Remarks to the Author):

Summary

In this manuscript Lazar et al. combine spatial transcriptomics (Visium HD) and single cell RNA-sequencing data from human embryonic hearts across a variety of timepoints spanning from 5-6 to 14 weeks post conception. This encompasses a period of cardiac development during which the four chambered heart and critical structures such as the atrial septum, the pacemaker-conduction system and atrioventricular/semilunar valves undergo maturation and growth. This manuscript aims to identify the different cell types present across this period of development, providing insight into their contribution, embryonic origin and developmental trajectories. In parallel, the spatial transcriptomic dataset was used to examine the localization of these populations in situ, allowing for examination of cardiac niches and possible signaling interactions between neighboring cell types. This work extends upon the stages of development previously characterized by spatial transcriptomics as well as utilizing a higher resolution (55um spots for Visium in comparison to 100um in Asp et al 2019). The data has also been made readily available in an online browser for ease of use by the community, as a highly valuable resource for examining gene expression in the human embryonic heart.

An overarching question was over the use of fine grain clustering- how distinct are these groups and could a statistical test demonstrate that these fine grain clusters are different to each other? Also, how many cells/stages/samples are these clusters drawn from and can this be used to conclude that these truly represent developmentally relevant and distinct cell populations? Overall, the data presented is clearest and most informative when multiple stages are shown providing a temporal overview, as well as validating the presence of these cell types across developmental stages and providing more confidence in the validation of the relative positions/organization of the cell types.

Please find other comments below:

Figure 1

- The figure legend seems to be missing a description of panel D, with the description of D corresponding to panel E and E to F.
- In panel E the images demonstrating the side specific enrichment of NREP (LV) and CKM (RV) are a little hard to conclude from and look a little 'stripey' perhaps this is an issue with generating the pdf but currently this also seems to affect the panels for FHL2 and LGALS3 too (although in these the localization/pattern is clearer).

Figure 2

- For the single-cell RNA sequencing were whole hearts used? - In supplementary figure 12 it appears that a few thousand cells were collected from some of the later stage hearts (for example 256 or 274) but was this due to dissociation or only a portion of the heart being made available? If only a subset of cells was collected from these later stage hearts, how can one ensure this is an accurate representation of the underlying population in order to draw conclusions about changes in the proportion of each cell type over time? For example, in Figure 2F a decrease in CM is observed but could this be due to the above-mentioned reasons?

Figure 3

- In Extended data fig 3, the phrasing used is slightly confusing, stating that the localization of the Ts_PF and PF_CMs supports 'their gradual separation', from the images presented it rather seems that the Ts_PF are not present in the sections of the earlier stages? Do they move/re-organize into this spatial orientation relative to each other, or differentiate in this manner?
- In figure 3D the magenta color scheme is slightly hard to distinguish- in general it seems these genes are expressed in the SAN but also the AVN, which would differ from the dot plot on the left which suggests expression should be lower in the AVN? Similarly for the yellow/green colour scheme on the left, for example if one assumes that GDF10 is expressed in both the SAN and AVN, there does not appear to be a difference in expression level, in contrast to the dot plot? In 3F, there is more of a sense that the expression of the 4 genes examined does differ with the SAN/AVN looking more 'yellow', perhaps a magnified ROI could help to make these differences clearer?

Figure 4

- This portion of the manuscript examines the neuronal/Schwann-cell related populations within the heart and finds human-specific chromaffin cells. In Figure 4c and the accompanying text parallel developmental trajectories are described, originating from a Schwann cell precursor state- in the figure it is slightly hard to determine if the SCP populations are the 'starting point' for these trajectories? Or is the main text referring to the Bridge cells?
- In Figure 4h, it is unclear what the two blue spots next to the beta1R are corresponding to, would this be the SAN and AVN?
- At the end of the figure legend for Figure 4, the anatomical annotations are for panel E not D.
- In the main text, it is mentioned that the autonomic neurons express somatostatin, are corresponding receptors present or is this thought to not act locally?

Figure 5

- In panel B is quite hard see/distinguish the underlying anatomy in the first set of images, in the second set the SLV is more convincing, but it would be lovely to see the ROI for B and C too and possibly another example of an AVV? The third set with the RA/LA CMs overlaid, also makes this easier to interpret as a reader. (In general, it might be nice to rotate the hearts to a more consistent orientation? Particularly as some of the sections are less intuitive or missing (parts of) chambers)
- In 5A where the different EC of the coronary vessels are described it would be nice to see an example of these occupying distinct/sequential domains if available?

Figure 6

- In the main text the FAPs are mentioned to have a presumed epicardial origin, correlating with their close association with the EDPC_1 population, would trajectory analysis be helpful in this case to understand how these different populations relate to each other and what these progenitor cells may give rise to? Would this also fit with the supplementary discussion?
- In 6D is this spatiotemporal expansion or a decrease in the FB1 cell type and shift towards green (Int_FB_2)? Is there a change in localization and/or a change in the proportion of each? (Do Int_FB1 become FB2/3?)
- In 6E would it be possible to see more examples of the localization of valve MC_1/2- from this their localization does not appear distinct. A graph or plot illustrating the difference in PENK1 expression between these cell types would also be nice, if the point that is being made is that it is PENK expression which distinguishes these two cell types. In Figure 6F and in the main text, PENK expression is seen within the SLV cusps, does this correspond to the VIC cells or the valve_MC_2 cells?
- The model presented in 6G is quite hard to put together with the images presented in B/C/E and F. Perhaps an image of an AVV with the VIC, valve MC_1/2 populations and also an overlay of the 3 populations in the bottom half of 6C (to explain the organization of layers in the model) would help?

Figure 7

- The scaling of co-detection scores is described in the methods as '...a co-detection graph network was built, where co-detection scores above a manually selected arbitrary threshold of 0.07 were kept. All the other scores were set to 0 since we considered building graphs with negative values not meaningful', as the negative values were removed, should they be shown in the key?
- In C are the dashed boxes highlighting interactions between the SAN and Autonomic neurons or 'highlighting ligand-receptor interactions....'?
- In D, the localization of valve_MC_1 remains hard to place, it does not appear to be near the vCM1 which was previously described as 'aligned with the elusive AVP_V spatial cluster'- where are these cells in relation to the AVV? In the main text it says 'Notably, Valve_MC_1 cells showed strong enrichment of RSPO2, implicated in the development of bicuspid aortic valve disease 78,79, and Valve_MC_2 cells of which has been proposed as a key molecule in the formation of a fibrous anchor area between cardiac valves and adjacent tissue segments78, consistent with the predominant localization of these clusters.' Is 7D meant to demonstrate the valve_MC1 cells are close to the avv? Or are they meant to be localizing near the SLV in this image? This seems to be better described in Extended data Fig 6F top panel- perhaps this explanation and figure could be within moved to the main?

Extended data figures

- In extended figure 2D is the n referring to the number of clusters or cells or something else?
- In extended figure 2C would it be possible to visualize a marker of the endothelium? From the images it is slightly hard to pinpoint the location of these PDE4C positive cells, in the coronary vessels they appear close to the lumen corresponding with the endothelium, but for the ventricular wall ROIs it is slightly harder to see where/what these cells might be?
- In extended figure 3B could the enrichment score axes be standardized? Could some of the fine grain CM clusters be removed from this comparison (e.g. the immature CM) for ease of looking at this figure? Do any of the vCM populations show characteristics of the left/right ventricle and/or overlap with the LV/RV from the spatial clusters?

- In extended figure 3F and main figure 3E why is VCM4 used as the contractile state, was this selection explained?
- In extended figure 4 the legend is missing a description of D and the remaining figure legends are shifted accordingly.
- For the marker gene expression maps shown in extended figures 5 and 6, perhaps the annotated version beside would help to remember what each cluster is meant to be (it is described in the text but it might be easier to correlate this as the reader if it was presented alongside)
- Extended figure 7A is slightly hard to gain information from, it is difficult to determine which 'nodes' (or the cell states they correspond to) are becoming more/less 'connected'? Also, it would be nice to have one example of a temporal change- the co-detection plots are slightly hard to interrogate, are the scores comparable across timepoints, there does not seem to be an obvious change in scores, but were there any cell types which showed a change over time that would be of interest? Could this be expanded upon? Are these relatively small changes in score of interest? (Also, visually from the spatial transcriptomics what do these changes correspond to, are cell states that were previously not in contact brought closer together or vice versa?)

Supplementary figures and discussion

- Line 58 refers to supplementary figure 8 but I believe should be referring to supp. Fig. 1?
- Lines 131 and 208 refer to Supp Fig 7C but this does not exist?
- In supplementary figure 10 the labels appear to be switched for B and C
- In, Supp figure 12B the cutoffs corresponding to lines 856 to 862 are shown, was a cut off implemented for nCount or what does the dashed line indicate?

Reviewer #2 (Remarks to the Author):

Lázár, Mauron, Andrusivová, et al. report a cutting-edge study of spatially resolved and cell specific gene expression in the developing human heart. Conceptually this study appears to be a follow-up study to the landmark article published by the same lab five years ago (Asp et al., Cell 2019). That being said, the current study displays quite remarkable improvements in scale: scRNA-seq on 77k vs. 4k cells; spatial seq on 69k vs. 3k spots. Making use of these richer and higher resolution data, the authors report novel mechanistic insights into the organization and transcriptional features of several key components of the developing heart, including the pacemaker cell system, the autonomic innervation system, and the mesenchymal cell layers surrounding the cardiac valves. The authors also discover a resident population of chromaffin cells in the developing heart and incorporate these cells into a spatially defined model for the autonomic control of fetal heart function. The findings written in the text likely only represent a small fraction of the potential value of this study, given the scale of their impressive data set. To help increase utility of their data, the authors provide access to their processed data through a custom-made online viewer. I found the viewer to be both intuitive and highly useful, and was able to both check some of the observations reported in the paper (for instance, ACTA1 expression around the papillary muscles) and to conduct some explorations of my own. Overall, this study represents an impressive, rigorous, and substantial contribution to our understanding of the spatial organization of the developing human heart.

The current study is in methodology and scope similar to another recently published study (ref 33; Farah et al., Nature 2024). This reference is mentioned in the introduction; however, it is not mentioned again in the main text (and only briefly referred to in the Supplementary discussion in the context of specific findings). Given the similarities of these two studies, readers would benefit from a high-level comparison of the overlap and differences between the two studies. Importantly, the nascent field of spatial omics would benefit from knowing what aspects of these kinds of studies are reproducible, and whether there are aspects of their results that appear irreproducible and/or contradictory.

The methodology and data analysis are extensive and rigorous, and the presentation and writing is clear. The quality of the Visium and Chromium data is high and not overstated. The authors also provide in situ sequencing (ISS) data; however, these data appear to be of much lower quality and it is unclear to me how these data contribute in ways that could not be achieved with the Visium datasets. From visual inspection, these data seem to vastly underreport the actual number of transcripts, even when compared with high-dropout methods such as scRNA-seq. In order to assess the true capture rate of ISS, it would have been ideal if the authors could provide for one of the transcripts a low-dropout reference such as single-molecule RNA FISH, however that likely falls outside the scope of the current study. In lieu of such a reference standard, the authors should provide at least an estimate of the ISS capture rate (or inversely, the ISS dropout rate). Finally, the authors claim in the abstract that they performed "subcellular imaging-based transcriptomics", which I assume refers to the ISS. I was excited to learn more about this feature of their data, however I saw no further mention of subcellular resolution in the main or supplementary text, nor any evidence for the subcellular resolution playing any role whatsoever in any of the results or conclusions. Thus, though the subcellular resolution may technically be an accurate descriptor, I found the use of this term in the abstract to set up misleading expectations for the rest of the article. Nevertheless, it is up to the authors whether they choose to keep this term in the abstract (as they did in their previous article in 2019).

Major comments:

1. Given the similarities and seeming overlap between this study and the study by Farah et al. (ref 33), readers would benefit from a comparison of the overlap and differences between the two studies. What aspects of the previous study is reproduced here? Are there aspects of the results that appear irreproducible and/or contradictory? Though it is not the authors job to try and reproduce any specific aspect of the previous study, an effort should be made to place the current article's findings in the context this existing literature.

2. Please provide an estimate of the ISS capture rate.

Minor comments:

1. p. 35 lines 622-631: Claiming that the study would have been improved and that rare cells would have been better captured with an increased size of the data set is lazy writing. Given the estimated drop-out rates for all methods used (scRNA-seq, ST, and ISS) it seems to me that improving the current methods would be a much better way to achieve these goals. As world-leading users (and inventors! thank you for your service) of some of these methods, it is imperative that the authors point out what aspects of these methods are most crucial to be improved – to me it seems like capture rate in all three, though especially in ISS; single-cell segmentation in ISS and in ST; library size in ISS; spatial resolution in ST.

2. p. 34 line 591: The authors claim their study is “the most detailed spatiotemporal atlas of heart development in the 1st trimester”. In what way is this study more detailed than the study by Farah et al. (ref 33)? Genetically or spatially? Please specify.

3. In Figure 1, legend 1D-F are incorrectly labeled.

4. Ext. Fig 4D legend has a typo: “D. Gene Expression Related to Axon Guidance and Synapse Formation across Fine-Grained Cardiomyocyte States.”

5. Figure 5C: Is it certain that these detected transcripts are within endothelial cells? Are they also expressed in non-endothelial cells?

6. When integrating (and batch correcting) the single-cell and Visium data sets from different developmental time points (for instance for the purpose of clustering and UMAP plotting), how do you ensure that the integration procedure preserves developmental changes in cell states? (E.g. Suppl. Fig. 10B&C)

Reviewer #3 (Remarks to the Author):

Lázár and co-authors constructed a comprehensive spatiotemporal atlas of the developing human heart during the 1st and early 2nd trimester. It comprises 74 extensive datasets that include 69,114 spatially barcoded tissue spot transcriptomes and 76,801 single-cell transcriptomes from a total of 38 fetal human hearts, complemented by imaging 150 selected genes via in situ sequencing (ISS) and imaging select proteins using immunofluorescence. They identified 23 spatial tissue features with distinct transcriptional profile, 11 primary cell types and 72 fine-grained cell states that were mapped to corresponding regions in cardiac tissue sections by integrating spatial and single-cell transcriptomes and cell type deconvolution. This is a highly valuable data resource, in particular, for the understanding of early human heart development in the first trimester that has not been well characterized in the past. This data resource already allowed the authors to discover so many intriguing novel findings. For example, they dissected the spatial developmental process and the underlying molecular programs in the development of cardiac pacemaker-conduction system, the interactions with the autonomic innervation. They discovered a novel resident chromaffin cell population within the fetal heart that has implication in cardiac diseases. They dissected spatially heterogeneous endothelial and mesenchymal cell types and states in the developing valves and atrial septum. Finally, they constructed a map of spatial niches by examining the co-occurrence of different cardiac cell states. This highly valuable data resource is available through an interactive data portal, which will definitely increase the accessibility of these datasets and foster the utilization for independent discovery in relation to heart development and diseases. It is worth noting that the manuscript is beautifully written and figures well organized, making it really a joy to read and the key messages clearly delivered. In sum, this work provides a highly valuable and timely data resource, and yields numerous new insights that are of high interest to the field. This reviewer has only a few minor comments as the following.

(1) The authors have previously investigated human heart development at different stages (REF 32) and other studies in field also reported spatial niches and cell types in developing or adult human heart (e.g., REF 33, 42). Could the authors also examine the data from these published works to either perform integrative analysis to extend the findings of key cell types and states at the first trimester to later stages or compare selected cell types or niches between fetal and adult heart. Even comparing the results without full integration could be still enlightening. Thus, this is a missed opportunity for the authors to uncover more insights.

(2) Similarly, although it is extremely interesting to show numerous CHD or other CVD risk genes are associated with cell types in the first trimester of the developing human heart. Could the authors compare spatial molecular profiles or cell types associated with these genes between this work and published studies of human CVD single-cell or spatial maps. It does not need to fully integrated data analysis, but comparing findings in development and disease would be still quite insightful.

(3) Figure 2b – single-cell clustering UMAP. Cardiomyocytes, endothelial cells, are shown in two distinct regions in UMAP. Immature CMs look similar to EC. Lymphatic ECs are located next to CMs. Although the authors have carefully annotated and confirmed the cell types, but it is still very interesting why these totally different cell types localized in the “vicinity” of the UMAP space. Could this be attributed to cell doublet or RNA cross contamination, or something biologically interesting?

(4) Fig 3d, three or four different genes are shown in the same color but varying intensities. This is difficult to see and distinguish them by naked eyes. Could different colors be chosen to visualize the spatial distributions of these genes in this figure panel.

(5) Ion channel profile is super interesting! Could the author further ion channel mediated cell-cell interaction or action potential propagation? No one has done that and this is really optional.

(6) Fig 4c,d – usually PCA has limited resolution to differentiate different cell subsets to investigate temporal dynamics in

pseudotime or RNA velocity analysis. Could you also try UMAP or monocle, and compare?

(7) Fig 6. It is interesting to see the development of multiple types of valve-related endothelial, fibroblast, and mesenchymal cells, in particular, the difference in endothelial cells on the opposite sides of the valve leaflet, which is likely attributed to different flow and mechanical stimulation. Another dimension to investigate the valvular cell heterogeneity is from the base to the tip. In particular, Fig 5f, using ISS, one can resolve more spatial features of valvular cells.

Version 1:

Decision Letter:

Our ref: NG-A66482R

28th Mar 2025

Dear Joakim,

Thank you for submitting your revised manuscript "Spatial Dynamics of the Developing Human Heart" (NG-A66482R). It has now been seen by the original referees and their comments are below. The reviewers find that the paper has improved in revision, and therefore we'll be happy in principle to publish it in Nature Genetics, pending minor revisions to satisfy the referees' final requests and to comply with our editorial and formatting guidelines.

Thank you again for your interest in Nature Genetics. Please do not hesitate to contact me if you have any questions.

Congratulations!

Sincerely,

Tiago

Tiago Faial, PhD
Chief Editor
Nature Genetics
<https://orcid.org/0000-0003-0864-1200>

Reviewer #1 (Remarks to the Author):

The authors have extensively addressed all my comments.

Reviewer #2 (Remarks to the Author):

The authors have done a fantastic job of thoroughly answering all the concerns. In particular, the supplementary discussion of a comparison to the Farah et al. study will be of great use to the field as it addresses inconsistencies that would otherwise have been passed on to scientists exploring and comparing these two data sets in the future. I found Supplementary Discussion 2 to be highly informative in how it addresses the specific aspects of the cell annotations that differ between the two studies. The ISS capture rate comparison was also useful and helps in interpreting the authors' results.

I have two remaining comments/requests and I hope to see the article in print soon.

1) Fig. 5A&B of Response to Reviewers is extremely useful for interpreting the ISS data in the context of other modalities such as scRNAseq. These two figure panels should therefore be part of the manuscript.

2) From looking at Fig. 5B of Response to Reviewers, it appears that the mean of the max expression for ISS is less than $5E-2$, and the mean of the max expression for scRNAseq is more than $5E0$, which means that the difference is greater than 100-fold. However, in the response letter the authors state that "Our analysis revealed that scRNAseq was 62.5 times more efficient than ISS across the 149 common genes assessed." Am I reading the graph wrong? In either case, the correct value should be reported in the manuscript as it is an important metric relevant to any interpretation of their data.

Reviewer #3 (Remarks to the Author):

The authors have done an excellent job in the revision that has fully addressed all my major questions. In particular, the comparison of key observations between Farah et al, and this manuscript really helped strengthened the key findings and the further analysis of spatial cell heterogeneity across the valves elucidated the potential mechanisms in line with biomechanics and physiological functions. I also took the liberty to briefly read the responses to other reviewers. Reviewer 2 and I raised some common concerns that have been well addressed. Also, agree with Reviewer 1 that this work presents the most comprehensive and informative datasets to dissect the spatial temporal dynamics of human heart development. Thus, this is a highly valuable resource article and reports a significant contribution to the field. I would recommend accept as it.

Dear Reviewers,

We sincerely thank you for taking the time and effort to evaluate our manuscript, and we were happy to receive your constructive feedback and the overall positive assessment of our work. We have now carefully addressed all your raised concerns during the revision of our manuscript, and we provide point-to-point replies to all your comments below.

We have incorporated many of your suggestions into the revised version of the manuscript, including the addition of several new figures and figure panels, as well as the rearrangement of some existing ones to enhance clarity and organization. Corresponding changes have been made throughout the text, including extensive updates to figure references. In the figures, we have also improved the spatial plots to provide better visualization of gene expression and cell state distribution, ensuring the presentation is as clear and informative as possible. We also identified and corrected several typographical errors, along with a few inaccuracies in figure references, and revised the style of the extended figure legends to ensure consistency with other figure legends in the manuscript. Apart from corrections for typographical and grammatical errors, and minor rearrangements of the text following the updated order of figure panels, all changes made to the text are highlighted in red for ease of review.

Importantly and in line with the request of Reviewer #2 and Reviewer #3, we conducted an in-depth comparison of our study with the recently published work of Farah et al.¹, which we have included in our revised submission as Supplementary Discussion 2. This comparison provides additional context and further supports the significance of our findings.

We would also like to note that we discovered a misalignment between the UMAP clusters and DEG panels in the single-cell module of our interactive viewer. This issue has now been corrected, ensuring the accuracy and reliability of the viewer for future use.

We believe these revisions have significantly strengthened the manuscript, and we hope that our responses to your comments demonstrate our commitment to addressing your feedback thoroughly.

Joakim Lundeberg and Enikő Lázár

29th December, 2024

Reviewer #1:

Summary

In this manuscript Lazar et al. combine spatial transcriptomics (Visium HD) and single cell RNA-sequencing data from human embryonic hearts across a variety of timepoints spanning from 5-6 to 14 weeks post conception. This encompasses a period of cardiac development during which the four chambered heart and critical structures such as the atrial septum, the pacemaker-conduction system and atrioventricular/semilunar valves undergo maturation and growth. This manuscript aims to identify the different cell types present across this period of development, providing insight into their contribution, embryonic origin and developmental trajectories. In parallel, the spatial transcriptomic dataset was used to examine the localization of these populations in situ, allowing for examination of cardiac niches and possible signaling interactions between neighboring cell types. This work extends upon the stages of development previously characterized by spatial transcriptomics as well as utilizing a higher resolution (55um spots for Visium in comparison to 100um in Asp et al 2019). The data has also been made readily available in an online browser for ease of use by the community, as a highly valuable resource for examining gene expression in the human embryonic heart.

We would like to thank you for your time and effort invested in evaluating our manuscript. We were happy to receive your comments affirming the value of our presented results, datasets, and the built-on interactive viewer, to the scientific community. We would like to emphasize that in this study we utilized standard Visium slides (and not Visium HD slides which were just released recently), and the high granularity of the presented datasets was, in fact, achieved through our selected analysis strategy, combining single-cell RNA-sequencing with unbiased (10x Visium) and targeted (*in situ* sequencing) spatial transcriptomics to investigate the cellular and molecular profiles of developing human hearts.

We have carefully considered your raised points and below we address all of them in detail. Furthermore, we have integrated several of your suggestions into the revised version of the manuscript and corrected the highlighted editing mistakes. We believe that through the implementation of these changes we managed to substantially improve our manuscript and address any unclarity.

An overarching question was over the use of fine grain clustering- how distinct are these groups and could a statistical test demonstrate that these fine grain clusters are different to each other? Also, how many cells/stages/samples are these clusters drawn from and can this be used to conclude that these truly represent developmentally relevant and distinct cell populations? Overall, the data presented is clearest and most informative when multiple stages are shown providing a temporal overview, as well as validating the presence of these cell types across developmental stages and providing more confidence in the validation of the relative positions/organization of the cell types.

Thank you for your valuable feedback. We appreciate the opportunity to address these points and provide clarity on our clustering approach, and our effort to showcase biological validity of the resulting cell populations.

Defining discrete clusters in populations representing a continuum of cell states is a general challenge for the single-cell omics field. While being a prerequisite for any comparative analysis between different populations, setting arbitrary cutoff points to separate closely related cell states carries the risk of introducing artificial groups without a clear biological character. In our work, we have employed several assessment levels to ensure the distinctiveness of our fine-grained clusters, including the following: i. Lack of batch effects or other technical parameters driving the clustering; ii. Relevant transcriptomic differences between the single-cell clusters based on calculating differentially expressed genes; iii. Predicted localization and spatial relations between clusters; iv. Alignment with known molecular and cellular events of cardiogenesis; v. Validation from independent datasets.

i. Firstly, we opted to perform unsupervised clustering of the dataset to generate clusters based on apparent transcriptomic variability, with a similar strategy used for subclustering in three selected populations of cardiomyocytes, used to refine distinct components of the pacemaker-conduction system (SAN_CM and AVN_CM; TsPF_CM and PF_CM) and less defined ones in the atrioventricular plane (vCM_1). We controlled that none of the clusters were formed by a single sequencing sample (potentially highlighting technical artifacts driving the clustering instead of true transcriptomic differences), with special attention to the potential presence of transitional developmental cell states. **Suppl. Fig. 13C** (Suppl. Fig. 10C of the original manuscript) highlights overall good integration of different heart samples even in an age-resolved manner, supporting that cluster identities are unlikely to be majorly defined by batch effects. Upon your request, we have now assembled detailed information on the distribution of different heart samples and sequencing runs in each coarse- and fine-grained cell state in updated versions of **Suppl. Table 17 and 18**, respectively, also supporting the contribution of several developmental stages, individual hearts, and sequencing samples to each of these populations.

ii. Next, by calculating and investigating differentially expressed genes between clusters across and within cellular subsets, we ensured that the obtained clusters are in fact characterized by biologically meaningful gene expression profiles, and largely similar populations still feature enrichment of several transcripts explaining their separation. The entire DEG tables are available in **Suppl. Table 2-6** for the coarse-grained and subsetted fine-grained clusters, with the top 5 DEGs also being visualized for coarse-grained clusters in the newly added **Suppl. Fig. 2A**, and per subset in **Ext. Fig. 3A, 4A, 5A and 6A**, and even for comparisons of more refined clusters of interest in **Fig. 6F, Ext. Fig. 5F**, and **Suppl. Fig. 3C, 5C and 11B-C** (corresponding to Ext. Fig. 5F and 6E, and Suppl. Fig. 2C and 4C of the original manuscript, and new figure panels). Additionally, we present many figure panels showing specific enrichment of known or novel markers of biologically relevant cell states across the whole range or selected populations of fine-grained cell states, often accompanied by spatial validation of these molecular features. Importantly, all this information is also made available for independent exploration in our updated interactive viewer.

iii. When selecting a biologically reasonable clustering granularity, we heavily relied on cell state mapping results obtained through the integration between our unbiased single-cell RNA sequencing and spatial transcriptomics datasets. This strategy led to confident, spatially aware identification of several minor cell states (such as spatially distinct components of the cardiac valves and pacemaker-conduction system, or endothelial cells in the atrial septum, among others) that could not have been distinguished from one

another without deducing the spatial coordinates of these cell states, providing further validation to our clustering approach.

iv. The previous approaches are widely accepted by the scientific community to address differences between single-cell clusters; however, they provide limited insight into the biological soundness of differentiating between the defined cell states. Therefore, we also took careful consideration of how the transcriptomic and topographical profiles of these clusters align with our current knowledge on human cardiogenesis on the molecular, cellular, and tissue structural tiers. We routinely considered these three aspects (i. distinct gene expression profile, ii. characteristic localization within the intact tissue, iii. alignment with known events of the cardiogenic process) of the obtained clusters when deciding on a reasonable clustering granularity and annotating the resulting cell states. Our labeling strategy reflects this effort by incorporating all these layers of information into the used annotation system, as detailed in the 'Annotation of Spatial and Single-Cell Clusters' section of the Methods (**lines 994-1009** of the revised manuscript). This approach goes well beyond standard label-transfer strategies, which substantially limit the potential to discover novel cell states, as well as the standard practice of retaining only cluster numbers on fine-grained clustering levels without providing any assistance in interpreting cluster identities or related findings.

v. Additionally, we have performed an in-depth comparison between our spatially annotated single-cell dataset and the one recently published by Farah et al.¹, which we present in a separate document enclosed to revised manuscript (Supplementary Discussion 2). Based on similar enrichment of selected markers and the results of a label transfer strategy, we could confirm the equivalents of several of our newly annotated, fine-grained cell states in this independent dataset, further validating their presence and relevance in the human cardiogenic process.

In summary, we provide a wide range of evidence for the distinctiveness and relevance of the fine-grained cell states identified in our study. Additionally, we have now employed clustree, a visualization tool for interrogating clusterings at increasing resolutions, arranging clusters in a hierarchical tree to highlight groups sharing varying levels of transcriptional similarities. This analysis supported the need for the applied high-resolution clustering to separate populations with distinct transcriptional and biological properties and tissue localization explored in detail in our study, such as the SAN_CM and AVN_CM, PF_CM and TsPF_CM, and vCM1 clusters in the cardiomyocyte subset; Endoc_EC_1 and Endoc_EC_2, Endoc_EC3 and Endoc_EC_4, Art_EC_2 and Arteriol_EC, Ven_EC and Venul_EC, and OF_VEC and AtrSept_EC in the endothelial cell subset; EPDC_1 and Infl_FB in the mesenchymal cell-fibroblast subset; and Aut_Neu_1 and Aut_Neu_2 in the innervation-related cellular subset. Furthermore, the separation of other fine-grained cell states even at lower clustering resolutions highlights more significant transcriptomic differences compared to the previously discussed cell types, supporting the validity of our clustering approach. For further insights, we have now included the results of the clustree analysis in the revised manuscript as **Suppl. Fig. 17A-D**.

Nevertheless, we agree that further temporal analysis could provide additional support for our clustering. However, the size of the current single-cell RNA sequencing dataset only permits such analysis with

sufficient statistical robustness at the coarse-grained clustering level. We trust that continued advancements in the field, both in terms of published single-cell and spatial datasets and advances of the applied molecular profiling technologies, will further validate our findings and offer even deeper insights into the cellular and molecular mechanisms underlying human cardiogenesis.

Please find other comments below:

Figure 1

- The figure legend seems to be missing a description of panel D, with the description of D corresponding to panel E and E to F.*

Thank you for drawing our attention to this error. We have now corrected the mislabeling in the legend of Figure 1 and added the following description for panel D: ‘Relative enrichment of consensus cell type markers and selected differentially expressed genes ($\log_2FC > 0$, $p_val < 0.05$) across the 23 spatial clusters.’ (**lines 115-116** of the revised manuscript).

- In panel E the images demonstrating the side specific enrichment of NREP (LV) and CKM (RV) are a little hard to conclude from and look a little ‘stripey’ perhaps this is an issue with generating the pdf but currently this also seems to affect the panels for FHL2 and LGALS3 too (although in these the localization/pattern is clearer).*

While part of your concern can, in fact, be explained by pdf conversion of the figures, we agree that the current visualization of these gene patterns could be improved. For genes with substantial expression across all regions of the tissue section (e.g. *NREP*, *CKM*, etc.) it is more challenging to highlight spatial enrichment than for the ones with more restricted expression patterns (e.g. *MASP1*, *MT3*, etc.). When generating spatial feature plots, we refrained from modifying the presented numerical scales by introducing minimal cutoff values, which might have helped the visualization but could present a misleading view on the real distribution of the affected transcripts. Therefore, in the revised manuscript, we opted to change the color scale for the spatial feature plots in **Fig. 1E**, which we believe provides a clearer intensity resolution and better demonstrates the regional enrichment of the highlighted transcripts.

Figure 2

- For the single-cell RNA sequencing were whole hearts used? - In supplementary figure 12 it appears that a few thousand cells were collected from some of the later stage hearts (for example 256 or 274) but was this due to dissociation or only a portion of the heart being made available? If only a subset of cells was collected from these later stage hearts, how can one ensure this is an accurate representation of the underlying population in order to draw conclusions about changes in the proportion of each cell type over time? For example, in Figure 2F a decrease in CM is observed but could this be due to the above-mentioned reasons?*

To generate samples for single-cell RNA sequencing, we always dissociated entire hearts, which we then subsampled to achieve cell numbers consistent with the capacity of multiplex runs in the Chromium

platform, in a range between 2,352 and 11,358 heart cells per sequencing run (2,667 and 22,706 cells per analyzed heart). In addition, we opted to increase the number of analyzed cells through including a higher number of hearts from the same developmental window instead of sampling more cells from a single heart, thus ensuring higher biological variability in our data. Accordingly, we did not introduce selection bias for any structural component of the developing heart, however, with its increasing size, cellular abundance, and complexity, the proportions of major cellular compartments have substantially changed along the temporal axis. Consequently, with the temporal divergence of the abundance of different cell states, scarcer populations might have been captured less often in the sequenced cellular subsets from older hearts. As mentioned above, we now provide detailed information on the contribution of different heart samples and sequencing runs to each coarse- and fine-grained cell state in an updated version of **Suppl. Table 17-18**, allowing for independent investigation of cluster composition of any heart sample analyzed in our single-cell RNA-sequencing dataset.

Even with these considerations, we were cautious to draw major conclusions from temporal changes in cell type proportions within our dataset, especially between clusters of different major cellular subsets. Different cell types might have pronouncedly different tolerance towards the tissue dissociation protocol (or even the medical abortion process beforehand), which might be reflected in their relative abundance after filtration of high-quality cells in the dataset. This effect might be reflected in the decreasing proportion of cardiomyocytes in our dataset; however, it is not possible to control for this technical aspect. Consequently, we refrained from drawing any conclusion from the temporal proportion change between major cellular subsets, illustrated in the lowermost panel of **Fig. 2F**, and rather described it as a metrics of our clustering in the figure legends ('Temporal changes in coarse-grained cluster proportions in the cardiomyocyte, endothelial cell, and fibroblast-mesenchymal cell subsets across four developmental age groups (5.5-6, 7-8, 9-11, and 12-14 pcw)', **lines 165-167** of the revised manuscript). We presumed that even if similar technical caveats might have some effect between clusters within the same cellular subset, they are likely less pronounced, thus temporal trends drawn from this data (presented in the upper three panels of **Fig. 2F** and discussed in **lines 88-91 and 117-125** of Supplementary Discussion 1) are likely more reliable than the ones observed between cellular subsets. This is largely supported by the alignment of the observed trends with known cardiogenic events, discussed in the above-mentioned paragraphs.

Figure 3

• In Extended data fig 3, the phrasing used is slightly confusing, stating that the localization of the Ts_PF and PF_CMs supports 'their gradual separation', from the images presented it rather seems that the Ts_PF are not present in the sections of the earlier stages? Do they move/re-organize into this spatial orientation relative to each other, or differentiate in this manner?

Thank you for highlighting this ambiguity. We agree that the wording of this figure legend leaves room for different interpretations. Based on the presented results, we can only reflect on the spatial relation between the two analyzed cell states, and not developmental connection, which, while highly plausible, could not be convincingly assessed in our study. For better visualization, we decided to include separate

plots for the PF_CM and TsPF_CM cell states along with the overlaid images in **Ext. Fig. 3C**, with the updated figure panel better illustrating the spatiotemporal evolution of the mentioned cell states.

As visualized in green, TsPF_CMs are mapped to the ventricular regions already in the earliest analyzed developmental stages, although in smaller per-spot proportions than PF_CMs. Even in these samples, PF_CMs appear more central towards the lumen of the chamber, while TsPF_CMs appear in a more peripheral position. Additionally, the spatiotemporal separation of these cell states is highlighted by the increasing ratio of spots marked only by green in later developmental stages. While part of this change is explained by the growth, and consequently decreasing cellular density of the developing heart, it also suggests a more pronounced regionalization of PF_CMs and TsPF_CMs in increasing depths from the endocardium. To avoid overstating this observation, we rephrased the figure legend as following: ‘Spatial mapping of PF_CMs (red) and TsPF_CMs (green) in 6, 8, 10, and 12 pcw heart sections, illustrating their temporally consistent arrangement along different depths of the ventricular wall.’ (**lines 46-47** of Extended Data Information).

• In figure 3D the magenta color scheme is slightly hard to distinguish - in general it seems these genes are expressed in the SAN but also the AVN, which would differ from the dot plot on the left which suggests expression should be lower in the AVN? Similarly for the yellow/green colour scheme on the left, for example if one assumes that GDF10 is expressed in both the SAN and AVN, there does not appear to be a difference in expression level, in contrast to the dot plot? In 3F, there is more of a sense that the expression of the 4 genes examined does differ with the SAN/AVN looking more ‘yellow’, perhaps a magnified ROI could help to make these differences clearer?

Thank you for identifying the issue with the visualization in these figure panels. The genes displayed in the dot plot and ISS images in **Fig. 3D-E** were chosen based on their differential enrichment in the SAN_CMs (all listed genes) and AVN_CMs (*SHOX2*, *ZNF385B*, *TENM2*, *GDF10*, *SLIT2*), compared to other fine-grained cardiomyocyte clusters (**Suppl. Table 3**). We recognize that using dot plots to present the relative enrichment of these genes can be misleading, as the color scale represents gene expression levels relative to the average value calculated only across the populations included in the plot.

To address this, we have replaced the dot plot panels in **Fig. 3D-E** with heatmaps, showing the scaled absolute expression levels of the selected genes on a positive scale. This approach provides a clearer representation of the DEG profiles in a manner more consistent with **Suppl. Table 3**, highlighting the enrichment of specific genes, including *GDF10*, in both SAN_CMs and AVN_CMs.

In the complementary ISS panels in **Fig. 3D-E**, we aimed to illustrate the overall spatial enrichment of these gene products in tissue segments consistent with SAN and AVN positions, however, limitations on the image size and resolution hinders the independent visualization of these transcripts. Additionally, while SAN_CMs and AVN_CMs show enrichment of some of the plotted genes compared to other cardiomyocyte states, cells from other subsets may also express these genes. For instance, certain fine-grained states of the innervation-related subset and the CALN1^{high}_FB population show substantial expression of teneurin (*TENM2*, *TENM3*, *TENM4*) genes, which play key roles in axon guidance and

synaptic stability, possibly important for establishing the newly forming cardiac innervation. In our manuscript, we map these cells to regions closely associated with the nodal tissue. Thus, these non-myocyte cell states likely contribute to the SAN- and AVN-related expression patterns detected in the ISS plots, with the assayed genes potentially playing roles in forming molecular contacts between nodal cells and their developing innervation.

To improve the transparency of our findings, we have assembled a new figure (**Suppl. Fig. 9A-C**) that presents the ISS signal for each gene listed in **Fig. 3D-E** independently (panel A and B, respectively), and an additional dot plot showing the relative enrichment of these genes across all fine-grained clusters of the single-cell dataset (panel C).

Figure 4

• This portion of the manuscript examines the neuronal/Schwann-cell related populations within the heart and finds human-specific chromaffin cells. In Figure 4c and the accompanying text parallel developmental trajectories are described, originating from a Schwann cell precursor state- in the figure it is slightly hard to determine if the SCP populations are the 'starting point' for these trajectories? Or is the main text referring to the Bridge cells?

We thank you for pointing out the need for further clarification regarding the position of Schwann cell precursors (SCPs) and bridge cells in the developmental trajectories observed in the innervation-related cellular subset.

In our dataset, we have annotated bridge cells based on their enrichment in *ASCLI*, while the non-bridge SCP population featured markers such as *SOX10* and *FOXD3* (**Fig. 4B**). Based on the marker and cluster assignment on the PCA embedding of our innervation-related cellular subset (**Fig. 4C** left panel, **D**), we observed that the connecting trajectories bifurcate at SCPs rather than at bridge cells, although some bridge cells fall in proximity to the bifurcation point in our embedding. This is most likely due to the lower technical resolution because of the overall low number of cells in the analyzed subset (n=870). We also carefully investigated the results of our trajectory analysis and found that the nesting of a bifurcation point aligned with SCPs and not the bridge state.

We took care to perform the trajectory analysis in a technically sound manner, after consulting the developer teams of the applied tools. In trajectory analysis using scFates² (**Fig. 4C**, right panel), we manually designated the transcript *PENK* as an input as recommended by the developer team, since *PENK* expression resides in locations most distal to the expression of markers demarcating relatively more mature cell types (*SST* for neuroblasts and *PMP22* for mature glial cells) in the PCA embedding. For RNA velocity analysis, we adhered strictly to the tutorial provided for the scVelo tool, which does not require manual root assignment. Instead, root cells are inferred automatically based on the directed velocity graph, as detailed in the publication by Bergen et al.³ introducing the tool.

Taken together, our trajectory analysis results align well with the current state of literature, describing SCPs as developmental origin to both glial and neuronal trajectories, and bridge cells representing a cell state with early commitment towards the neuronal lineage.

• In Figure 4h, it is unclear what the two blue spots next to the beta1R are corresponding to, would this be the SAN and AVN?

In **Fig. 4H**, we added colored dots after the displayed gene products to highlight their relative enrichment in specific components (SAN_CM - blue dot, AVN_CM - green dot, PF_CM - pink dot) compared to relevant contractile cardiomyocyte states, consistent with results displayed in **Fig. 3F** and **Ext. Fig. 3F**. To clarify this, we added the following text to the related figure legend: ‘Blue, green and pink dots beside the displayed genes represent relative enrichment in SAN_CMs, AVN_CMs and PF_CMs, respectively, in comparison to relevant contractile cardiomyocyte states displayed in Fig. 3F and Ext. Fig. 3F.’ (**lines 349-351** of the revised manuscript).

• At the end of the figure legend for Figure 4, the anatomical annotations are for panel E not D.

Thank you for drawing our attention to this mistake, we have now corrected the figure legend.

• In the main text, it is mentioned that the autonomic neurons express somatostatin, are corresponding receptors present or is this thought to not act locally?

SST is expressed in the majority of developing autonomic neurons; however, its receptors (*SSTR1–5*) are minimally expressed in cardiac cells within our dataset. Cardiomyocytes, including components of the pacemaker-conduction system, show virtually no expression of these receptors. Therefore, it is unlikely that SST directly influences cardiac automacy or conduction during this developmental stage. Interestingly, Art_EC_1 cells, corresponding to endothelial cells of the great arteries, differentially express *SSTR1*, while IF_VECs, representing endothelial cells on the ‘inflow’ side of the cardiac valves, express *SSTR2*. This suggests that SST may regulate these endothelial cell populations during the analyzed period of heart development.

Moreover, in the single-cell dataset of developing hearts (9th-16th postconceptional weeks) presented by Farah et al.¹, a subset of developing neurons (misannotated as neural crest cells by the authors) expresses *SSTR2*. This observation suggests a possible autocrine feedback loop, where *SSTR2* expression in these neurons could limit neurotransmitter release from the same population. However, we could not confirm this finding in our dataset. Differences in developmental stages, as well as variations in technical approaches such as sample processing and data filtering (discussed in detail in Supplementary Discussion 2) may explain this discrepancy.

Figure 5

• In panel B is quite hard see/distinguish the underlying anatomy in the first set of images, in the second set the SLV is more convincing, but it would be lovely to see the ROI for B and C too and possibly another example of an AVV? The third set with the RA/LA CMs overlaid, also makes this easier to interpret as a

reader. (In general, it might be nice to rotate the hearts to a more consistent orientation? Particularly as some of the sections are less intuitive or missing (parts of) chambers)

Thank you for highlighting this issue. As requested, we have updated **Fig. 5B** to include independent panels for the hematoxylin-eosin image (without Visium spots overlaid) and magnified views of ROI B and C, in addition to the previously included ROI A and D. We have also created a new supplementary figure panel, showcasing another example of the arrangement of IF_VECs (green) and OF_VECs (red) in the atrioventricular valve (**Suppl. Fig. 11D**).

Additionally, we have introduced new symbols to **Fig. 5B-C** and **Suppl. Fig. 11D** to mark the opposite sides of the sampled cardiac valves (asterisks for the ‘outflow’ and arrowheads for the “inflow” side), ensuring consistency across these image panels. To maintain the logical flow of the manuscript, we have moved the atrial septum-related images into a separate figure panel (now **Fig. 5D**), relabeled the original panels, and rearranged the manuscript text accordingly (**lines 368-383** of the revised manuscript).

We agree that consistent orientation of heart sections throughout the spatial dataset would improve clarity, however, certain limitations of the Visium assay make this challenging. The Visium capture areas are square-shaped, and achieving consistent orientation would require varying degrees of rotation and flipping of images and the spatially detected gene expression data. While possible, this approach introduces technical risks, including potential misalignment between gene expression data and histological images, and could reduce spatial efficiency in the figure panels. Furthermore, the significant structural changes in the heart during the analyzed developmental period precluded consistent positioning of samples during embedding and sectioning, resulting in non-uniform sectioning planes and structures. Considering these factors, we have retained the original orientations of the sections in both the ISS and Visium datasets within the online viewer, and even in most figure panels, with marking major anatomical landmarks in the corresponding hematoxylin-eosin images for clarity regarding their positions. However, for Visium panels highlighting temporal trends across developmental stages (**Fig. 1C, 2D and 6D; Ext Fig. 2B and 3C; Suppl. Fig. 1A, 7B, 11B-C and 12D**) or side-specific cell state or gene enrichment (**Fig. 1E-F, Ext. Fig. 1B, D-E, 3D-E and 7E**), and figure panels with ISS data (**Fig. 3D-F, 5C and 6G; Suppl. Fig. 9A-B**), we have standardized the section orientations and related transcriptomic data according to anatomical references. To remain consistent, we occasionally adjusted the orientation of Visium panels for single sections as well when they also appeared with a modified orientation in other panels of the same figure (e.g. illustrating temporal trends, as mentioned above).

• In 5A where the different EC of the coronary vessels are described it would be nice to see an example of these occupying distinct/sequential domains if available?

Upon your request, we have now generated spatial plots highlighting the predicted localization of several endothelial cell types in the great arteries and coronary vasculature in a 10 pcw heart section (**Suppl. Fig. 11A-B** of the revised manuscript). As described in our manuscript and visualized in the upper panel of **Suppl. Fig 11A**, out of the two populations with clear arterial character, Art_EC_1 (red) shows specific spatial enrichment in the intima of the great arteries (aorta highlighted in ROI A), while Art_EC_2 (green)

is enriched in coronary arteries in the atrioventricular and interventricular sulci (ROI B). In close proximity to the structures featuring a high proportion of the latter cell type, we also find a predominant localization of the Venul_EC population (blue) with venular transcriptomic characteristics, in positions consistent with coronary veins (ROI B). In the lower panel, predicted spatial distributions of Arteriol_EC (red), Cap_EC_1 (green) and Cap_EC_2 (blue) cell states are displayed, representing the predominant localization of the arteriolar and two capillary subpopulations, respectively. Interestingly, Cap_EC_1 and Cap_EC_2 appear in different depths of the cardiac walls, with Cap_EC_2 mostly present in the outer, and Cap_EC_1 in the inner layers of the ventricular myocardium (**Suppl. Fig. 11B** of the revised manuscript). For additional insights, we have included a dot plot with the top differentially enriched markers between these populations, highlighting several important determinants of endothelial phenotype and modulators of vasculogenesis in the developing heart (e.g. *KIT*, *INSR* and *CXCR4* vs. *KLF2*, *KLF4* and *THBD*); however, further exploration of these populations is beyond the scope of the current study. Additionally, we have also included an image panel depicting the position and differentially expressed markers of the four fine-grained endocardial cell states (**Suppl. Fig. 11C**), providing a close-to-comprehensive overview of the most relevant endothelial cell states in the developing heart.

Figure 6

• In the main text the FAPs are mentioned to have a presumed epicardial origin, correlating with their close association with the EPDC_1 population, would trajectory analysis be helpful in this case to understand how these different populations relate to each other and what these progenitor cells may give rise to? Would this also fit with the supplementary discussion?

We appreciate your insightful suggestion regarding the use of trajectory analysis to further elucidate the developmental relationship between the EPDC_1 and FAP populations. We attempted the proposed analysis with several different parameters and cellular subsets of the epicardial and mesenchymal cell-fibroblast populations, however, this effort did not yield conclusive results. Several technical challenges likely contributed to this outcome, including the relatively limited size and heterogeneity of the developmentally diverse dataset, and presumably less pronounced transcriptional transitions between cell states in the mesenchymal subset (compared to other compartments, such as the neural-glia lineage, where meaningful trajectories were detectable despite the dataset's size limitations). It is also worth noting that, while FAPs show pronounced spatial enrichment in the atrioventricular groove, they are also detected in a more dispersed pattern, albeit at lower per-spot proportions, throughout the ventricular myocardium at later developmental stages. This broader distribution, which we have now briefly reflected on in **line 454** of the revised manuscript, highlights additional complexity in their spatial and potentially developmental dynamics.

Importantly, we based our prediction regarding a possible developmental connection between these populations primarily on their spatial enrichment observed in the atrioventricular groove, where FAPs are closely associated with the EPDC_1 cell state. This finding aligns with the widely accepted notion that most cardiac fibroblasts are derived from an epicardial origin, with fibroblasts of proposed endocardial or neural crest origins were described to localize to largely distinct cardiac regions (valves, parts of the

subendocardium, ventricular septum, etc.). While not conclusive proof, the expression of *TBX18* - a consensus marker of epicardial cells - by the FAPs (along with other mesenchymal fibroblast populations) in our dataset provides indirect support for the proposed developmental connection between the epicardium and these cell states. We have now added the related dot plot as **Suppl. Fig. 5E** to the revised manuscript and included these considerations in **lines 266-275** of Supplementary Discussion 1 for further clarity.

• *In 6D is this spatiotemporal expansion or a decrease in the FB1 cell type and shift towards green (Int_FB_2)? Is there a change in localization and/or a change in the proportion of each? (Do Int_FB1 become FB2/3?)*

In **Fig. 6D**, we aimed to highlight the gradual expansion of Int_FB_1 and Int_FB_2 cell states from the subepicardial layer towards different depths of the myocardium. These cell states are consistently observed throughout the entire age range of our single-cell dataset, confirming that the regionalized distribution in the ventricular wall, as indicated by the cell state mapping results, genuinely reflects differences in the localization of these populations. Additionally, Int_FB_3 populates the innermost layers of the ventricular walls and the region closest to the atrioventricular plane, however, this cell state does not appear to originate from the same subepicardial layer like the other two. These findings align with observations by Farah et al.¹, who identified transcriptionally distinct fibroblast populations in the compact and trabecular layers of 13 pcw hearts using MERFISH analysis. Notably, some markers (e.g. *SCN7A*, *COLEC11*) associated with the Int_FB_1 and Int_FB_2 cells in our study (compared to other fine-grained mesenchymal cell-fibroblast clusters) demonstrated consistent relative enrichment between the Compact vFibro and Trabecular vFibro populations in their work.

Figure 1. Age-resolved proportions of the Int_FB_1, Int_FB_2 and Int_FB_3 cell states.

Notably, the Int_FB_1, Int_FB_2, and Int_FB_3 populations exhibit relatively stable ratios over time in our single-cell dataset (**Fig. 1 of Response to Reviewers**). This stability suggests that these clusters may represent parallel cell states; however, a developmental relationship between them remains plausible, with their transcriptional differences potentially reflecting gene expression changes driven by the evolving molecular microenvironment as epicardium-derived cells migrate into deeper layers of the myocardium.

As previously discussed, trajectory analysis of the epicardial and mesenchymal cell-fibroblast subsets, including Int_FB_1 and Int_FB_2 cells, was inconclusive due to various technical and biological factors. Consequently, we were unable to either confirm or rule out a direct developmental connection between these cell states.

• In 6E would it be possible to see more examples of the localization of valve MC_1/2- from this their localization does not appear distinct. A graph or plot illustrating the difference in PENK1 expression between these cell types would also be nice, if the point that is being made is that it is PENK expression which distinguishes these two cell types. In Figure 6F and in the main text, PENK expression is seen within the SLV cusps, does this correspond to the VIC cells or the valve_MC_2 cells?

Thank you for drawing our attention to the unclarity of differences between the Valve_MC_1 and Valve_MC_2 populations. Among several other differentially expressed genes, one of the main distinguishing features between these cell states is indeed the expression of *PENK*, which is strongly enriched in Valve_MC_2 but is only present at low levels in Valve_MC_1 cells. This difference is evident in the dot plot shown in **Ext. Fig. 6A**. However, for easier reference, we have relocated Ext. Fig. 6E of the original manuscript, illustrating the relative enrichment of marker genes such as *PENK* and *LEF1* (also visualized spatially in the accompanying ISS panel) across the three fine-grained valve-related mesenchymal cell states, to **Fig. 6F** of the revised manuscript. Although the scarcity of the ISS data and limited target panel size prevented us from confidently assigning the detected *PENK* and *LEF1* transcripts to segmented cells in our ISS-based spatial analysis, their spatial distribution (with *PENK* enriched in regions near the root of the semilunar valve cusps) and alignment with our Visium data-based cell type mapping suggest that most of the *PENK* signal detected in the semilunar valves likely originates from the Valve_MC_2 population.

Importantly, beyond their gene expression profiles, we also observed important differences in the localization patterns for the Valve_MC_1 and Valve_MC_2 populations, based on our cell state mapping results. Valve_MC_2 cells appeared associated with both sets of cardiac valves, localizing close to the roots of the semilunar cusps while reaching deeper into the atrioventricular valve leaflets. Furthermore, this population also extended well into the connective tissue between separate valves, substantially contributing to the internal regions of the annulus fibrosus, in contrast to AnnFibr_FB cells, which were more prominent in the subepicardial region and likely forming the outer rim of this structure.

In contrast, Valve_MC_1 cells appeared specifically associated with the semilunar valves and were largely absent in and around the atrioventricular valves. Within the semilunar valves, Valve_MC_2 cells were more frequently observed along the outer rim of the valves, while Valve_MC_1 cells extended deeper into the cusps from the roots. However, the resolution of the standard Visium approach limits precise spatial delineation of these populations in relation to each other within these structures. Furthermore, it is challenging to capture both semilunar and atrioventricular valves within the same section, which hinders the understanding of the relative positions of these cell states. Additionally, we observed temporal patterns in our dataset, with Valve_MC_2 cells showing gradual expansion and increasing per-spot proportions in the structures discussed above along the investigated timeframe, while Valve_MC_1 showing gradually

decreasing per-spot proportions associated with the semilunar valves. This observation aligns with slight temporal changes in the proportion of valve-related mesenchymal single-cell states along the investigated time frame (*Fig. 2 of Response to Reviewers*), suggesting that the Valve_MC_1 population is more prominently present at earlier developmental stages.

Figure 2. Age-resolved proportions of the VIC, Valve_MC_1 and Valve_MC_2 cell states.

To present the arrangement of valve-related mesenchymal cell states more effectively, we replaced the image panel in **Fig. 6E** with the one from Ext. Fig. 6F of the original manuscript (as suggested in your later comment), which includes the semilunar valves along with extended regions of the annulus fibrosus in a 12 pcw heart section. Additionally, we created a new figure panel (**Ext. Fig. 6G**) representing an earlier developmental stage (7 pcw), which includes structures related to both the semilunar and atrioventricular valves, offering a clearer depiction of their spatial relationships and temporal changes in distribution compared to the section shown in **Fig. 6E**. We have also expanded the corresponding section of the text to provide a more detailed description of the observed spatial arrangements of VICs and Valve_M C_1 and 2 cells (**lines 462–470** of the revised manuscript). Importantly, these cell states can be visualized in any of the 38 Visium sections in our dataset via our interactive viewer for independent evaluation. Higher-resolution spatial analysis in future studies will provide a more comprehensive understanding of these populations' spatial relationships and their potential roles in valve development and disease.

• The model presented in 6G is quite hard to put together with the images presented in B/C/E and F. Perhaps an image of an AVV with the VIC, valve MC_1/2 populations and also an overlay of the 3 populations in the bottom half of 6C (to explain the organization of layers in the model) would help?

Thank you for pointing out the lack of clarity in our proposed model of mesenchymal cell arrangement within the atrioventricular plane. In designing the original model (Fig. 6G in the original manuscript, now **Fig. 6H** in the revised version), our aim was to integrate general insights from the complete Visium dataset. However, only a few analyzed sections include all relevant structures - such as the semilunar and atrioventricular valves, the internal and external regions of the annulus fibrosus, and the atrioventricular groove - making it challenging to convincingly depict the relative positions of all relevant cell states in a

single section. Systematic comparisons across the entire dataset are therefore essential to understand the proposed spatial relationships.

To address this, we have made substantial updates to the relevant figure panels, as partly discussed in our reply to your previous comment. For example, in **Fig. 6E**, we now visualize all six relevant cell states within one section: in the upper panel, VIC is shown in red, Valve_MC_2 in green, and Valve_MC_1 in blue; in the lower panel, EPDC_1 is shown in red, AnnFibr_FB in green, and FAP in blue. These states are depicted around the semilunar valves (ROI slv), the internal and external regions of the annulus fibrosus (af, outlined), and the atrioventricular groove (ROI avg). Additionally, we have included a new panel (**Ext. Fig. 6G**) showcasing these populations in a 7 pcw heart section (except for FAPs, which are sparse at this developmental stage). This section encompasses both the inflow and outflow regions of the left ventricle, including both atrioventricular and semilunar valves, further confirming the enrichment of Valve_MC_1 cells around semilunar valves and the presence of Valve_MC_2 cells associated with both valve types. Additionally, we have reorganized the related segments of the text to follow the order of the new and revised figure panels (**lines 437-461** of the revised manuscript).

Sections containing only atrioventricular valves do not reliably visualize the Valve_MC_1 cell state. Given these limitations and manuscript length restrictions, we opted not to include an additional panel separately for these structures. However, our interactive viewer enables the visualization of these cell states in any of the 38 sections in the Visium dataset.

Finally, we have refined our model in **Fig. 6H** by adjusting colors and repositioning Valve_MC_1 cells to better illustrate the spatial arrangement of the highlighted cell states. We have also expanded the related figure legend to support the interpretation of the proposed model (**lines 505-510** of the revised manuscript). We believe these changes provide a clearer representation of mesenchymal cell and fibroblast states in the atrioventricular plane and cardiac valves, robustly supporting our conclusions.

Figure 7

• The scaling of co-detection scores is described in the methods as ‘...a co-detection graph network was built, where co-detection scores above a manually selected arbitrary threshold of 0.07 were kept. All the other scores were set to 0 since we considered building graphs with negative values not meaningful’, as the negative values were removed, should they be shown in the key?

Indeed, although the graph construction method is explained in detail in the Methods (**lines 1038-1052** of the revised manuscript), the related figure panels might require further clarification; therefore, we have now included the utilized cut-off value (0.07) into the legend of **Fig. 7A** (**line 587** of the revised manuscript).

Importantly, this cut-off value was determined arbitrarily, in order to generate a co-detection graph well-aligning with expected structural domains of the developing heart. Nevertheless, co-detection values lower than 0.07 also occur even between cell states within the same niche and structural domain and might carry useful information about relative cellular arrangements within these units. For this reason and to retain the

highest level of transparency in our analysis, we opted to include all co-detection values in the heatmaps supporting our niche detection (**Fig. 7A-B and D; Ext. Fig. 7B-D**), highlighted by the zero-centered scales of these plots.

• In C are the dashed boxes highlighting interactions between the SAN and Autonomic neurons or 'highlighting ligand-receptor interactions....'?

The two panels of **Fig. 7C** display the results of cell-cell communication analysis between the SAN_CMs, their cellular neighbors, and chromaffin cells (exerting paracrine mediation on nodal activity, thus not necessarily being in direct contact with SAN_CMs), using a modified version of the recent neural-GPCR module extension of CellPhoneDB⁴. This figure panel displays the enrichment of ligand-receptor pairs, analyzed in a direction-specific manner in the paired clusters, highlighting potentially relevant interactions determining the development and function of the nodal region. The ligand-receptor pairs outlined by dashed boxes between SAN_CMs and Aut_Neu_2 cells are the same ones included in **Ext. Fig. 4G**. Since this distinction does not convey any further information in **Fig. 7C**, we decided to remove the dashed boxes from this figure panel.

• In D, the localization of valve_MC_1 remains hard to place, it does not appear to be near the vCM1 which was previously described as 'aligned with the elusive AVP_V spatial cluster' - where are these cells in relation to the AVV? In the main text it says 'Notably, Valve_MC_1 cells showed strong enrichment of RSPO2, implicated in the development of bicuspid aortic valve disease 78,79, and Valve_MC_2 cells of which has been proposed as a key molecule in the formation of a fibrous anchor area between cardiac valves and adjacent tissue segments⁷⁸, consistent with the predominant localization of these clusters.' Is 7D meant to demonstrate the valve_MC1 cells are close to the avv? Or are they meant to be localizing near the SLV in this image? This seems to be better described in Extended data Fig 6F top panel- perhaps this explanation and figure could be within moved to the main?

In **Fig. 7D**, we aimed to demonstrate that the six cell types (IF_VEC, OF_VEC, VIC, Valve_MC_1, Valve_MC_2, and vCM_1) identified through co-detection analysis are not only closely associated spatially but also exhibit high and specific per-spot densities in the regions of the cardiac valves and surrounding tissue segments. This confirms their role as major cellular components of these structures.

As outlined in our response to earlier questions on the spatial arrangement of valve-related mesenchymal cell clusters, the Valve_MC_1 population is specifically enriched in and around the root regions of the semilunar valves. In contrast, it is largely absent near the atrioventricular valves. Meanwhile, the Valve_MC_2 population is present in the roots of both semilunar and atrioventricular valves, as well as within the internal connective tissue segments of the annulus fibrosus. We believe that the extensive revisions to Fig. 6 and Ext. Fig. 6, as discussed in our previous responses, including the suggested relocation of the original Ext. Fig. 6F into a main image panel (**Fig. 6E**) of the revised manuscript, more effectively convey these findings.

As discussed in our manuscript, the vCM_1 population shows substantial spatial overlap with the AVP_V spatial cluster, occupying a highly specific position at the base of the ventricular myocardium adjacent to

the atrioventricular plane (**Ext. Fig. 3E**). This unique localization underscores its distinction as the only cardiomyocyte state closely associated with valve-related endothelial and mesenchymal cell states, further supported by a novel figure panel and related comments (**Ext. Fig. 7E, lines 143-147** of Extended Data Information, **lines 205-206** and **550-552** of the revised manuscript), along with our co-detection graph and corresponding scores in **Fig. 7D**. Notably, vCM_1 shows the greatest spatial overlap with the Valve_MC_2 population and significantly less overlap with Valve_MC_1, reflecting the broader distribution of these cell states across the atrioventricular plane.

Interestingly, Farah et al.¹ identified a ventricular cardiomyocyte population with transcriptional similarities to the vCM_1 state in our dataset, which they mapped to the atrioventricular region, including atrioventricular valve leaflets and the attachment points of semilunar cusps (**lines 141-155** and **434-436** of Supplementary Discussion 2). This population, labeled as vCM-LV/RV-AV, reinforces our findings, and highlights the relative arrangement of these cell populations within and around the developing cardiac valves.

Extended data figures

• In extended figure 2D is the n referring to the number of clusters or cells or something else?

Similar to Ext. Fig. 2D (**Suppl. Fig. 2B** in the revised manuscript), we have included the number of cells or spatial spots presented on each UMAP in the main or extended figures of the manuscript (**Fig. 1B, 2B, 2E, 3A, 4A, 5A and 6A; Ext. Fig. 1C**) to facilitate their evaluation. Importantly, we corrected a previously erroneous value of the number of cells displayed on the UMAP in **Fig. 2B** (n=73,946). For conformity, we have now included this parameter into UMAPs displayed in other supplementary figures, too (**Suppl. Fig. 1B, 3A, 4A, 5A, and 13B-C**). Accordingly, in **Suppl. Fig. 2B**, ‘n’ refers to the number of cells displayed on each UMAP, representing cell composition changes in consecutive age groups. The four age groups defined in our Chromium dataset consisted of different numbers of single cells, 8,742 cells being the number of cells in the least abundant age group. To meaningfully visualize temporal changes between these age groups, we randomly downsampled the more abundant ones to the same population size. This strategy is described in the Methods under the section ‘Coarse-Grained Clustering and Analysis of the Single-Cell RNA Sequencing Data’ (**lines 934-936** of the revised manuscript), and now we have added this information to the legend of **Suppl. Fig 2B** (**lines 81-83** of Supplementary Data Information). A similar downsampling strategy was used to display the temporal evolution of spatial clusters in the Visium dataset, presented in **Ext. Fig. 1C**. This is described in the Methods under the section ‘Processing and Analysis of Visium Spatial Gene Expression Data’ (**lines 726-729** of the revised manuscript), and now also included in the related figure legend (**lines 18-20** of Extended Data Information).

• In extended figure 2C would it be possible to visualize a marker of the endothelium? From the images it is slightly hard to pinpoint the location of these PDE4C positive cells, in the coronary vessels they appear close to the lumen corresponding with the endothelium, but for the ventricular wall ROIs it is slightly harder to see where/what these cells might be?

In line with your request, we have performed new immunostainings on 9 pcw heart sections with the addition of fluorescently conjugated Ulex Europaeus Agglutinin I (UEA I) lectin, an established reagent for highly specific labeling of human endothelial cells. Additionally, we targeted PDE4C or ATF3 proteins, identified as genes enriched in the PDE4C^{high}_EC and PDE4C^{high}_FB populations, along with the broad cilia marker ARL13B and the nucleus marker Hoechst. Due to limitations on available fluorescent channels, this time we did not target the basal body and centrosome marker PCNT in the antibody panel. Due to their increased information content and better resolution, we decided to replace the original images in Ext. Fig. 2C with the new ones including the endothelial cell labeling and made consistent modifications to the related figure legend (**lines 31-39** of the Extended Data Information), parts of the Methods (**lines 821-841** of the revised manuscript), and interpretation (**lines 106-116** of the Supplementary Discussion 1). Nevertheless, upon editorial request we can readily provide the original figure panel in an additional supplementary figure.

Similarly to the immunostaining results included in our original manuscript, we detected ciliation, as well as PDE4C and ATF3 protein expression across the entire fetal heart, including, but not exclusive to, vessel walls. Both the endothelial and subendothelial cell layers of larger coronary arteries on the surface and somewhat deeper layers of the myocardium showed subcellular enrichment of both PDE4C and ATF3 proteins in discrete structures closely associated with the labeled cilia, consistent with the position of basal bodies.

These results support our earlier conclusions that PDE4C and ATF3 are previously uncharacterized components of the ciliary machinery in a wide range of cells in the developing human heart, and the PDE4C^{high}_EC and PDE4C^{high}_FB populations might indicate distinct cell states related to ciliary signaling. Importantly, our results provide a basis for future investigations of specific components and regulators of cardiac cilia during development, with the potential to expand our slowly increasing understanding of molecular diversity of these intriguing organelles⁵.

• In extended figure 3B could the enrichment score axes be standardized? Could some of the fine grain CM clusters be removed from this comparison (e.g. the immature CM) for ease of looking at this figure? Do any of the vCM populations show characteristics of the left/right ventricle and/or overlap with the LV/RV from the spatial clusters?

According to your request, we modified the **Ext. Fig. 3B** through standardizing the y axis and including only fine-grained cardiomyocyte clusters which were further discussed in the related segments of the manuscript.

Regarding the fine-grained cardiomyocyte clusters with ventricular cardiomyocyte characteristics, we observed side-specific enrichment of vCM_4 in the left and vCM_5 in the right ventricular walls based on our deconvolution results (**Fig. 3A of Response to Reviewers**). However, this lateralization did not appear as distinct as in case of the side-specific atrial cardiomyocyte populations (Left_aCM and Right_aCM), thus we refrained from annotating these clusters based on this observation. Furthermore, the comparison approach between transcriptomic profiles, used for generating the plots included in Ext. Fig.

3B, did not demonstrate consistent, side-specific similarities between the vCM_4 and vCM_5 single-cell and the LV_C and RV_C, or LV_T and RV_T spatial clusters (**Fig. 3B of Response to Reviewers**), respectively, therefore we decided not to include these results in the same figure panel.

Figure 3. **A**, Side-specific spatial enrichment of vCM_4 and VCM_5 is less consistent across the investigated developmental phases. **B**, Comparison between transcriptomic profiles of vCM_4 and VCM_5 single-cell and LV_T, LV_C, RV_T and RV_C spatial clusters does not support side-specific alignment between these populations.

One possible reason for this discrepancy might be the more complex cellular composition of the ventricular wall compared to the other highlighted tissue regions (represented by the displayed LA, RA, AVP_V, AVP_A and VCS spatial clusters), which might obscure side-specific characteristics of the resident cardiomyocyte cell states. Importantly, these populations displayed consistent enrichment of the left and right ventricular markers *SLCIA3* and *PRRX1* recently described by Farah et al.¹, respectively, supporting their side-specific arrangement within the developing ventricular myocardium.

• *In extended figure 3F and main figure 3E why is VCM4 used as the contractile state, was this selection explained?*

We selected vCM_4 for comparisons with the ventricular conduction system components because it displays the highest level of maturation markers among the fine-grained ventricular cardiomyocyte states, thus we consider it the closest in its transcriptomic profile to mature contractile cardiomyocytes (**Fig. 4 of Response to Reviewers**). Because of length restrictions we could not include this image as a figure panel in the manuscript, but we added the following explanation to the figure legend of **Fig. 3F (lines 262-264 of the revised manuscript)** and **Ext. Fig. 3F (lines 53-55 of Extended Data Information)**: ‘...and/compared to vCM_4, characterized by the highest maturation state among the contractile ventricular cardiomyocyte clusters.’.

Figure 4. Differences in maturation state between fine-grained cardiomyocyte clusters.

• In extended figure 4 the legend is missing a description of D and the remaining figure legends are shifted accordingly.

Thank you for highlighting this issue. The legend for **Ext. Fig. 4** indeed lacks a caption for image panel D, resulting in misalignment of the captions for panel D, E, and F with their respective figure panels. We apologize for this oversight. In the revised manuscript, we have included a caption for panel D as the following (**lines 67-69** of Extended Data Information): ‘D. Ligand-receptor scores, adapted from a recently published neural-GPCR module of CellPhoneDB, highlighting different sources of adrenergic and cholinergic mediation of SAN_CM function between the Chrom_C and Aut_Neu_2 states.’ We have also correctly aligned the other captions with the corresponding figure panels.

• For the marker gene expression maps shown in extended figures 5 and 6, perhaps the annotated version beside would help to remember what each cluster is meant to be (it is described in the text but it might be easier to correlate this as the reader if it was presented alongside).

Upon your suggestion, we have now added the annotated UMAPs to **Ext. Fig. 5B** and **6B** to facilitate clearer interpretation and improve the overall readability of the presented data.

• Extended figure 7A is slightly hard to gain information from, it is difficult to determine which ‘nodes’ (or the cell states they correspond to) are becoming more/less ‘connected’? Also, it would be nice to have one example of a temporal change- the co-detection plots are slightly hard to interrogate, are the scores comparable across timepoints, there does not seem to be an obvious change in scores, but were there any cell types which showed a change over time that would be of interest? Could this be expanded upon? Are these relatively small changes in score of interest? (Also, visually from the spatial transcriptomics what do these changes correspond to, are cell states that were previously not in contact brought closer together or vice versa?)

In our analysis, we introduced co-detection scores as metrics to quantify the spatial overlap between fine-grained cell states. These scores were calculated based on pairwise correlations of spatial cell state proportion predictions derived from the stereoscope analysis. To illustrate the utility of this parameter, we generated co-detection graphs, which provide insights into the cellular composition of major cardiac structures and finer niches, as well as support spatially informed downstream analyses.

Our primary objective in presenting age-resolved co-detection graphs was to demonstrate the relative consistency of defined niches and cardiac regions across the analyzed time frame (**lines 523–525** of the revised manuscript). This consistency is critical to ensure that downstream spatial analyses, such as cell-cell communication within cellular niches, can reliably build on the co-occurrence of these cell states across the entire temporal span of the dataset. Most co-detection heatmaps (**Fig. 7B, D; Ext. Fig. 7B-C**) in our study highlight the relatively high co-detection scores among cell states within specific cardiac structures or niches.

However, the co-detection scores as defined in our study have technical limitations, making them less suitable for temporal analysis. These include the inability to account for changes in cell state complexity or the temporal decrease in cell density during early cardiac development, introducing a technical bias when comparing age-resolved co-detection scores. As a result, a temporal decrease in co-detection scores may reflect developmental separation, simple tissue growth (and reduced cell density), or a combination of both. Additionally, the co-detection scores are influenced by variations in the cardiac areas sampled in our Visium dataset, which could not be perfectly standardized across age groups. Therefore, comparisons of temporally resolved co-detection scores should not be used to draw major biological conclusions.

Instead, we choose an alternative approach. To gain insights into temporal patterns, we visualized the relative co-occurrence of selected cell states using heatmaps that are standardized within each developmental window. For example, in **Ext. Fig. 7D**, we applied this method to analyze coronary endothelial cell states, revealing a temporal decrease in relative co-occurrence between endothelial populations along the capillary-great artery axis without major changes in the absolute co-detection scores. These heatmaps suggest an increasing separation between most cell states over time, but as noted earlier, this effect could partly reflect decreasing cell density in the tissue. Similarly, heatmaps in **Ext. Fig. 7B**, which include components of the ventricular subendocardium and conduction system, show a temporal shift in the relative co-detection values between pairs of the Endo_EC_1 and Endo_EC_2, and PF_CM and TsPF_CM population. This trend supports a gradual separation between the PF_CM and Endo_EC_2, and the TsPF_CM and Endo_EC_1 populations in the most internal and slightly deeper regions of the trabecular myocardium, respectively.

Supplementary figures and discussion

- *Line 58 refers to supplementary figure 8 but I believe should be referring to supp. Fig. 1?*

Thank you for pointing out this mistake, we have corrected the figure references in this paragraph (**lines 58-59** and 73 of the revised Supplementary Discussion 1) from Suppl. Fig. 8 to **Suppl. Fig. 1**.

- *Lines 131 and 208 refer to Supp Fig 7C but this does not exist?*

We apologize for this error. The statements ought to refer to **Ext. Fig. 7C** and not Suppl. Fig. 7C. We have now corrected the figure references in the text accordingly (**lines 132 and 208** of the revised Supplementary Discussion 1).

- *In supplementary figure 10 the labels appear to be switched for B and C*

Thank you for highlighting this editing error. Now we have corrected both the labels in the image file and the order of the related figure legends.

- *In, Supp figure 12B the cutoffs corresponding to lines 856 to 862 are shown, was a cut off implemented for nCount or what does the dashed line indicate?*

Thank you for highlighting this inconsistency. In Suppl. Fig. 12B of the original manuscript (**Suppl. Fig. 15B** in the revised manuscript), we presented several quality metrics for the single-cell RNA-sequencing dataset, including nFeatures, percentage of mitochondrial transcripts, percentage of ribosomal protein-coding transcripts, and percentage of hemoglobin gene expression. The dashed lines in these plots represent the filtering cutoffs applied during data processing. However, we did not apply any filtering based on nCount values, so the dashed line at 4000 UMIs in the nCount plot may be misleading, therefore we removed it from the revised version of this figure.

Reviewer #2:

Lázár, Mauron, Andrusivová, et al. report a cutting-edge study of spatially resolved and cell specific gene expression in the developing human heart. Conceptually this study appears to be a follow-up study to the landmark article published by the same lab five years ago (Asp et al., Cell 2019). That being said, the current study displays quite remarkable improvements in scale: scRNAseq on 77k vs. 4k cells; spatial seq on 69k vs. 3k spots. Making use of these richer and higher resolution data, the authors report novel mechanistic insights into the organization and transcriptional features of several key components of the developing heart, including the pacemaker cell system, the autonomic innervation system, and the mesenchymal cell layers surrounding the cardiac valves. The authors also discover a resident population of chromaffin cells in the developing heart and incorporate these cells into a spatially defined model for the autonomic control of fetal heart function. The findings written in the text likely only represent a small fraction of the potential value of this study, given the scale of their impressive data set. To help increase utility of their data, the authors provide access to their processed data through a custom-made online viewer. I found the viewer to be both intuitive and highly useful, and was able to both check some of the observations reported in the paper (for instance, ACTA1 expression around the papillary muscles) and to conduct some explorations of my own. Overall, this study represents an impressive, rigorous, and substantial contribution to our understanding of the spatial organization of the developing human heart.

We would like to thank you for the thoughtful and encouraging feedback on our work. We are pleased to hear that you found the expanded scale and resolution of our data helpful for exploring the spatial organization of the developing human heart, and that the online viewer provided a useful platform for both validating our findings and enabling additional exploration. The recognition of our efforts to build on our previous work while offering novel mechanistic insights into human heart development is greatly appreciated.

The current study is in methodology and scope similar to another recently published study (ref 33; Farah et al., Nature 2024). This reference is mentioned in the introduction; however, it is not mentioned again in the main text (and only briefly referred to in the Supplementary discussion in the context of specific findings). Given the similarities of these two studies, readers would benefit from a high-level comparison of the overlap and differences between the two studies. Importantly, the nascent field of spatial omics would benefit from knowing what aspects of these kinds of studies are reproducible, and whether there are aspects of their results that appear irreproducible and/or contradictory.

Thank you for highlighting this important point. We agree that a detailed comparison with the study by Farah et al.¹ would benefit readers and provide valuable insight into the reproducibility and potential differences in findings within the field of spatial omics. In response, we have conducted an in-depth comparison between our study and theirs, examining methodological overlaps as well as unique findings. We have added this comparison as Supplementary Discussion 2 to provide easier access for readers to our comparative analysis. We hope this addition will help clarify the scope and reproducibility of our work in relation to theirs.

In summary, a substantial portion of our findings on developmental cardiac cellular components and their spatial organization align with or could be validated by the study of Farah et al., despite differences in developmental periods (5.5th-14th vs. 9th-16th postconceptional weeks for the single-cell and 6th-12th vs. 12-13th and 15th postconceptional weeks for the spatial analysis), sample sizes in the single-cell (76,801 vs. 142,946 cells) and spatial (38 sections from 16 hearts for Visium and 9 sections from 4 hearts for *in situ* sequencing vs. 4 sections from 2 hearts for MERFISH analysis) datasets, and technical approaches (dissociation of whole hearts vs. selected regions, dissociation and cell enrichment method, sequencing and data processing strategy for single-cell analysis), integration between the spatial and single-cell datasets (spatial deconvolution vs. transcriptome imputation), these studies highlight the complementary strengths of whole-transcriptome and targeted spatial transcriptomics. MERFISH provides cellular-resolution *in situ* profiling but is limited by its target panel (238 genes in Farah et al.), while Visium offers unbiased spatial transcriptome analysis, albeit constrained to multicellular capture spots. Using Visium, we achieved greater granularity in mapping heterogeneous cardiac components, identifying novel or poorly characterized cell states not resolved in the MERFISH dataset. Conversely, MERFISH's cellular resolution captured molecular gradients within the ventricular walls less apparent in Visium data. These differences are also reflected in the applied niche discovery strategy (per-spot cell state co-detection vs. co-occurring cell states within individual cell zones of 150 μm radius), resulting in largely consistent results between the two studies on the level of major cell types, but with substantially higher granularity in the current study.

To reflect on these observations, we have included a shorter version of this paragraph in the Discussion (**lines 649-661** of the revised manuscript).

The methodology and data analysis are extensive and rigorous, and the presentation and writing is clear. The quality of the Visium and Chromium data is high and not overstated. The authors also provide in situ sequencing (ISS) data; however, these data appear to be of much lower quality and it is unclear to me how these data contribute in ways that could not be achieved with the Visium datasets. From visual inspection, these data seem to vastly underreport the actual number of transcripts, even when compared with high-dropout methods such as scRNA-seq. In order to assess the true capture rate of ISS, it would have been ideal if the authors could provide for one of the transcripts a low-dropout reference such as single-molecule RNA FISH, however that likely falls outside the scope of the current study. In lieu of such a reference standard, the authors should provide at least an estimate of the ISS capture rate (or inversely, the ISS dropout rate). Finally, the authors claim in the abstract that they performed “subcellular imaging-based transcriptomics”, which I assume refers to the ISS. I was excited to learn more about this feature of their data, however I saw no further mention of subcellular resolution in the main or supplementary text, nor any evidence for the subcellular resolution playing any role whatsoever in any of the results or conclusions. Thus, though the subcellular resolution may technically be an accurate descriptor, I found the use of this term in the abstract to set up misleading expectations for the rest of the article. Nevertheless, it is up to the authors whether they choose to keep this term in the abstract (as they did in their previous article in 2019).

Thank you for your thoughtful and detailed feedback. We appreciate your observations regarding our use of *in situ* sequencing (ISS) data. As noted, we introduced ISS analysis to validate the spatial distribution of selected gene targets with a higher spatial resolution than that achievable with standard Visium technology. The ISS-based figure panels in our manuscript highlight this effort, but we did not intend to interpret ISS patterns or perform analysis that imply subcellular resolution. To avoid setting misleading expectations, we have replaced the term "subcellular" with "high-resolution" in the Abstract, as we believe this better reflects the actual application of the ISS data in our study.

We also acknowledge the limitations of ISS in terms of capture efficiency and agree that a comparative low-dropout reference, such as single-molecule RNA FISH, would further contextualize our findings; however, this falls beyond the scope of the current study. We address the ISS capture rate limitations in our reply to Major Comment #2 both conceptually and even quantitatively, through the comparison to cluster-specific read numbers in our single-cell RNA-sequencing dataset. Thank you again for these valuable insights, which have helped us clarify our manuscript.

Major comments:

1. Given the similarities and seeming overlap between this study and the study by Farah et al. (ref 33), readers would benefit from a comparison of the overlap and differences between the two studies. What aspects of the previous study is reproduced here? Are there aspects of the results that appear irreproducible and/or contradictory? Though it is not the authors job to try and reproduce any specific aspect of the previous study, an effort should be made to place the current article's findings in the context this existing literature.

As mentioned above, we have performed an in-depth comparison of study design, technical approaches and major findings between our current study and Farah et al., which is available in Supplementary Discussion 2. We have also briefly reflected on the main findings of this analysis in the Discussion (**lines 649-661** of the revised manuscript).

2. Please provide an estimate of the ISS capture rate.

In lack of a low-dropout reference (e.g., smFISH), we evaluated the capture rate of our ISS dataset by comparing it with the complementary single-cell RNA sequencing (scRNAseq) dataset. To facilitate this comparison, we first identified common cell types in both datasets by intersecting cell type indices derived from the coarse-grained clustering of the scRNAseq analysis. We filtered both datasets to retain only these shared cell types. Next, we calculated the maximum gene expression values for the ISS and scRNAseq datasets (**Fig. 5A of Response to Reviewers**). These values were combined into a single DataFrame, stacked, and then reformatted to a long format with columns for 'method,' 'gene,' and 'expression.' We then computed the mean expression for each method, enabling a quantitative comparison between the spatial and scRNAseq data (**Fig. 5B of Response to Reviewers**).

Our analysis revealed that scRNAseq was 62.5 times more efficient than ISS across the 149 common genes assessed. This result aligns with the significantly lower capture rate of ISS compared to scRNAseq,

as reported in the literature. In this study, we primarily used the ISS data to validate spatial transcript patterns observed in the Visium dataset rather than for quantitative assessments. Therefore, the relatively low capture rate of ISS does not materially impact the conclusions of our analysis.

Figure 5. Comparison of maximal gene expression between the ISS and scRNAseq datasets. **A.** Scatter plot showing maximal expression values per gene in the ISS and scRNAseq datasets. **B.** Box plot showing maximal expression values per gene in the ISS and scRNAseq datasets, with the mean values calculated for the entire gene panel.

Minor comments:

1. p. 35 lines 622-631: Claiming that the study would have been improved and that rare cells would have been better captured with an increased size of the data set is lazy writing. Given the estimated drop-out rates for all methods used (scRNA-seq, ST, and ISS) it seems to me that improving the current methods would be a much better way to achieve these goals. As world-leading users (and inventors! thank you for your service) of some of these methods, it is imperative that the authors point out what aspects of these methods are most crucial to be improved – to me it seems like capture rate in all three, though especially in ISS; single-cell segmentation in ISS and in ST; library size in ISS; spatial resolution in ST.

Thank you for raising this point. We strongly agree that improvements to the methods used - particularly in capture rate, spatial resolution, and eventual cell segmentation - would significantly enhance data quality and the resulting insights. However, in our manuscript, we aimed to reflect on feasible approaches that were available at the time of data collection for this study. In that context, we still believe that an increased sample size, both for the single-cell and spatial transcriptomics (ST) datasets, would have provided additional support for identifying and refining cell states and spatial domains.

For instance, in the single-cell dataset, our novel insights largely involve rare cell states where increased sample size could better capture temporal dynamics and enable further characterization. Similarly, in the Visium and ISS datasets, the number of sections analyzed per heart sample (2-4) can not potentially capture the entire depth of spatial heterogeneity of the developing cardiac tissue. Additional samples and sections from the current, and even from an expanded time window would provide a more comprehensive

view of key cardiogenic events, including early endocardial cushion development and cardiac tube looping in the earlier, and ventricular wall maturation in the later developmental time points.

Nevertheless, we agree that advancements in the technologies themselves, as noted, would indeed offer substantial benefits. To acknowledge this, we have expanded the marked lines to highlight the potential of these technical improvements in advancing our understanding of cardiogenesis and related biological processes (**lines 661-665** of the revised manuscript).

2. p. 34 line 591: The authors claim their study is “the most detailed spatiotemporal atlas of heart development in the 1st trimester”. In what way is this study more detailed than the study by Farah et al. (ref 33)? Genetically or spatially? Please specify.

As we discuss in Supplementary Discussion 2, there are substantial differences in sample selection and experimental approach between the current study and the work of Farah et al., based on which we made the statement that our work reports a more detailed spatiotemporal view of 1st trimester cardiogenesis compared to this study and the current state of literature in the field. In terms of spatial analysis, Farah et al. uses the MERFISH technology to interrogate the expression of 238 selected gene targets in 4 sections, collected from 2 hearts representing 2 developmental stages from the second trimester (12th-13th and 15th postconceptional weeks). Our Visium spatial dataset is substantially larger, consisting of whole transcriptome-wide analysis of 38 heart sections, collected from 16 hearts between the 6th-12th postconceptional weeks, overlapping with the second half of the first trimester. We further supported some of the observed expressional patterns with *in situ* sequencing in 4 hearts from the same developmental window (6.5th, 8.5th, 9th and 11.5th postconceptional weeks), providing higher-resolution spatial account of 150 selected gene targets. In this sense, our spatial datasets are more detailed both genetically, spatially and temporally, allowing for the unbiased exploration of molecular dynamics in heart development, and also providing a better overlap with the period of the first trimester (0-12 postconceptional weeks), when the most prominent structural changes of cardiogenesis take place.

Nevertheless, we would like to highlight that the study of Farah et al. has some important advantages over our work. The utilized MERFISH technology allows to efficiently localize targeted transcripts with a subcellular resolution due to its relatively high capture efficiency, even compared to other imaging-based spatial transcriptomics approaches (such as ISS). Due to this characteristic of the selected method, Farah et al. efficiently profile a large number of cells *in situ* (258,237 cells across three experiments from 12-13 pcw hearts), but only collects information on spatial distribution of the preselected 238 gene targets. This approach allows for high resolution analysis of the distribution of selected molecular interaction partners (such as between members of the semaphorin and plexin families, explored by the authors); however, it also provides an intrinsically biased view on the molecular architecture of the developing cardiac tissue, and thus limits further explorative analysis of the collected data. Another important asset of the study of Farah et al. is the extensive single-cell RNA-sequencing dataset (142,946 single cells) which they leverage to impute gene expression profiles into their spatially identified cell types. This dataset, however, is collected from selected pieces of only 8 hearts from 9th-16th postconceptional weeks, representing a later developmental stage (with substantially higher total cell numbers per heart compared to earlier

developmental stages) compared to the dataset included in our study (76,991 high-quality cells after initial filtering of red blood cells, collected from 15 whole hearts between 5.5th-14th postconceptional weeks).

Considering all these factors we are still convinced that our dataset represents the most detailed spatiotemporal reference of the first trimester of human cardiogenesis.

3. In Figure 1, legend 1D-F are incorrectly labeled.

Thank you for bringing this error to our attention. We have now corrected the mislabelling in the legend of Figure 1 and added the following caption for panel D: ‘Relative enrichment of consensus cell type markers and selected differentially expressed genes ($\log_2FC > 0$, $p_val < 0.05$) across the 23 spatial clusters.’ (**lines 115-116** of the revised manuscript).

4. Ext. Fig 4D legend has a typo: “D. Gene Expression Related to Axon Guidance and Synapse Formation across Fine-Grained Cardiomyocyte States.”

Thank you for pointing out this issue. In fact, the referred figure legend lacks a caption for the image panel D, and thus the captions D, E and F are misaligned with the corresponding figure panels. We apologize for this error. Now we have added a caption for panel D as the following: ‘**D.** Ligand-receptor scores, adapted from a recently published neural-GPCR module of CellPhoneDB, highlighting different sources of adrenergic and cholinergic mediation of SAN_CM function between the Chrom_C and Aut_Neu_2 states.’ (**lines 67-69** of Extended Data Information), and correctly aligned the consecutive captions with the related figure panels.

5. Figure 5C: Is it certain that these detected transcripts are within endothelial cells? Are they also expressed in non-endothelial cells?

The transcripts (*PTHLH*, *WNT4*, *DKK2*, *CNRI*, *LGR5*, *BMP4*) displayed in **Fig. 5C** were selected based on their enrichment in one of the two fine-grained endothelial single-cell clusters (IF_VEC, OF_VEC) traced to opposite sides of the developing cardiac valves, compared to other fine-grained endothelial cell clusters. The ISS plots confirm largely consistent spatial arrangement of these transcripts close to the surface of the imaged valve structures, however, our target panel did not contain any general endothelial marker with high enough signal density that we could have used to spatially confirm the cell type specificity of these transcript within this experimental setting. As an independent confirmation, we assessed the expression of these transcripts in other valve-related cell states, with special focus on valve interstitial cells (VICs), which are in direct contact with the endothelial cells, and thus expression of the highlighted transcripts in this population could potentially contribute to the observed ISS signal. As visualized in a new heatmap included in **Fig. 5C** displaying the scaled expression of the six transcripts in question, transcripts enriched in the inflow side of the valves (*PTHLH*, *WNT4*, *DKK2*) show highly specific expression in the IF_VECs across all valve-related cell states, while the ones localized to the outflow side of the valves show strong enrichment in the OF_VECs, but are also expressed in VICs and the Valve_MC_1-2 populations, although at substantially lower levels. Based on these insights, we believe

that most of the ISS signal presented in **Fig. 5C** does in fact belong to the valve-related endothelial cell populations.

6. When integrating (and batch correcting) the single-cell and Visium data sets from different developmental time points (for instance for the purpose of clustering and UMAP plotting), how do you ensure that the integration procedure preserves developmental changes in cell states? (E.g. Suppl. Fig. 10B&C)

We thank you for raising this important question. Integrating multiple samples and/or sequencing batches poses a significant challenge, particularly when analyzing data across different time points. This necessitates a crucial decision for downstream analysis: whether to integrate only biological replicates (samples from the same time point) or to integrate the entire dataset (including all time points). Both approaches are valid but come with inherent trade-offs. Integrating only replicates ensures that the analysis captures cell states specific to each time point. However, this limits the ability to explore developmental transitions over time. On the other hand, integrating data across time points may obscure certain transcriptomic differences that reflect true biological changes, such as age-specific gene expression, but it enables an unbiased exploration of cell populations across the developmental timeline. After careful consideration, we opted for the second approach, integrating data from all time points to place the datasets within a common analytical framework. While this integration inevitably involves trade-offs - balancing the conservation of biological variation against the removal of batch effects - the transcriptomic signatures we detect post-integration are robust and form the foundation of our downstream analyses.

To further assess the robustness of our approach, we generated age-resolved UMAPs (**Suppl. Fig. 13B-C** of the revised manuscript). These plots allowed us to visually disentangle true temporal transitions from potential sample-specific separations on the UMAP. We recognize that more focused analyses, such as reprocessing clusters or splitting data by time points, may uncover additional molecular trends. To facilitate independent exploration, we have made the code used for data processing and analysis fully available, with preprocessed data provided upon request. This ensures transparency and allows others to delve deeper into specific aspects of the dataset if needed.

Reviewer #3:

Lázár and co-authors constructed a comprehensive spatiotemporal atlas of the developing human heart during the 1st and early 2nd trimester. It comprises 74 extensive datasets that include 69,114 spatially barcoded tissue spot transcriptomes and 76,801 single-cell transcriptomes from a total of 38 fetal human hearts, complemented by imaging 150 selected genes via in situ sequencing (ISS) and imaging select proteins using immunofluorescence. They identified 23 spatial tissue features with distinct transcriptional profile, 11 primary cell types and 72 fine-grained cell states that were mapped to corresponding regions in cardiac tissue sections by integrating spatial and single-cell transcriptomes and cell type deconvolution. This is a highly valuable data resource, in particular, for the understanding of early human heart development in the first trimester that has not been well characterized in the past. This data resource already allowed the authors to discover so many intriguing novel findings. For example, they dissected the spatial developmental process and the underlying molecular programs in the development of cardiac pacemaker-conduction system, the interactions with the autonomic innervation. They discovered a novel resident chromaffin cell population within the fetal heart that has implication in cardiac diseases. They dissected spatially heterogeneous endothelial and mesenchymal cell types and states in the developing valves and atrial septum. Finally, they constructed a map of spatial niches by examining the co-occurrence of different cardiac cell states. This highly valuable data resource is available through an interactive data portal, which will definitely increase the accessibility of these datasets and foster the utilization for independent discovery in relation to heart development and diseases. It is worth noting that the manuscript is beautifully written and figures well organized, making it really a joy to read and the key messages clearly delivered. In sum, this work provides a highly valuable and timely data resource, and yields numerous new insights that are of high interest to the field. This reviewer has only a few minor comments as the following.

Thank you for your thoughtful and thorough review. We greatly appreciate your positive feedback and are pleased that you find our work valuable and insightful. Your recognition of the dataset's potential for further discoveries in heart development and disease is especially encouraging. We have addressed your minor comments carefully to further enhance the clarity and quality of the manuscript.

(1) The authors have previously investigated human heart development at different stages (REF 32) and other studies in field also reported spatial niches and cell types in developing or adult human heart (e.g., REF 33, 42). Could the authors also examine the data from these published works to either perform integrative analysis to extend the findings of key cell types and states at the first trimester to later stages or compare selected cell types or niches between fetal and adult heart. Even comparing the results without full integration could be still enlightening. Thus, this is a missed opportunity for the authors to uncover more insights.

We appreciate your insightful suggestion and agree that extending our findings to later stages of heart development or comparing them to adult heart datasets could provide valuable insights. Our work is an addition to the steadily growing body of spatial omics-based investigations in the cardiac field and builds upon the foundational work of our group presented in Asp et al.⁶. This current study represents a

significant expansion, benefiting from the substantially larger single-cell and spatial datasets and the technical advancements in spatial transcriptomics technology (**Table 1 of Response to Reviewers**). These improvements have enabled us to explore the cellular and molecular mechanisms of early cardiogenesis with previously unattainable molecular detail and spatiotemporal resolution. The massive difference in granularity, both in terms of single-cell and spatial clusters, makes it difficult to make an in-depth comparison between the findings of the two papers. Nevertheless, our new analysis supported and elaborated on some of the most novel observations of the earlier study, such as the presence of *FABP3-MYOZ2*-enriched cardiomyocyte populations, more prominent enrichment of *TCF21* in epicardium-derived cells (in comparison to epicardial cells), and spatial heterogeneity between fibroblast-like cell states. Additionally, it also helped to clarify the identity of the endocardial cell population, which, in lack of reliable single-cell reference datasets at the time, was mislabelled as capillary endothelium in the earlier study.

Study		Asp et al., 2019, Cell	Lázár et al., in review
Technology		10x Chromium	10x Chromium
Size	# Hearts	1 (whole)	15 (whole)
	# Cells	3,717	76,991
	# Clusters	15	72 (w/o RBCs)
Developmental stage		7 pcw	5.5-14 pcw
Technology		1 st generation ST	10x Visium
Size	# Hearts	3	16
	# Sections	19	38
	# Spots	3,115	69,114
	# Clusters	10	23 clusters, 14 regions
Developmental stage		4.5, 6, 9 pcw	6-12 pcw
Spatial resolution	Spot \varnothing	100 μm	55 μm
	Spot \leftrightarrow	200 μm	100 μm
Gene resolution		Whole transcriptome	Whole transcriptome
Technology		ISS	ISS
Size	# Hearts	3	4
	# Sections	8	9
	# Clusters	15	-
Developmental stage		4.5, 6, 9 pcw	6.5-11.5 pcw
Spatial resolution		Cellular	Cellular
Gene resolution		69 target genes	150 target genes

Table 1. Comparison between datasets presented in Asp et al.⁶ and the current study.

Early human cardiogenesis has been in the limelight of several recently published works and preprints, including the article by Farah et al.¹. To this date, this study has the highest similarity with our current work in terms of the analyzed period of cardiogenesis and the analysis strategy, combining single-cell and spatial transcriptomics technology. Therefore, we believe that an in-depth comparison between these two works provides the highest value for the scientific community, and in line with your and Reviewer #2's request, we performed integrated analysis of these two works, discussed in detail in Supplementary

Discussion 2. Because of technical challenges of integrating spatial omics datasets using two different spatial principles, we performed this comparison using a single-cell data-driven approach but utilizing the insights from our spatially aware annotation strategy and the MERFISH-based spatial viewer of Farah et al., we extensively comment on similarities and differences between the spatial components of these two works.

While undoubtedly relevant, studies describing molecular patterns of the adult heart do not reflect the unique events of early heart formation, and direct integration of developmental and adult datasets is especially challenging, due to substantial biological (profound changes in transcriptomic profiles of cardiac cells) and technical discrepancies (analysis of aborted tissue vs. cardiac biopsies or postmortem tissue). Based on these considerations, we believe that the most informative way to compare the foundational work of Kanemaru et al.⁴, describing spatial multiomics-based analysis of cardiac niches in the adult heart, is on the level of results drawn separately from the two datasets. We have included several such observations in our manuscript related to the molecular characteristics of the developing pacemaker-conduction system (**lines 210-248** of the revised manuscript), or similarities between the CALN1^{high}_FB state and a glial cell state in the adult sinoatrial node (**lines 536-538** of the revised manuscript). While further assessment of this work is beyond the scope of the current study, the rapidly expanding number of developmental datasets will allow for a more robust and systematic comparison of molecular characteristics between the adult and developing human heart tissue and its components, less prone to technical biases between different studies.

(2) Similarly, although it is extremely interesting to show numerous CHD or other CVD risk genes are associated with cell types in the first trimester of the developing human heart. Could the authors compare spatial molecular profiles or cell types associated with these genes between this work and published studies of human CVD single-cell or spatial maps. It does not need to fully integrated data analysis, but comparing findings in development and disease would be still quite insightful.

Thank you for appreciating our effort linking cardiac pathology-related gene panels and specific genes to developmental cell types identified in our study. We agree that comparing spatial molecular profiles or cell type associations between our developmental and published cardiac disease-related transcriptomics datasets would indeed provide valuable insights, however, there are certain technical challenges regarding this effort.

As highlighted by our gene panel enrichment results, specific genes are often linked to several cardiovascular diseases (CVD) in GWAS studies, thus this type of analysis rarely yields highly specific results in terms of affected cell states. Currently available transcriptomics datasets related to CVD are predominantly generated from postnatal or even adult heart specimens, even in case of diseases which fall under the category of congenital heart diseases (CHD), likely due to easier access to such tissue samples. Additionally, these datasets are only representative of the analyzed tissue regions and feature substantial age-related changes both in terms of cellular composition and molecular profiles compared to developmental samples. For these reasons, more accurate insights might be drawn from systematic comparisons with age-matched healthy controls or reference datasets, even in cases of developmental

anomalies, such as performed by Hill et al.⁷. Moreover, most CHD cases arise from abnormal developmental processes during the first trimester. As a result, the critical transcriptomic landscape that influences the formation of heart defects has typically already dissipated by the time these anomalies are clinically discovered. Finally, alterations on the protein-coding genome captured in transcriptomic datasets does only explain a small fraction of CHD cases, and an increasing number of publications recognize the importance of mutations in the non-coding regions of the genome⁸, affecting promoters or enhancers, in the development of these pathologies. While our developmental dataset can effectively support the validation of such findings, it requires significant additional analysis which is beyond the scope of the current study.

Importantly, the potential of our dataset to support the delineation of genetic contributors to CHD has been highlighted by a recent preprint, describing the enrichment of risk variants for congenital heart defects on valve interstitial cells (VICs) in the developing heart⁹. This study extensively leverages the molecular marker profiles of our spatially annotated cell states to identify cell populations within their single-cell multiomic dataset, and connect rare coding and common non-coding variants associated with CHD and valve traits to valvular interstitial cells (VICs), shifting the focus from working cardiomyocytes as the primary cellular contributors to CHD to the involvement of other cell types.

(3) Figure 2b – single-cell clustering UMAP. Cardiomyocytes, endothelial cells, are shown in two distinct regions in UMAP. Immature CMs look similar to EC. Lymphatic ECs are located next to CMs. Although the authors have carefully annotated and confirmed the cell types, but it is still very interesting why these totally different cell types localized in the “vicinity” of the UMAP space. Could this be attributed to cell doublet or RNA cross contamination, or something biologically interesting?

Thank you for this valuable comment. Lymphatic endothelial cells (LECs) usually feature a very distinct transcriptomic profile, thus their clear separation from other endothelial cell subtypes in the UMAP space is not unexpected. On the other hand, it is indeed notable that the cluster annotated as immature cardiomyocytes (Immat_CM) appears closer to the endothelial cell (EC) clusters than to cardiomyocyte (CM) clusters in the UMAP 2D representation. We have taken careful measures to ensure the validity of our analysis and address the potential concerns raised.

First, all identified doublets were rigorously filtered out using the DoubletFinder (v2.0.3) tool¹⁰, as described in the Methods section (“Processing and Filtering of the Single-Cell RNA Sequencing Data”; **lines 902-905** of the revised manuscript). This step minimizes the likelihood of doublets contributing to the observed cluster proximity. Second, despite the spatial proximity in the UMAP 2D space, the Immat_CM population did not show enrichment of any consensus marker of non-cardiomyocyte cell types (**Fig. 2C**), which effectively rules out substantial RNA cross-contamination between these populations. Finally, we acknowledge the inherent limitations of dimensionality reduction techniques like UMAP, particularly when visualizing high-dimensional data in a compressed 2D space. To further investigate the spatial relationships, we examined the UMAP clustering in a 3D projection. This analysis revealed that the Immat_CM cluster occupies an intermediate position between the larger cardiomyocyte and endothelial cell clusters, providing a more nuanced understanding of its localization. Although it is

challenging to represent this finding in a static figure, this 3D perspective reinforces our conclusion that the observed 2D proximity does not imply biological similarity or contamination.

Figure 6. Relative positions of the Immat_CM, other cardiomyocyte and endothelial cell coarse-grained clusters in the 3D UMAP projection from various angles (A-D).

In summary, the proximity of Immat_CM to endothelial cell clusters in the UMAP 2D space likely reflects limitations of the dimensionality reduction visualization rather than biological or technical artifacts. This observation highlights the importance of complementary analyses to interpret such spatial relationships accurately. Additionally, the distinct temporal pattern of the Immat_CM cluster within the cardiomyocyte subset (**Fig. 2F**, uppermost panel, discussed in **lines 88-91** of Supplementary Discussion 1) underscores the biological significance of distinguishing this population, featuring lower maturation state and gradually decreasing proportion, from other cardiomyocyte clusters.

(4) Fig 3d, three or four different genes are shown in the same color but varying intensities. This is difficult to see and distinguish them by naked eyes. Could different colors be chosen to visualize the spatial distributions of these genes in this figure panel.

Thank you for highlighting the difficulty of distinguishing different transcripts in the ISS panels included in **Fig. 3D**, a concern also raised by Reviewer #1. Due to figure size limitations and the spatial enrichment of the visualized transcripts in the sinoatrial and atrioventricular nodal regions, we were not convinced that changing the colors for different transcripts would satisfactorily resolve this issue. Instead, we have introduced a new supplementary figure with independent ISS panels for each transcript visualized in **Fig. 3D (Suppl. Fig. 9A)** and **Fig. 3E (Suppl. Fig. 9B)**, enabling clearer assessment of their spatial distributions and mitigating the challenge of interpreting overlapping or similarly shaded signals.

(5) Ion channel profile is super interesting! Could the author further ion channel mediated cell-cell interaction or action potential propagation? No one has done that and this is really optional.

Thank you for your encouraging comment and for appreciating the focus on the ion channel profiles of CPCS cells in our manuscript. While we agree that exploring ion channel-mediated cell-cell interactions or action potential propagation in greater depth would be highly valuable, we found it challenging to extend our analysis further for several reasons. Our current dataset is based on transcriptomic analysis, and the relationship between RNA expression and functional protein levels is often non-linear. Post-transcriptional and post-translational regulation, such as protein degradation and translational control, can significantly influence the repertoire of functional ion channels and gap junction proteins. Additionally, many ion channels and gap junction proteins function as part of larger molecular complexes. The presence and assembly of these complexes depend on additional subunits or interacting partners, which are not fully captured in our RNA-level analysis. Finally, the activity of ion channels and gap junction proteins is modulated by multiple factors, including membrane potential changes, intracellular calcium concentration, and signaling pathways, adding another layer of complexity beyond gene expression data.

Although these limitations prevent us from building a precise model for action potential generation and electrical impulse propagation in the developing heart, we have highlighted several key molecular features of cardiomyocyte cell states contributing to these processes in and beyond the pacemaker-conduction system, and apparent differences compared to adult equivalents of these cell types, as reported by Kanemaru et al.⁴ (**lines 210-248** of the revised manuscript). Addressing this question in more depth would require integrating electrophysiological data with transcriptomics, which is beyond the scope of this study. However, adaptations of technologies like Patch-seq, which enable simultaneous electrophysiological, transcriptomic, and morphological characterization in neurons, could offer exciting opportunities for this type of investigation in future studies on cardiomyocytes, using animal models.

(6) Fig 4c,d – usually PCA has limited resolution to differentiate different cell subsets to investigate temporal dynamics in pseudotime or RNA velocity analysis. Could you also try UMAP or monocle, and compare?

Thank you for this insightful comment. Pseudotime and RNA velocity analyses indeed present significant challenges, especially in datasets of limited size (n=870 in this case) and we have carefully explored various methodologies to derive meaningful insights. As presented in the manuscript (**Fig. 4C**), we opted for visualizing the developmental trajectories on a PCA plot supported by differentiation marker

embeddings (**Fig. 4D**, **Suppl. Fig. 10**), with the observed patterns well-aligning with our current understanding on glial and sympatho-adrenal differentiation paths. For direct comparison with the PCA embedding (**Fig. 7A of Response to Reviewers**), the trajectory computed on the UMAP embedding is visualized below in **Fig. 7B**. As apparent, the UMAP-based analysis did not yield biologically meaningful insights into potential developmental differentiation, further supporting the choice of PCA for this dataset.

Initially, we tested tools such as Velociraptor¹¹ and Slingshot¹² but found their results to be inconclusive for our dataset. Subsequently, we employed pseudotime and RNA velocity analyses using tools based on the spliced/unspliced transcript ratio, including scFates² and scVelo³. While these analyses were initially performed on the UMAP embedding, several limitations rendered this approach suboptimal. Specifically, the innervation-related cellular subset was separated into distinct clusters with largely separate positions on the UMAP, due to the inherent smoothing effect of this embedding. This fragmented appearance of the data complicated the identification of dynamic relationships between individual cells, obscuring more subtle transitions along the investigated trajectories.

Figure 7. Trajectory analysis of the innervation-related cell subset, based on **A**, PCA or **B**, UMAP embedding.

Given these limitations, we utilized the principal components (PC1 and PC2) instead (**Fig. 7A of Response to Reviewers**), which effectively separated the dataset into the two expected trajectory branches, with placing Schwann cell progenitors (SCPs) at the root of the developmental trajectories. Unlike UMAP (**Fig. 7B of Response to Reviewers**), the PCA representation reduced biases related to cluster identities, with immature cells localized closer to the root and more mature cells progressively distributed along the branches, aligning with the distribution of known differentiation markers. Furthermore, using PCA mitigates the interpretational challenges associated with UMAP-based trajectories, particularly when clusters are spatially distant in the 2D space. In such cases, the assumption that neighboring clusters reflect a developmental continuum is less convincing. By contrast, PCA provided a more robust framework for visualizing and interpreting the displayed trajectories.

(7) Fig 6. It is interesting to see the development of multiple types of valve-related endothelial, fibroblast, and mesenchymal cells, in particular, the difference in endothelial cells on the opposite sides of the valve

leaflet, which is likely attributed to different flow and mechanical stimulation. Another dimension to investigate the valvular cell heterogeneity is from the base to the tip. In particular, Fig 5f, using ISS, one can resolve more spatial features of valvular cells.

We agree that the region of the cardiac valves is especially interesting in terms of molecular and cellular regionalization, and our spatial datasets provide useful insight into exploring this heterogeneity. As you highlighted, our work expands on different valve endothelial cell populations that determine side-specific molecular patterns in the cardiac valves, but importantly, we also highlight the regionalization of some of this signal on the ‘inflow’ side of the valves, showing further spatial enrichment of relevant transcripts in the coaptation region of the valves. This observation is in line with previous works describing similar regionalization in mouse heart valves during postnatal maturation, with valve endothelial cells in the coaptation zone of aortic and mitral valves showing enrichment of key markers of the IF_VEC population described in our study¹³. Additionally, our analysis identified three mesenchymal cell populations in the cardiac valves with different localizations, with valve interstitial cells (VICs) present in the free segments of the atrioventricular and semilunar valves, the Valve_MC_2 population localizing to the roots and the surrounding regions of the internal annulus fibrosus, and Valve_MC_1 cells showing enrichment in the roots of and around the semilunar valves. (For further insights, please find our reply to related questions of Reviewer #1). This cellular arrangement contributes to base-to-root molecular diversity in the valves, which is showcased by the ISS-based detection of two defining markers of the Valve_MC_2 and VIC populations, *PENK* and *LEF1*, showing regionalized enrichment closer to the roots and tips of the sampled atrioventricular valves, respectively (**Fig. 6G**). (In the sampled semilunar valves, the tips were not captured due to an angled sectioning plane, and while a similar molecular arrangement can be suspected, it could not be clearly visualized in this sample.)

These two examples highlight complementary strengths of the unbiased molecular profiling by the Visium platform, and the higher-resolution spatial data provided for selected targets by the ISS technology. However, one important limitation of the presented ISS dataset is the overall low signal density, which makes high-granularity, segmentation-based cell identification unfeasible in our analysis. On the other hand, it is still possible to observe transcripts with apparent regionalization or gradients across the captured valve structures in this dataset, even beyond the ones discussed above (**Fig. 8 of Response to Reviewers**). For example, *NSG1* and *BMPEP* appear along both valve sets in the endothelial layer without any apparent regionalization on the base-to-tip axis, while several other transcripts show enrichment in the roots of the valves, such as *RSPO3* in both valve sets, *SEMA3D* in the semilunar valves, and *ID4* in the root of atrioventricular valves, along with extended expression in the semilunar valves, among others. Based on insights from our single-cell dataset, these latter transcripts are predominantly derived from various mesenchymal cell components of the valves. Additionally, one can observe cardiomyocyte marker genes (*HSBP7*, *CORIN*) in the atrioventricular valves, arranged in a distinct layer in the subendocardium on the ventricularis side, thus contributing to the side-to-side heterogeneity of the atrioventricular valves. This signal likely corresponds to cardiomyocyte remnants present after the developmental delamination of

these structures from the ventricular walls, a population that was not separately identified by the single-cell analysis and would have been overlooked by the Visium analysis due to its lower spatial resolution.

Figure 8. Regionally enriched transcripts in the cardiac valves of a 9 pcw heart section. Slv - semilunar valves, avv - atrioventricular valves.

Taken together, high-resolution spatial analysis of the cardiac valves, such as presented in our ISS dataset, has the potential to furnish an even more detailed understanding of the molecular structure of the cardiac valves. Importantly, targeted sampling of both valve sets in various developmental stages will be essential for future studies to reliably assess differences and similarities in spatiotemporal molecular patterns in these structures.

References:

1. Farah, E. N. *et al.* Spatially organized cellular communities form the developing human heart. *Nature* (2024) doi:10.1038/s41586-024-07171-z.
2. Faure, L., Soldatov, R., Kharchenko, P. V. & Adameyko, I. scFates: a scalable python package for advanced pseudotime and bifurcation analysis from single-cell data. *Bioinformatics* **39**, btac746 (2023).
3. Bergen, V., Lange, M., Peidli, S., Wolf, F. A. & Theis, F. J. Generalizing RNA velocity to transient cell states through dynamical modeling. *Nat. Biotechnol.* **38**, 1408–1414 (2020).
4. Kanemaru, K. *et al.* Spatially resolved multiomics of human cardiac niches. *Nature* **619**, 801–810 (2023).
5. Hansen, J. N. *et al.* Intrinsic Diversity In Primary Cilia Revealed Through Spatial Proteomics. *Cell Biology* (2024).
6. Asp, M. *et al.* A Spatiotemporal Organ-Wide Gene Expression and Cell Atlas of the Developing Human Heart. *Cell* **179**, 1647–1660.e19 (2019).
7. Hill, M. C. *et al.* Integrated multi-omic characterization of congenital heart disease. *Nature* **608**, 181–191 (2022).
8. Richter, F. *et al.* Genomic analyses implicate noncoding de novo variants in congenital heart disease. *Nat. Genet.* **52**, 769–777 (2020).
9. Ma, X. R. *et al.* Molecular convergence of risk variants for congenital heart defects leveraging a regulatory map of the human fetal heart. *medRxiv* (2024) doi:10.1101/2024.11.20.24317557.
10. McGinnis, C. S., Murrow, L. M. & Gartner, Z. J. DoubletFinder: Doublet Detection in Single-Cell RNA Sequencing Data Using Artificial Nearest Neighbors. *Cell Syst* **8**, 329–337.e4 (2019).
11. Rue-Albrecht, K., Lun, A., Soneson, C. & Stadler, M. *Velociraptor: Toolkit for Single-Cell Velocity. R Package Version 1.* (2024).
12. Street, K. *et al.* Slingshot: cell lineage and pseudotime inference for single-cell transcriptomics. *BMC Genomics* **19**, 477 (2018).
13. Hulin, A. *et al.* Maturation of heart valve cell populations during postnatal remodeling. *Development* **146**, dev173047 (2019).

Reviewer #1:

Remarks to the Author:

The authors have extensively addressed all my comments.

We would like to thank the Reviewer again for their time and energy invested in evaluating our manuscript.

Reviewer #2:

Remarks to the Author:

The authors have done a fantastic job of thoroughly answering all the concerns. In particular, the supplementary discussion of a comparison to the Farah et al. study will be of great use to the field as it addresses inconsistencies that would otherwise have been passed on to scientists exploring and comparing these two data sets in the future. I found Supplementary Discussion 2 to be highly informative in how it addresses the specific aspects of the cell annotations that differ between the two studies. The ISS capture rate comparison was also useful and helps in interpreting the authors' results.

We sincerely thank the Reviewer once again for the time and effort dedicated to evaluating our manuscript. We are pleased that the Reviewer found our comparative analysis with the dataset of Farah et al. informative and of general use to the field. We also agree that the comparison between the ISS and scRNAseq capture rates provide relevant insights for the evaluation of the presented datasets.

I have two remaining comments/requests and I hope to see the article in print soon.

1) Fig. 5A&B of Response to Reviewers is extremely useful for interpreting the ISS data in the context of other modalities such as scRNAseq. These two figure panels should therefore be part of the manuscript.

We have included these figure panels as Supplementary Figure 15A-B in the newly revised version of our manuscript, and added the following text to the relevant part of the new Supplementary Methods file: 'Capture efficiency of ISS was assessed relative to that of scRNAseq, by calculating maximum gene expression values in coarse-grained cell types shared between the two datasets (Suppl. Fig. 15A) and comparing the mean expression for each method, revealing that scRNAseq was 62.5 times more efficient than ISS across the 149 common genes assessed (Suppl. Fig. 15B).

2) From looking at Fig. 5B of Response to Reviewers, it appears that the mean of the max expression for ISS is less than $5E-2$, and the mean of the max expression for scRNAseq is more than $5E0$, which means that the difference is greater than 100-fold. However, in the response letter the authors state that "Our analysis revealed that scRNAseq was 62.5 times more efficient than ISS across the 149 common genes assessed." Am I reading the graph wrong? In either case, the correct value should be reported in the manuscript as it is an important metric relevant to any interpretation of their data.

We thank the Reviewer for drawing our attention to this issue, which most likely stems from ambiguity regarding the calculation and visualization of relative capture efficiency between the ISS and scRNAseq datasets. The reported 62.5 times difference was, in fact, calculated based on the mean of the maximal expression values per gene in coarse-grained cell clusters shared between the ISS and scRNAseq datasets. On the figure panel, however, the horizontal line within the boxplot represents the median, and not the mean value, as commonly applied in such plots. To enhance clarity, we have added the following figure legend to this image panel, included as Supplementary Figure 15B in the newly revised manuscript: ‘Box plot showing maximal expression values per gene in coarse-grained cell clusters shared between the ISS and scRNAseq datasets, with the mean values calculated for the entire gene panel for summative quantification. The horizontal center line indicates the median, the box edges represent the upper and lower quartiles, and the whiskers extend to 1.5 times the interquartile range for each experimental setting.’

Reviewer #3:

Remarks to the Author:

The authors have done an excellent job in the revision that has fully addressed all my major questions. In particular, the comparison of key observations between Farah et al, and this manuscript really helped strengthened the key findings and the further analysis of spatial cell heterogeneity across the valves elucidated the potential mechanisms in line with biomechanics and physiological functions. I also took the liberty to briefly read the responses to other reviewers. Reviewer 2 and I raised some common concerns that have been well addressed. Also, agree with Reviewer 1 that this work presents the most comprehensive and informative datasets to dissect the spatial temporal dynamics of human heart development. Thus, this is a highly valuable resource article and reports a significant contribution to the field. I would recommend accept as it.

We thank the Reviewer for their positive feedback on the manuscript’s overall content and the scientific value of the presented resources. We are grateful for their thoughtful comments, which have contributed to strengthening our work.